



# ENSO and internal sea surface temperature variability in the tropical Indian Ocean since the Maunder Minimum

Maike Leupold[1], Miriam Pfeiffer[2], Takaaki K. Watanabe[3], Lars Reuning[2], Dieter Garbe-Schönberg[2], Chuan-Chou Shen[4,5,6], Geert-Jan A. Brummer[7]

[1]EMR-Group, Geological Institute, RWTH Aachen University, Aachen, 52062, Germany
[2]Institute of Geosciences, Kiel University, Kiel, 24118, Germany
[3]Department of Natural History Sciences, Faculty of Science, Hokkaido University, Sapporo, 060-0810, Japan
[4]High-Precision Mass Spectrometry and Environment Change Laboratory (HISPEC), Department of Geosciences, National Taiwan University, Taipei, 10617, Taiwan ROC
[5]Research Center for Future Earth, National Taiwan University, Taipei, LC6L73, Taiwan ROC
[6]Global Change Research Center, National Taiwan University, Taipei, 10617, Taiwan, ROC
[7]Department of Ocean Systems, Royal Netherlands Institute for Sea Research (NIOZ), and Utrecht University, 1790 AB Den Burg, The Netherlands

*Correspondence to*: Maike Leupold (maike.leupold@emr.rwth-aachen.de)

**Abstract.** The dominant modes of climate variability on interannual timescales in the tropical Indian Ocean are the El Niño Southern Oscillation (ENSO) and the Indian Ocean Dipole. El Niño events have occurred more frequently during recent decades and it has been suggested that an asymmetric ENSO teleconnection (warming during El Niño events is stronger than cooling during La Niña events) caused the pronounced warming of the western Indian Ocean. In this study, we test this hypothesis using coral Sr/Ca records from the central Indian Ocean (Chagos Archipelago) to reconstruct past sea surface temperatures (SST) in time windows from the Maunder Minimum to the present. Three sub-fossil massive *Porites* corals were dated to the 17-18th century (one sample) and 19-20th century (two samples), and were compared with a published, modern coral Sr/Ca record from the same site. All corals were sub-sampled at a monthly resolution for Sr/Ca measurements, which were measured using a simultaneous ICP-OES. All four coral records show typical ENSO periodicities, suggesting that the ENSO-SST teleconnection in the central Indian Ocean was stationary since the 17th century. To determine the symmetry of ENSO events, we compiled composite records of positive and negative ENSO-driven SST anomaly events. We find similar magnitudes of warm and cold anomalies indicating a symmetric ENSO response in the tropical Indian Ocean. This suggests that ENSO is not the main driver of central Indian Ocean warming.

## 1 Introduction

In times of increasing impacts of global climate change, paleoclimate research seems more important than ever. Especially the Indian Ocean is of major relevance to global ocean warming as the western Indian Ocean has been warming faster than any



other ocean basing during the last century and is the largest contributor to the current rise of global mean sea surface temperatures (Roxy et al., 2014).

As tropical corals can be used to reconstruct past changes of environmental parameters, such as sea surface temperatures (SST),
by measuring Sr/Ca, they can help to determine changes in past climate variabilities. Most coral paleoclimatological studies covering periods before 1900 conducted in the tropical Indian Ocean predominantly focused on $\delta^{18}O$ measurements (e.g. Abram et al., 2015; Charles et al., 2003; Cole et al., 2000; Nakamura, et al., 2011; Pfeiffer et al., 2004). There are only some studies including Sr/Ca measurements for SST reconstructions (e.g. Pfeiffer et al., 2006), while few studies included Sr/Ca measurements for SST reconstructions (e.g. Pfeiffer et al., 2006). Most studies are focusing on either the western or the eastern
Indian Ocean (Abram et al., 2003; Watanabe et al., 2019) and/or are sampled at only bimonthly (Zinke et al., 2004; Zinke et al., 2008) or annual resolution (Zinke et al., 2014; Zinke et al, 2015). The lack of data in the central, tropical Indian Ocean still limits the understanding of the relationship between interannual and decadal climate variabilities in the tropical Indian Ocean associated with transregional or global climate phenomena.

Strong El Niño Southern Oscillation (ENSO) events occur more frequently since the early 1980s (Baker et al., 2008; Sagar et
al., 2016) and have a strong influence on the tropical Indian Ocean demonstrating an existing stable SST-ENSO teleconnection between the Pacific Ocean and Indian Ocean (Pfeiffer & Dullo, 2006; Wieners et al., 2017). In fact, it was suggested that El Niño events have a stronger influence on the Indian Ocean than La Niña events (Roxy et al., 2014). In their study, Roxy et al. (2014) propose that El Niño events cause strong warming of the western Indian Ocean, whereas La Niña events do not cause significant cooling of the region. This asymmetric ENSO teleconnection has been suggested to contribute to the overall
warming of the tropical Indian Ocean. Here, we test this hypothesis using three sub-fossil massive *Porites* coral samples and one modern coral core from the central Indian Ocean (Chagos Archipelago) to reconstruct past SST. The modern core was included in a composite reconstruction of large-scale SST (Pfeiffer et al., 2017) and the core top (1950-1995) was shown to record SST variability at Chagos on grid-SST scale (Pfeiffer et al., 2009). The sub-fossil corals record 41 years of the Maunder Minimum (1675-1716), 31 years of the late Little Ice Age (1836-1867) and 39 years of the mid-19th to early 20th century
(1870-1909) covering 39 years. We identify past warm and cold events in each record and use these events to compile composites to evaluate the symmetry of positive and negative ENSO-driven SST anomaly events in the tropical Indian Ocean.

## 2 Regional setting

### 2.1 Location

The Chagos Archipelago is located in the tropical Indian Ocean, about 500 km south of the Maldives. It consists of several
atolls with islands, submerged and drowned atolls, and other submerged banks with the Great Chagos Bank being the world's largest atoll (Fig. 1). The Great Chagos Bank covers an area of 18.000 km² with eight islands totaling 445 ha of land. Its lagoon has a maximum depth of 84 m and a mean depth of 50 m. Due to its large size and submerged islands, water exchange with the open ocean is substantial. The Salomon atoll is located about 135 km towards the northeast of Eagle Island. Its atoll area



is about 38 km$^2$ and has an enclosed lagoon and an island area > 300 ha. The greatest depth of its lagoon is 33 m, with mean
depth of 25 m.

## 2.2 Climate

Chagos is situated in a region characterized by monsoon climate (Sheppard et al., 2012). The austral summer is the wet season, with the Northeast monsoon lasting form October to February (Pfeiffer et al., 2004). From October to April, light to moderate north-west trades blow. During the rest of the year, strong winds from the southeast dominate (Sheppard et al., 1999).

Chagos lies at the eastern rim of the so-called Seychelles-Chagos thermocline ridge (SCTR). Along that region, a shallow thermocline causes open-ocean upwelling of cold waters. Upwelling along this region was first identified by McCreary et al. (1993) and is forced by both negative and positive wind stress curl (Hermes & Reason, 2009; McCreary et al., 1993). Compared to other upwelling regions of the Indian Ocean, the sea surface temperatures of the SCTR are relatively high. They vary between 28.5°C and 30°C in austral summer. The SCTR is believed to play a major role in the climate variability of this region

on different timescales (e.g. Hermes & Reason, 2008; Vialard et al., 2009) with very strong air-sea interaction due to open ocean upwelling combined with relatively warm SST (Sheppard et al., 2012).

On interannual timescales, the dominant mode of climate variability in the SCTR is the El Niño Southern Oscillation (ENSO). During El Niño events, the West Pacific warm pool is displaced towards the East resulting in cooler than normal SST in the Western Pacific and a basin-wide warming of the Indian Ocean (Izumo et al., 2014; Sheppard et al., 2013). Figure 2 compares

the positive SST anomalies during El Niño with the negative SST anomalies during La Niña events in the Indian and Pacific Ocean between 1982 and 2016, as inferred from 'Reynolds' OI v2 SST data (Reynolds et al., 2002; averaged over December-February). Even if not as strong as in the Pacific Ocean, an ENSO response in the tropical Indian Ocean can be observed (Fig. 2). Coupled ocean-atmosphere instabilities centered in the tropical Indian Ocean result in Indian Ocean Dipole (IOD) events (Saji et al., 1999; Sheppard et al., 2013; Webster et al., 1999). A negative (positive) IOD event is defined by warmer (cooler)

than normal SST in the eastern part of the Indian Ocean and cooler (warmer) than normal SST in the western Indian Ocean. Several studies showed that the IOD is an inherent mode of variability of the Indian Ocean (e.g. Ashok et al., 2003; Krishnaswamy et al., 2015; Saji et al., 1999; Webster et al., 1999). However, IOD events tend to co-occur with ENSO events (e.g. Luo et al, 2010; Saji and Yamagata, 2003). The instrumental record of past IOD events does not go back further than 1960 (Saji and Yamagata, 2003). A coral-based reconstruction of past IOD events extends until 1846 and suggests a recent

intensification of the IOD (Abram et al., 2008). The coral index shows few strong IOD events in the 20[th] and late 19[th] century (i.e. 1997/98, 1961 and 1877/78), of which only the event in 1961 is independent of ENSO (Abram et al., 2008).

We therefore decided to treat positive SST anomaly events found in our records as El Niño events even if they could be a result of IOD events independent from or overlapping with ENSO events.



## 2.3 Instrumental data

High-resolution SST data of the AVHRR satellite product (Casey et al., 2010) reveal different mean SST and seasonality at
Chagos depending on the reef setting (Leupold et al., 2019; Fig. 3). At the open ocean reefs, where ocean upwelling occurs,
seasonal minima in SST are more pronounced than in the lagoon, whereas maximum temperatures are not significantly
different (t-value = 0.27; p-value = 0.79). Averaged over the entire area of the Chagos (70-74° E; 4-8° S), SST is similar to
SST measured in the lagoon. Long-term monthly SST anomalies (i.e. mean seasonal cycle removed) reveal that extreme SST

events, such as El Niño in 1997/98 or La Niña in 2010/11, have the same magnitude in both lagoon and open ocean settings
(Fig. 3b). Both anomaly records are not significantly different (t-value = 0.34; p-value = 0.37). This suggests that the
magnitudes of ENSO signals at Chagos should be recorded in all coral records analyzed in this study, as it is independent from
the reef setting.

## 2.4 ENSO indices

The instrumental record of past ENSO events is restricted to the late 19th and early 20th century. Nino 3.4 SST anomalies are
taken from NOAA ERSSTv5 (Huang et al. 2017). These have been interpolated from sparse observational data and extend
back until 1870. The annual El Niño Index *Nino3.4* (Wilson et al., 2010), which was reconstructed using using data from the
central Pacific (corals), the TexMex region of the USA (tree rings) and other regions in the Tropics (corals and an ice core),
was used as a time series of past ENSO events that extends beyond the instrumental period, until 1607. Evidence for the

occurrence and magnitude of historical ENSO events have been compiled in Quinn (1993) extending back until 1500. As this
reconstruction is based on historical observations of various aspects of ENSO, it should be relatively independent from
statistical biases. Additionally, Brönnimann et al. (2006) obtained a balanced view of ENSO events (both El Niño and La Niña
events) by combining several reconstructed ENSO indices, climate field reconstructions and early instrumental data. Their
reconstruction period extends back to 1500 (La Niña events) and 1511 (El Niño events), respectively.

## 3 Methods and materials

### 3.1 Coral collection and preparation

For this study, three sub-fossil coral samples were collected in February 2010, from boulder beaches and derelict buildings of
former settlements at Chagos (Fig. S1). Samples E3 (1870-1909) and E5 (1675-1716) were taken from Eagle Island (S
6°11.39'; E 71°19.58'), an island on the western rim of the Great Chagos Bank (Fig. 1). Sample B8 (1836-1867) was taken

from the lagoon-facing site of Boddam Island (S 5°21.56'; E 72°12.34') in the southwestern part of the Salomon atoll. The
samples were cross-sectioned into 0.7-1.0 cm thick slabs and X-rayed.
Core GIM, a modern core described in previous publications (Pfeiffer et a., 2004; Timm et al., 2005; Pfeiffer et al., 2009;
Pfeiffer et al., 2017) was drilled underwater in 1995 in the lagoon of Peros Banhos, located in the northwest of Chagos. The
monthly coral Sr/Ca record of GIM extends from 1880-1995. Analytical procedures have been described in Pfeiffer et al.



(2009). The core top (1950-1995) was shown to record SST variability at Chagos on a grid-SST scale (Pfeiffer et al., 2009).
The entire record was included in a composite reconstruction of large-scale western Indian Ocean SST (Pfeiffer et al., 2017).
In this study, we use this core to estimate the magnitude of modern ENSO events.

From the slabs of the sub-fossil corals, powder samples were drilled at 1 mm increments using a micro-milling machine (type
PROXXON FF 500 CNC). This depth resolution can be translated to monthly temporal resolution with average growth rates
being 12 mm/yr. The subsampling paths were always set along the optimal growth axis that was determined based on x-ray
images (Fig. S2).

### 3.2 Coral Sr/Ca analysis

Sr/Ca ratio measurements were performed at Kiel University using a Spectro Ciros CCD SOP inductively coupled plasma
optical emission spectrometer (ICP-OES). Elemental emission signals were simultaneously collected and subsequently
processed following a combination of techniques described by Schrag (1999) and de Villiers et al. (2002). Between 0.13 and
0.65 mg of coral powder was dissolved in 1.00 mL 0.2 M $HNO_3$. Prior to analysis, the solution was diluted with 0.2 M HNO3
to a final concentration of approximately 8ppm Calcium. Strontium and Calcium intensity lines used are 421 nm and 317 nm,
respectively. The intensities of Strontium and Calcium were converted into Sr/Ca ratios in mmol/mol. An in-house coral
reference standard (Mayotte) was measured after every six samples and was used for drift-correction of the measured Sr/Ca
ratios. Average analytical precision of Sr/Ca measurements as estimated from sample replicates was typically around 0.08 %
RSD, translating into a temperature of less than 0.2°C.

### 3.3 Chronology

Each sub-fossil coral sample was dated by U-Th in 2016. U-Th isotopic measurements were performed with an MC-ICPMS
(Thermo Electron Neptune) in the High-Precision Mass Spectrometry and Environment Change Laboratory (HISPEC) of the
Department of Geosciences, National Taiwan University (NTU), following techniques described in Shen et al. (2012). U-Th
isotopic compositions and concentrations are listed in Table 1.

The chronology of the samples was developed based on seasonal cycles of coral Sr/Ca and by analyzing the density bands
visible on x-ray images (Fig. S2). We assigned the highest Sr/Ca value to the SST minimum of each year and interpolated
linearly between these anchor points to obtain a time series with equidistant time steps.
Sample E5 covers the period from 1675 to 1716, herein further referred to as E5 (1675-1716). Sample B8 covers the period
from 1836 to 1867, E3 from 1870 to 1909, both referred to as B8 (1836-1867) and E3 (1870-1909), respectively. The
uncertainties of the age models are approximately ±1.9 years (E5), ±2.2 years (B8) and ±2.4 years (E3). All age models were
verified by a second, independently measured U-Th age of each sample, measured in 2017 in the HISPEC laboratory of the
Department of Geosciences, NTU, following techniques described in Shen et al. (2012). These age determinations are
consistent with our Sr/Ca chronologies.



### 3.4 Diagenesis screening

A combination of X-ray diffraction (XRD), optical and scanning electron microscopy (SEM) was used to investigate potential diagenetic alteration in the sub-fossil coral samples from Chagos that may affect the Sr/Ca values (Figs. S3, S4, and S5). Representative samples for thin-section, scanning electron microscopy (SEM) and X-ray diffraction (XRD) analysis were

selected from all corals based on the X-ray images. The 2-D-XRD system Bruker D8 ADVANCE GADDS at the Rheinisch-Westfaelische Technische Hochschule (RWTH) Aachen was used for non-destructive XRD point-measurements directly on thin-section blocks with a calcite detection limit of ∼ 0.2 % (Smodej et al., 2015). For each coral sample diagenetic modifications were analyzed using one thin-section, one sample for SEM, one 2D-XRD measurement and one powder-XRD measurement.

### 3.5 Statistics

Composite were generated calculating the mean of positive and negative anomaly events taken from centered monthly coral SST anomaly records. By centering the coral records to their mean and focusing on interannual variability, we eliminate the largest uncertainty of single-core Sr/Ca records, as it known from Sayani et al. (2019) that generating a composite Sr/Ca record using five coral sample records instead of one coral record offers a ca. 2.5 times smaller error in mean SST reconstructions.

Power spectra analysis was performed twice using the open source software *PAST* (Hammer et al., 2001). One run was performed with the time series before detrending them, one run after detrending. All time series were detrended using the softwares *breakfit* (Mudelsee, 2009) or *rampfit* (Mudelsee, 2000), respectively (Fig. S6).

Singular spectrum analysis (SSA) (Vautard & Ghil, 1989) and wavelet coherence plots were generated using the *MATLAB* software toolboxes.

T-test were conducted using the free web application *T-Test Calculator* (GraphPad QuickCalcs, 2019).

As the significance of the means calculated for the composite records depends on the numbers of events, standard errors (SE) were used and calculated as follows:

$$SE = \frac{\text{standard deviation } (\sigma)}{\sqrt{\text{Number of events (n)}}} , \tag{1}$$

### 4 Results and Interpretation

### 4.1 Diagenesis

Only trace amounts of diagenetic phases were detected in the sub-fossil coral samples, which show a good to excellent preservation according to the criteria defined in Cobb et al. (2013). Isolated scalenohedral calcite cement crystals were observed in the thin-section of E5 (1675-1716) (Fig. S3 a-d). However, XRD results and SEM analysis confirm that the calcite abundance is below the detection limit of XRD (0.2%) in this sample (Fig. S3 e-f). B8 (1836-1867) shows trace amounts of

patchily distributed, thin aragonite cements (Fig. S4 a-f). E3 (1870-1909) is devoid of diagenetic phases (Fig. S4 a-f), but in





some areas of the thin-section dissolution of centers of calcification can be seen (Fig. S5 c-d). Slight dissolution and microborings are also visible under SEM (Fig. S5 f). However, microborings are always open and therefore will not influence the geochemistry.

### 4.2 Sr/Ca measurements

Table 2 gives an overview of the Sr/Ca ratios of each sub-fossil coral core and statistical key figures of the records. All coral Sr/Ca records were centered, i.e. normalized with respect to their mean values (Pfeiffer et al., 2009) and translated into SST using a temperature dependence of -0.06 mmol/mol per 1°C for Porites corals at Chagos (Leupold et al., 2019; Pfeiffer et al., 2009). The values are shown in Figure 4.

#### 4.2.1 17-18th century

A total of 472 subsamples from E5 (1675-1716) was measured for Sr/Ca. The average Sr/Ca value is 8.96 ± 0.07 mmol/mol. The range of all Sr/Ca values over the 41-year sample span is 0.41 mmol/mol, between a minimum of 8.73 mmol/mol and a maximum of 9.14 mmol/mol.

#### 4.2.2 19-20th century

From B8 (1836-1867), Sr/Ca of 375 subsamples was measured. The average value is 9.02 ± 0.07 mmol/mol over a range of 200   0.506 mmol/mol. The maximum Sr/Ca value for the 31-year sample span is 9.36 mmol/mol, the minimum Sr/Ca value is 8.85 mmol/mol.

For E3 (1870-1909), Sr/Ca measurements were conducted on 415 subsamples. The average Sr/Ca value is 8.95 ± 0.06 mmol/mol for the 39-year sample span, over a range of 0.376 mmol/mol from a minimum value of 8.79 mmol/mol to a maximum of 9.17 mmol/mol.

**4.3 Seasonal cycle**

The mean annual cycles of all sub-fossil coral SST records are not significantly different as indicated by p-values around 1 in the t-tests (Table 3). The seasonal amplitudes in coral SST [°C] are slightly higher in E5 (1675-1716) (1.99°C) compared to B8 (1836-1867) (1.81°C) and E3 (1870-1909) (1.71°C). A shift of mean maximum temperatures from February (E5 and B8) to April (E3) can be observed (Fig. 4). Seasonal amplitudes explain 26-32% of the coral-SST variance (see supplementary 210  material and Fig. S7).

### 4.4 ENSO Interannual SST variability

The modern and the sub-fossil coral SST records were compared with the annually resolved El Niño index *Niño3.4* that extends back until 1607 (Wilson et al., 2010) and the monthly resolved *Niño3.4* index based on NOAA ERSSTv5 (Huang et al., 2017;





only used for power spectrum analysis). All coral records show positive and negative SST anomalies, which occur in years

where ENSO (positive and negative) events have been reported (Fig. 5). For a better comparison of the coral SST records' and the *Niño3.4* index' frequencies, power spectra of detrended time series were computed (Fig. 6). All coral SST records show the typical ENSO periodicity between 3 and 8 years (Fig. 6a-d). Those periodicities can also be found in the power spectra of the Nino3.4 indexes (Fig. 6e & f). Even after detrending, the power spectrum of the GIM coral SST record still shows the highest power at low-frequencies, which translates to a period of 21-22 years. This cannot be explained by ENSO activities,

but may be related to tropical Pacific forcing (Pfeiffer et al., 2009). The power spectrum analysis results were confirmed by both singular spectrum analysis (SSA; Figs. S8-S10) and Wavelet coherence analysis (Fig. S11) results (see supplementary material).

All coral records show variations in the frequency of ENSO events (Figs. 7-9 and Tables 4-6). Our results show that, compared to the 17-18th century, ENSO events are more frequently recorded in coral records of the central Indian Ocean in recent

periods: According to the AVHRR satellite data and coral records, an El Niño event occurs on average every 4 years between 1981 and 2017 (AVHRR) or every 5 years between 1965 and 1995 (coral record), respectively (see Figs. 7 & 9 and Tables 4-6). This is supported by the events listed in Quinn (1993). The average recurrence interval for the period that covers the coral E5 (1675-1716) record is 5.1 yrs, whereas it is 3.6 yrs between 1965 and 1995. The same holds for the negative SST anomaly events (La Niña and non-La Niña events): based on the AVHRR satellite data and the coral records, negative anomaly events

occurred more frequently between 1981 and 2017 (every 2.6 years; AVHRR) and between 1965 and 1995 (every 6 years in the coral record or 5 yrs in Brönnimann et al., 2006), respectively, than during the 17-18th century. For the 17-18th century, the recurrence interval for negative SST anomalies is only 6.8 years (coral record) or 10.3 yrs (Brönnimann et al., 2006) (see Figs. 8 & 9 and Tables 4 & 5).

## 4.5 ENSO composites

Composites of monthly coral SST anomalies were produced for ENSO (positive and negative) events to assess their magnitudes. Each composite was produced using coral records of several individual ENSO events. An overview of the events used for generating each composite can be found in Table 4 and Table 5. An overview of all events found in the coral Sr/Ca records and of ENSO events of the corresponding time periods listed in Quinn (1993) and Brönnimann et al. (2006) is given in Table 6. Positive SST anomalies in the coral records were interpreted as positive ENSO events when the year of occurrence

was listed as one with large-scale ENSO event in Quinn (1993) and Brönnimann et al. (2006) within the error of each coral age model and when the anomaly exceeds 1.5 standard deviations of the mean of each coral record (Fig. S6). In addition to the strong La Niña events listed in Brönnimann et al. (2006), we added negative SST anomalies occurring in years after the El Niño years to the composite.

The composite record for El Niño events comprises 35 events, and 31 events for the La Niña composite (Table 4). To

investigate changes in the magnitude of ENSO anomalies over time, composites for the time periods 17-18th century and 19-20th century, respectively, were generated. For the 17-18th century, six events (five events) were used for the El Niño (La



Niña) composite. For the composite for the 19-20th century, we included events taken from the GIM record. For the 19-20th century, 29 events (26 events) were used for the El Niño (La Niña) composite. The 19-20th century composites, in turn, were split into three sub-periods: 1830-1929 (18 El Niño events, 16 La Niña events), 1930-1964 (five El Niño events, five La Niña events; Table 5) and 1965-1995 (six El Niño events, five La Niña events). These sub-periods were chosen because ENSO activity was reduced between 1930 and 1965 compared to before 1930 and after 1965 (e.g. Cole et al., 1993).

Observations indicate that some upwelling events in the central Indian Ocean are not forced by large-scale ENSO or IOD variability but associated with cyclonic wind stress curls in the southern tropical Indian Ocean (Dilmahamod et al., 2016; Hermes & Reason, 2009). Such an upwelling event occurred in August 2002 and was found in both the coral and satellite SST records at Chagos (see Leupold et al., 2019). To investigate the effect of these negative anomaly events on the La Niña composites, the 19-20th century composites were split up into composites of La Niña events and other negative anomaly events, which are not related to La Niña. La Niña and negative anomalies other than La Niña events were selected based on the months they occurred in, i.e. November-May (La Niña), June-September (Non-La Niña).

As such events are also observed recently, we compared modern (1981-2018) satellite SST composites for El Niño events (nine events), La Niña events (10 events) and negative anomalies other than La Niña events (four events) with our coral SST composites. We used the AVHRR satellite SST (Casey et al., 2010) averaged over entire Chagos (4-8° S; 70-74° W). All SST anomalies were generated by subtracting the long-term monthly mean from each monthly mean SST value and converting coral Sr/Ca into SST using a temperature dependence of -0.06 mmol/mol per 1°C (see Leupold et al., 2019).

### 4.5.1 Positive anomalies in coral and satellite SST composites

Coral SST proxy of the central Indian Ocean record similar, but higher anomalies during El Niño events compared to the satellite composites (Fig. 7), which may reflect the greater sensitivity of the corals to reef-scale temperatures (Leupold et al., 2019) or the different time periods covered by these records (only two El Niño events in the AVHRR record overlap with the coral data). The coral composite records of the 17-18th century show higher anomalies than the coral composites of the 19-20th century (Fig. 7).

All positive SST anomalies identified as El Niño events in the coral records show on average a maximum value of $1.5 \pm 0.1$°C (Fig. 7). The average maximum temperature anomaly value of El Niño events during the 17-18th century were $2.2 \pm 0.2$°C, higher than and significantly different ($p \ll 0.01$) from the average maximum El Niño temperature anomaly during the 19-20th century ($1.3 \pm 0.1$°C). The average maximum temperature of El Niño events picked from the AVHRR satellite SST (covering the period from 1981 to 2018) of $0.8 \pm 0.1$°C is also lower than and significantly different ($p \ll 0.01$) from those in the 19-20th century. This suggests a greater impact of El Niño events on Indian Ocean SST during the 17-18th century compared to the 19-20th century and during the last decades.





### 4.5.2 Negative anomalies in coral and satellite SST composites

No statistically significant differences were found between negative anomalies in coral SST in the central Indian Ocean during the 17-18th century and the 19-20th century and between La Niña and non-La Niña events (Fig. 8).

All negative SST anomalies identified as La Niña and non-La Niña events in the coral records show a minimum temperature anomaly of -1.6 ± 0.1°C on average (Fig. 8). The average minimum temperature anomalies of La Niña and non-La Niña events during the 17-18th century were slightly less extreme (-1.5 ± 0.3°C), but not significantly different (p = 0.73) from the average minimum temperature anomalies of the 19-20th century (-1.6 ± 0.2°C).

Separating the composites into La Niña and non-La Niña events shows that La Niña events in the coral records are slightly

more negative, but not statistically different from non-La Niña events (p = 0.60). The same is observed in the AVHRR satellite SST anomaly composites, where average La Niña minimum temperature anomalies are -0.8 ± 0.1°C and non-La Niña anomalies are -0.6 ± 0.1°C (p = 0.17).

On average, the minimum temperature anomaly is -1.6 ± 0.1°C for all La Niña events and -1.5 ± 0.4°C for La Niña events during the 19-20th century of the coral SST records (p = 0.75).

### 4.5.3 Interannual SST anomalies during the 19th and 20th century

Dividing the 19-20th century into three sub-periods (1830-1929; 1930-1964; 1965-1995) and compiling SST anomaly composites allows us to assess changes the magnitude of ENSO-driven warm and cold anomalies over time (Fig. 9). The El Nino composites do not show any systematic changes during the 19-20[th] century in the Indian Ocean For the period between 1830 and 1929, the average maximum temperature anomaly is 1.4 ± 0.1°C, while between 1930 and 1964 the average

maximum temperature anomaly of 1.2 ± 0.1°C is slightly less extreme than the previous period, but not significantly different (p = 0.5). For the last period of the 20th century, 1965 to 1995, the average maximum temperature anomaly is again to 1.4 ± 0.1°C (Fig. 9). The magnitude of cooling during La Niña and non-La Niña events tend to reduce from 1830-1929 to 1965-1995 (Fig. 9).

For the period between 1830 and 1929, the average minimum temperature anomaly is -1.9 ± 0.2°C. Between 1930 and 1964

the average minimum temperature anomaly increases by 0.58°C to -1.3 ± 0.1°C, and for 1965 to 1995, the average minimum temperature anomaly is -1.1 ± 0.1°C. However, for both El Nino and La Niña events, the differences between the means of the first period (1830-1929) and the last period (1965-1995) are not statistically significant (p = 0.93; p = 0.07, respectively).

### 5 Discussion

During the 17-18th and 19-20th century, all coral records show typical ENSO periodicities in their SSA, power spectra and/or

wavelet analysis. We therefore can say that the ENSO teleconnection to SST in the central Indian Ocean was stable back until the Maunder Minimum, consistent with previous studies that showed a stable ENSO-SST teleconnection over the late 19th



and early 20th century (Pfeiffer and Dullo, 2006). Overall, predominantly strong ENSO events are recorded by the coral records from Chagos, as indicated in the list of events presented in (Brönnimann et al, 2006) (Table 6).

The coral record dated to the 17-18th century covers a period of long-term cooling and succeeds periods of initially reduced
ENSO activity between the early 1500s and early 1600s (Hereid et al., 2013), followed by a phase of protracted ENSO events in the 1620s (Grove, 2018). Between the early 15th to middle 17th century, average amplitudes of El Niño events appear to be similar to modern coral records in the West Pacific (Hereid et al., 2013). Furthermore, it is described that ENSO events in the first half of the 17th century resulted in the failure of the India monsoon and thus, in droughts and famines in South India (Grove, 2018). Our 17-18th century coral record with its higher amplitudes of El Niño events but generally fewer ENSO events
(relative to modern satellite and coral records) reflects both the observations summarized in Grove (2018) and the reduced ENSO activity as found in Hereid et al. (2013). This suggests a shift towards an increased ENSO activity around the first half of the 17th century back to a reduced ENSO activity, but with higher amplitude (El Niño) events compared to modern ones.

Furthermore, our results show that El Niño events resulted in stronger SST anomalies in central Indian Ocean corals in the 17-18th century, i.e. during a cooler mean climate, than they did in the 19th and 20th century. This result is consistent with Pfeiffer
et al., (2017), who found larger amplitude El Nino events in the late 19th century, when mean SSTs in the tropical Indian Ocean were cooler. It is also consistent with Zinke et al. (2004) who found highest $\delta^{18}O$ amplitude variations in the interannual ENSO band between 1645–1715 in a coral from Ifaty, Madagascar. Comparing both periods, the La Niña and non-La Niña events show no significant changes suggesting a stable negative SST anomaly pattern in the Indian Ocean.

Overall, the magnitudes of El Niño and La Niña events recorded in the Chagos coral records during the past century are
comparable (Fig. 9). This suggests the ENSO teleconnection in the tropical Indian Ocean remained symmetric. Only in times of cooler mean climates, the corals seem to indicate higher amplitude ENSO-induced warm anomalies in the tropical Indian Ocean, although these differences are not statistically significant. Hence, our results do not support the notion that an asymmetric ENSO teleconnection with strong warming during El Niño years drives the recent warming of the tropical Indian Ocean as suggested by Roxy et al. (2014). The modern coral records from the western Indian Ocean all show a steady warming
during the 20th century, and this warming continuous in the time interval of reduced ENSO activity between 1930 and 1965 (e.g. Charles et al., 1997; Pfeiffer and Dullo, 2006; Abram et al., 2016). This suggests that neither the magnitude, nor the frequency of past ENSO events explains the centennial-scale warming of the Indian Ocean.

The coral records from Chagos also record upwelling events in boreal summer, which are independent of ENSO, poorly represented in satellite data of SST (see Leupold et al., 2019), and which may result in the failure of the Indian monsoon. Such
an upwelling event occurred for example in 2002 and lead to a drought over the Indian subcontinent (Jayakumar & Gnanaseelan, 2012; Krishnan et al., 2006). At present, little is known about the frequency or magnitudes of these events in past decades or centuries. Coral proxy data from Chagos thus allow us to better understand these events non-La Nina upwelling events.

In contrast to the stable teleconnection between ENSO and SST in the central Indian Ocean, the ENSO-precipitation
teleconnection was shown to be non-stationary (Timm et al., 2005). The impact of ENSO on rainfall in the central Indian



Ocean depends on mean SSTs, and these surpassed a critical threshold for atmospheric convection in the mid-1970s, strengthening the El Nino signal in rainfall. However, our study does not indicate an increase in the magnitude of El Nino-related SST anomalies following this shift compared to earlier time periods of strong ENSO activity.

In summary, this study confirms that the ENSO-SST teleconnection between the Pacific and Indian Ocean is stationary over
the 19th/20th century and back into the Maunder Minimum. We have shown that it is possible to reconstruct interannual SST variations in the tropical Indian Ocean. This is important because so far there exist no reliable high-resolution SST reconstructions in the Indian Ocean covering the periods we studied. SST reconstruction studies of the Pacific Ocean also show ENSO and decadal-scale variability covering the periods from 1998 back to 1886 (Cobb et al., 2001) and 928-961, 1149-1220, 1317-1464 and 1635-1703 (Cobb et al., 2003). Cobb et al. (2003) spliced three overlapping coral records of the 14-15th
century and five coral records of the 17-18th century together. We have shown that this approach would be applicable in the tropical Indian Ocean using sub-fossil corals from boulder beaches and historical buildings. This is important because recent studies have shown that the tropical Indian Ocean plays a pivotal role in 20[th] century global temperature rise (Funk et al., 2008; Roxy et al., 2014; Pfeiffer et al., 2017).

## 6 Conclusions

We have shown that the ENSO-SST relationship in the central Indian Ocean was stationary since the 17th century. All four coral records showed typical ENSO periodicities, but variations in the frequency of ENSO events and in intensity. Whereas both El Niño and La Niña events occur more often in modern periods (inferred from satellite and coral records) than in the 17-18th century, El Niño events were more extreme in the 17-18th century relative to ENSO signals in modern coral (1965-1995) and satellite (1981-2018) records. El Niño events cause average positive anomalies of $2.2 \pm 0.2°C$ during the 17-18th century
and $1.3 \pm 0.1°C$ during the 19-20th century, while La Niña events cause average negative anomalies of $-1.5 \pm 0.3°C$ during the 17-18th century and $-1.6 \pm 0.2°C$ during the 19-20th century in the central Indian Ocean. However, not all cooling events are related to La Niña events, but also to processes internal to the Indian Ocean causing negative anomalies of $-1.5 \pm 0.4°C$ during the 19-20th century. The magnitudes of El Niño and La Niña events during the last century are comparable indicating a symmetric ENSO teleconnection. An asymmetric ENSO teleconnection being the cause for the overall warming of the central,
tropical Indian Ocean appears therefore unlikely. However, we suggest compiling composite records of negative and positive SST anomaly events from sub-fossil corals to further test the hypothesis of an asymmetric ENSO teleconnection in the western Indian Ocean.

**Author contribution:** M.L. conceived the study, wrote the paper and produced all figures. M.P., L.R., T.K.W. and D.G.-S.
helped with analyzing and interpreting the data. L.R. assessed the preservation of the coral samples. T.K.W., C.-C.S. and G.-J.B. helped dating the samples and with the development of the age models. M.P. acquired the funding for this project, contributed feedback and helped refine the writing.

**Competing interests:** The authors declare that they have no competing interests.





**Data and materials availability:** All methods needed to evaluate the conclusions in the paper are present in the paper and/or
the supplementary material. The data plotted in all figures will be available to the public over the Paleoclimatology Branch of
NOAA's National Center for Environmental Information (NCEI) (http://www.ncdc.noaa.gov/data-access/paleoclimatology-
data) after the completion of the dissertation of M. Leupold.

**Acknowledgments:** We thank Karen Bremer for laboratory assistance and the Deutsche Forschungsgemeinschaft (DFG) for
funding the projects PF 676/2-1 and PF 676/3-1. Coral U-Th dating was supported by grants from the Science Vanguard
Research Program of the Ministry of Science and Technology, Taiwan, ROC (108-2119-M-002-012), the Higher Education
Sprout Project of the Ministry of Education, Taiwan, ROC (108L901001), and National Taiwan University (109L8926).

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





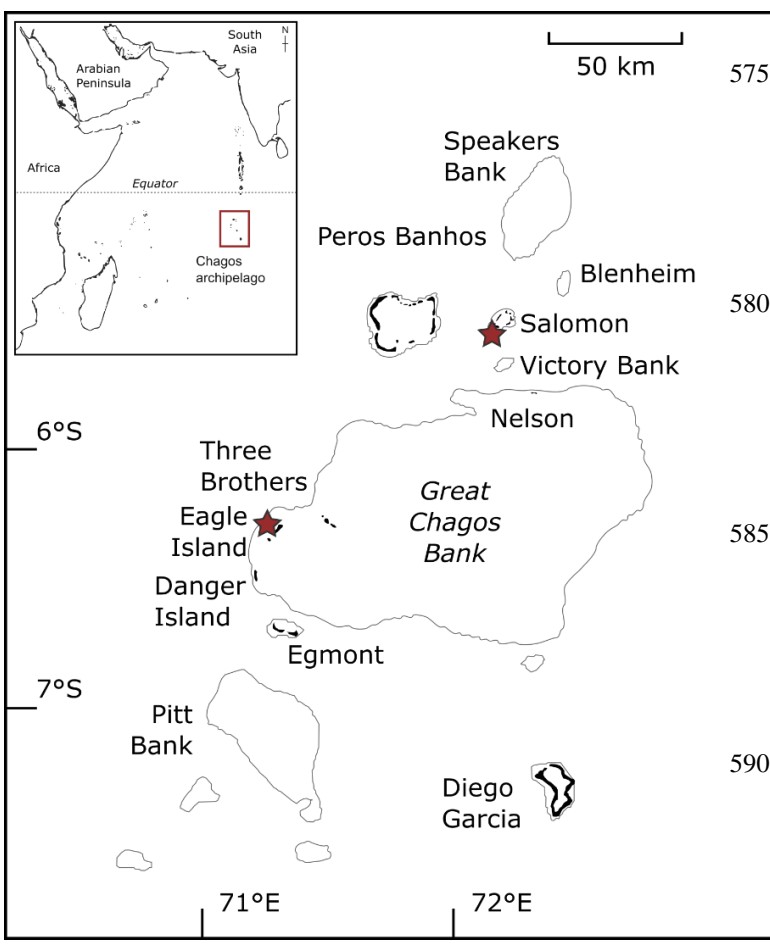

Figure 1: Location of study area and coral sample locations. The Chagos Archipelago is located in the central Indian Ocean, about 550 km south of the Maldives (map upper left). Fossil coral samples were collected on Eagle Island and on Boddam Island (Salomon atoll; red stars).

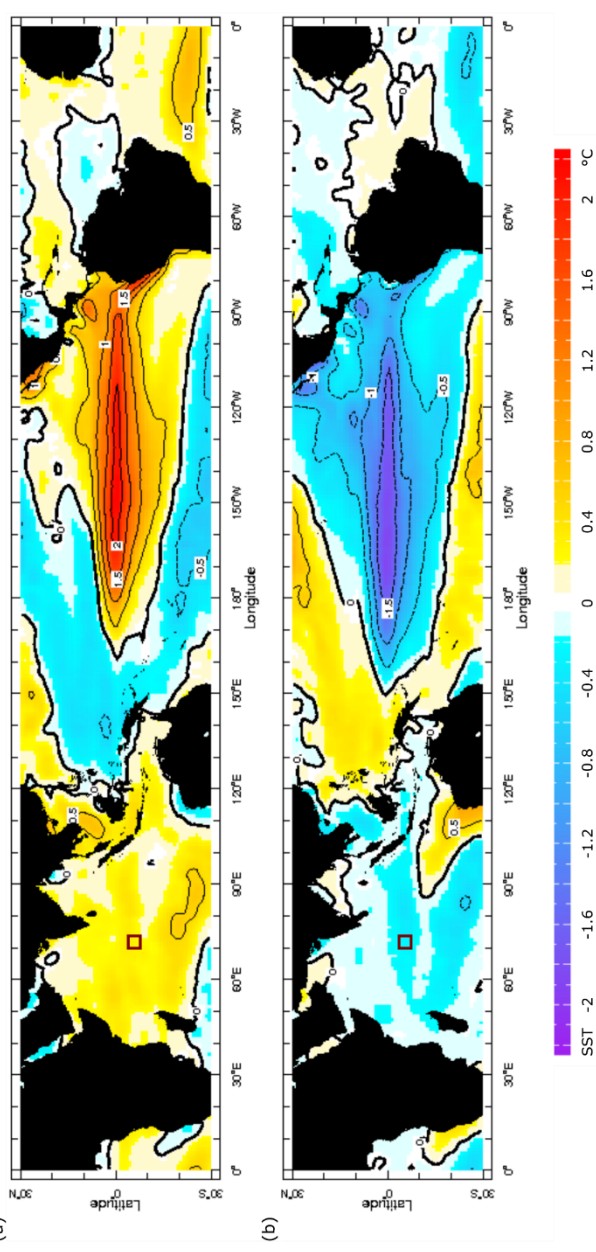


**Figure 2: Composite maps of SST anomalies [°C] in the Indian and Pacific Ocean during ENSO events. (a) El Niño SST anomalies for the period 1982 to 2016 averaged over December to February. (b) same as in (a), but for La Niña events. SST anomaly maps were computed with NOAA 'Reynolds' OI v2 SST (Reynolds et al., 2002) using the free web application Data Views of the IRI Data Library (https://iridl.ldeo.columbia.edu/). Date accessed: 17 September 2018. Red squares indicate the location of the study area.**
**An overview of all events used for each composite map can be found in Table S1 in the supplementary material.**



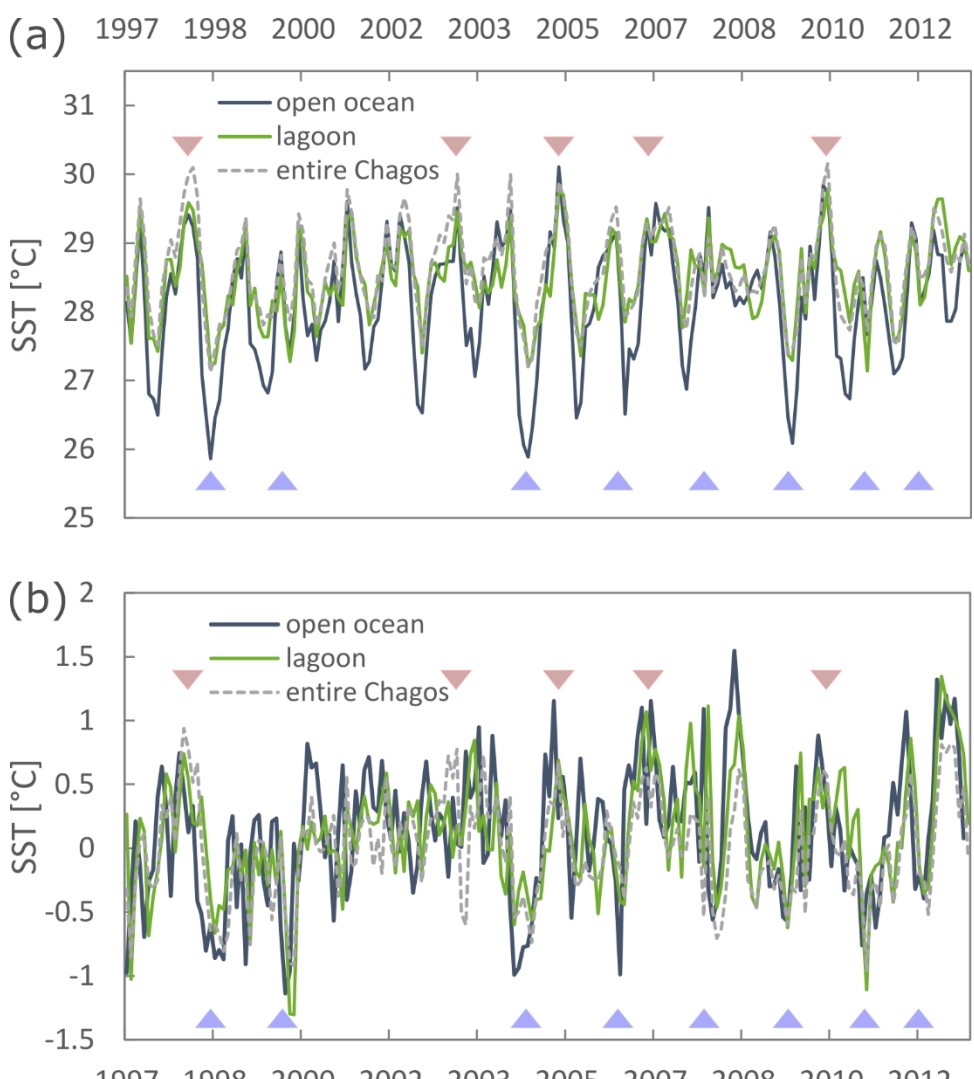

Figure 3: Satellite SST for different settings (lagoon: green; open ocean: blue) and entire Chagos (grey; averaged over 70-74° E; 4-8° S). (a) Monthly satellite SST means and (b) satellite SST anomalies. For the open ocean and lagoon setting we used the high-resolution satellite SST product AVHRR (Casey et al., 2010) and for entire Chagos we used NOAA 'Reynolds' OI v2 SST (Reynolds et al., 2002). Arrows indicate El Niño (red) and La Niña events (blue) based on Brönnimann et al. (2006) and the Oceanic Niño Index ONI (https://www.ggweather.com/enso/oni.htm; Date accessed: 18 October 2018).





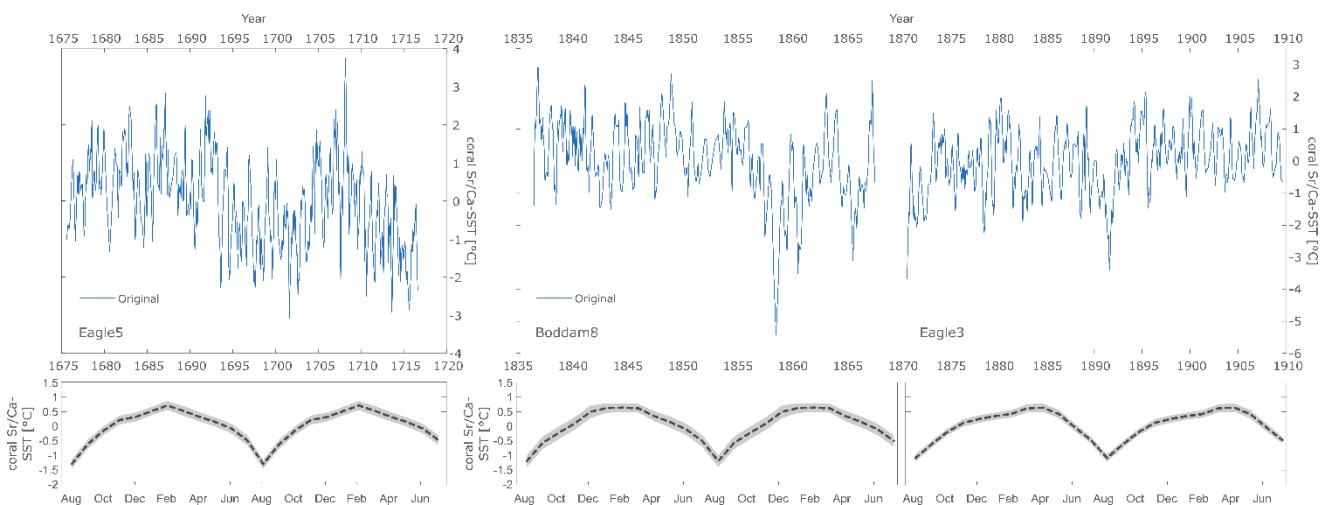

**Figure 4: Monthly Sr/Ca records (blue lines; converted into coral Sr/Ca-SST in °C) of E5 (1675-1716), B8 (1836-1867) and E3 (1870-1909) and mean annual cycles (black lines and corresponding standard errors highlighted in gray, lower plot).**





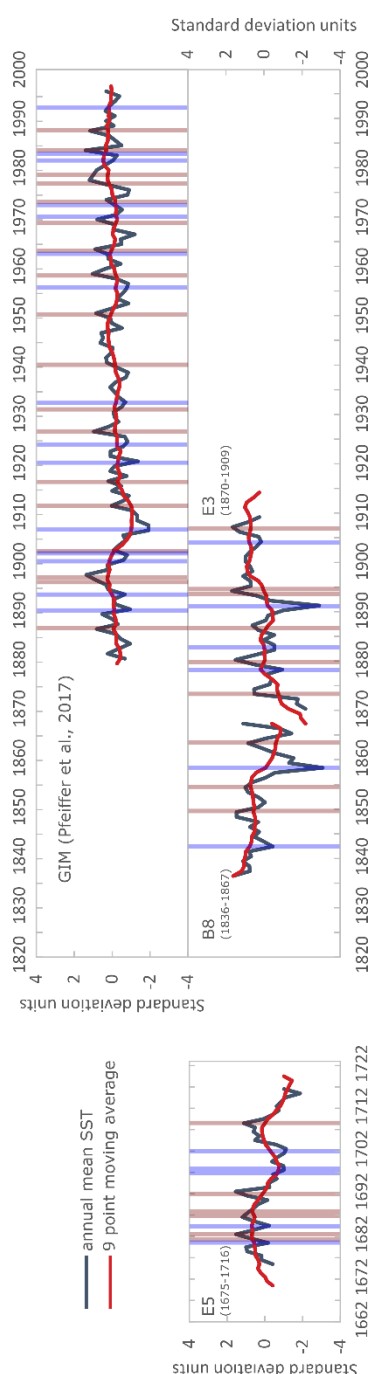

**Figure 5: Annual SST anomalies for Chagos corals (this study and GIM from Pfeiffer et al. 2009, 2017) Red- (El Niño) and blue-(La Niña) shaded boxes indicate years used for the composite records (Figs. 7-9). Red thick lines are 9 point moving averages. See text Sect. 4.5 for how ENSO events were picked.**



Figure 6: Power spectrum analysis plots for detrended coral SST, the annually resolved Niño3.4 index (Wilson et al., 2010) and the monthly resolved Niño3.4 index based on NOAA ERSSTv5 (Huang et al., 2017) time series.





# POSITIVE SST ANOMALIES COMPOSITES

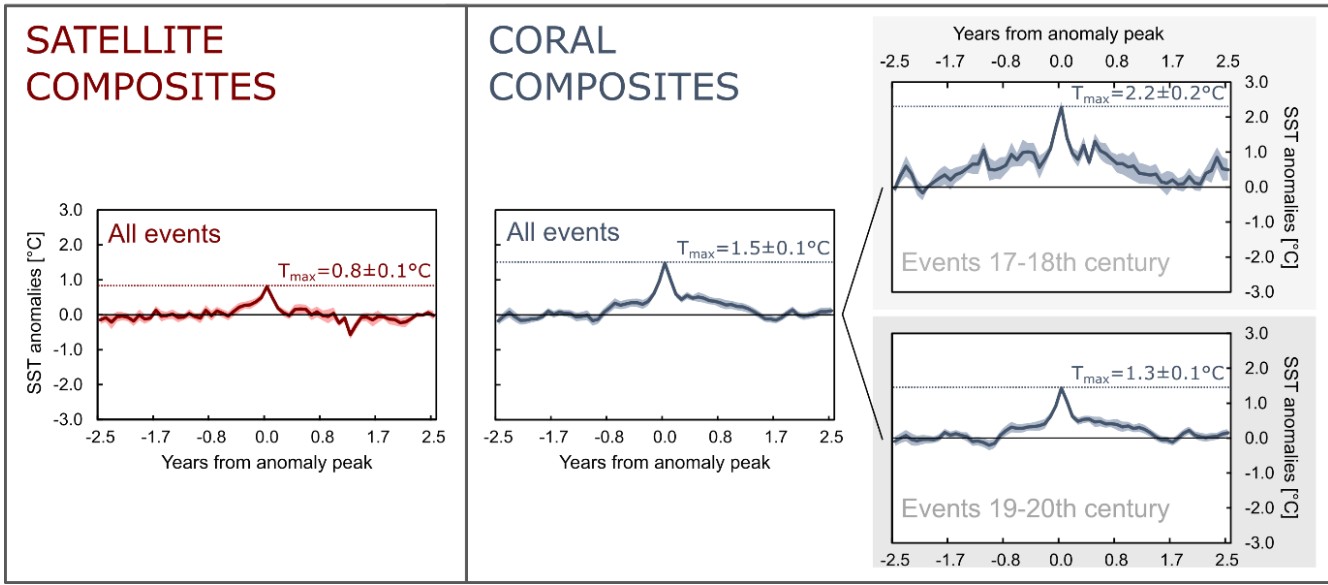

**Figure 7: Positive SST anomalies (El Niño) composite records of AVHRR (left; red) and coral SST (right; blue) records. Separate composites of anomaly events during the 17-18th and 19-20th century were generated from the coral SST records. Shaded areas below and above the curves show the standard error for the mean values of the composite records. Table 4 for an overview of the events that were selected for generating the composites.**






# NEGATIVE SST ANOMALIES COMPOSITES

**Figure 8: Negative SST anomalies (La Niña and non-La Niña) composite records of AVHRR satellite (left) and coral SST (right) records. Additionally, composites of anomaly events separated by 17-18th and 19-20th century events and by La Niña and non-La Niña events were generated. Shaded areas below and above the curves show the standard error for the mean values of the composite records. See Table 4 for an overview of the events that were selected for generating the composites.**





**Figure 9: Positive (El Niño; left) and negative SST anomalies (La Niña and non-La Niña; right) composite records of the 19-20th century coral SST records separated by the time intervals 1830-1929 (upper row), 1930-1964 (middle row) and 1965-1995 (lower row). Shaded areas below and above the curves show the standard error for the mean values of the composites records. See Table 5 for an overview of the events that were selected for generating the composites.**





**Table 1: Overview of Uranium and Thorium isotopic compositions and $^{230}$Th ages and corresponding years for fossil coral samples E5 (1675-1716), B8 (1836-1867) and E3 (1870-1909) measured with MC-ICPMS, Thermo Electron Neptune, at NTU. Location of measurement numbers are indicated on x-ray images in Figure S2. Chemistry was performed on March.11th, 2016 and on July 16, 2017 (Shen et al., 2003), and instrumental analysis on MC-ICP-MS (Shen et al., 2012).**

| Sample ID | Measurement No. | $^{238}$U (ppb[a]) | $^{232}$Th (ppt) | d$^{234}$U (measured[a]) | [$^{230}$Th/$^{238}$U] (activity[c]) | [$^{230}$Th/$^{232}$Th] (ppm[d]) | Age (uncorrected) | Age (corrected[c,e]) | d$^{234}$U$_{initial}$ (corrected[b]) | Corresponding year (BP) |
|---|---|---|---|---|---|---|---|---|---|---|
| E5 (1675-1716) | 1st | 2265,7 ±2,3 | 74,1 ±3,0 | 146,4 ±1,3 | 0,003250 ±0,000019 | 1639 ±66 | 309,5 ±1,9 | 308,8 ±1,9 | 146,6 ±1,3 | 1706 ±1.9 |
| | 2nd | 2293,9 ±2,2 | 16,1 ±1,3 | 145,0 ±1,6 | 0,003594 ±0,000018 | 8458 ±675 | 342,8 ±1,8 | 342,6 ±1,8 | 145,2 ±1,6 | 1674 ±1.8 |
| B8 (1836-1867) | 1st | 2212,7 ±2,5 | 37,1 ±4,1 | 144,1 ±1,5 | 0,001872 ±0,000023 | 1840 ±203 | 178,5 ±2,2 | 178,1 ±2,2 | 144,2 ±1,5 | 1838 ±2.2 |
| | 2nd | 2386,1 ±2,1 | 515,4 ±1,4 | 146,2 ±1,3 | 0,001650 ±0,000029 | 126 ±2 | 157,1 ±2,8 | 152,1 ±3,7 | 146,2 ±1,3 | 1865 ±3.7 |
| E3 (1870-1909) | 1st | 2551,9 ±2,5 | 56,7 ±3,9 | 145,4 ±1,3 | 0,001194 ±0,000025 | 886 ±64 | 113,8 ±2,4 | 113,3 ±2,4 | 145,4 ±1,3 | 1903 ±2.4 |
| | 2nd | 2694 ±2,8 | 643 ±2 | 144,7 ±1,7 | 0,0015 ±0,00002 | 106 ±1 | 146 ±2 | 141 ±3,2 | 145 ±1,7 | 1876 ±3.2 |

Analytical errors are 2σ of the mean.
[a][$^{238}$U] = [$^{235}$U] x 137.818 (±0.65‰) (Hiess et al., 2012); δ$^{234}$U = ([$^{234}$U/$^{238}$U]$_{activity}$ - 1) x 1000.
[b]δ$^{234}$Uinitial corrected was calculated based on $^{230}$Th age (T), i.e., δ$^{234}$U$_{initial}$ = δ$^{234}$U$_{measured}$ X e$^{λ 234*T}$, and $T$ is corrected age.
[c][$^{230}$Th/$^{238}$U]$_{activity}$ = 1 - e$^{-λ230T}$ + (δ$^{234}$U$_{measured}$/1000)[ λ$_{230}$/( λ$_{230}$ - λ$_{234}$)]( λ - e$^{-( λ230 - λ234) T}$), where $T$ is the age.
[d]The degree of detrital $^{230}$Th contamination is indicated by the [$^{230}$Th/$^{232}$Th] atomic ratio instead of the activity ratio.
[e]Age corrections, relative to chemistry date, for samples were calculated using an estimated atomic $^{230}$Th/$^{232}$Th ratio of 4 ± 2 ppm.
Those are the values for a material at secular equilibrium, with the crustal $^{232}$Th/$^{238}$U value of 3.8. The errors are arbitrarily assumed to be 50%.

**Table 2: Statistical overview for raw Sr/Ca data.**

| Sample | Amount subsamples | Sr/Ca [mmol/mol] | | | | | | median RSD [%] |
|---|---|---|---|---|---|---|---|---|
| | | Mean | Median | Std dev | Min | Max | Range | |
| E5 (1675-1716) | 472 | 8.96 | 8.96 | 0.07 | 8.73 | 9.14 | 0.410 | 0.076 |
| B8 (1836-1867) | 375 | 9.02 | 9.02 | 0.07 | 8.85 | 9.36 | 0.506 | 0.075 |
| E3 (1870-1909) | 415 | 8.95 | 8.95 | 0.06 | 8.79 | 9.17 | 0.376 | 0.074 |

**Table 3: Statistical overview for mean annual cycle data of the coral Sr/Ca-SST [°C] records.**

| Sample | Max | Min | Amplitude | Mean | SD | P-value of t-test (two-tailed) | | |
|---|---|---|---|---|---|---|---|---|
| | | | | | | E5 (1675-1716) vs. | B8 (1836-1867) vs. | E3 (1870-1909) vs. |
| E5 (1675-1716) | 0.70 | -1.29 | 1.99 | 0.0026 | 0.5459 | | 0.9979 | 0.9991 |
| B8 (1836-1867) | 0.61 | -1.21 | 1.82 | 0.0033 | 0.5450 | 0.9979 | | 0.9969 |



| | | | | | | | | |
|---|---|---|---|---|---|---|---|---|
| E3 (1870-1909) | 0.60 | -1.11 | 1.71 | 0.0024 | 0.5089 | 0.9991 | 0.9969 | |

*Note*. **SD is the standard deviation.**





665

| Composite | | Years with events | Number of events | Records used |
|---|---|---|---|---|
| **Postive SST anomalies** / Coral Composites | all events | 1679, 1682, 1686, 1687, 1691, 1708, 1849, 1853, 1863, 1873, 1879, 1881, 1886 (2x), 1889, 1894, 1895, 1896, 1897, 1902, 1907, 1911, 1916, 1926, 1932, 1940, 1951, 1958, 1963, 1969, 1973, 1977, 1979, 1983, 1987 | 35 | E5 (1675-1716), B8 (1836-1867), E3 (1870-1909), GIM (1880-1995) |
| | 17-18th century | 1679, 1682, 1686, 1687, 1691, 1708 | 6 | E5 (1675-1716) |
| | 19-20th century | 1849, 1853, 1863, 1873, 1879, 1881, 1886 (2x), 1889, 1894, 1895, 1896, 1897, 1902, 1907, 1911, 1916, 1926, 1932, 1940, 1951, 1958, 1963, 1969, 1973, 1977, 1979, 1983, 1987 | 29 | B8 (1836-1867), E3 (1870-1909), GIM (1880-1995) |
| Satellite Composite | all events | 1983, 1987, 1988, 1998, 2003, 2005, 2007, 2015, 2016 | 9 | AVHRR SST (1981-2018) |
| **Negative SST anomalies** / Coral Composites | all events | 1680, 1684, 1697, 1698, 1702, 1846, 1858, 1860, 1865, 1872, 1883, 1890, 1891, 1893, 1895, 1900, 1902, 1903, 1906, 1920, 1924, 1932, 1947, 1952, 1956, 1964, 1970, 1974, 1982, 1984, 1994 | 31 (22 LN, 9 NLN) | E5 (1675-1716), B8 (1836-1867), E3 (1870-1909), GIM (1880-1995) |
| | 17-18th century | 1680, 1684, 1697, 1698, 1702 | 5 | E5 (1675-1716) |
| | 19-20th century | 1846, 1858, 1860, 1865, 1872, 1883, 1890, 1891, 1893, 1895, 1900, 1902, 1903, 1906, 1920, 1924, 1932, 1947, 1952, 1956, 1964, 1970, 1974, 1982, 1984, 1994 | 26 (19 LN, 7 NLN) | B8 (1836-1867), E3 (1870-1909), GIM (1880-1995) |
| Satellite Composite | all events | 1984, 1989, 1989, 1992, 1995, 1996, 1998, 2000, 2004, 2008, 2011, 2012, 2014, 2017 | 14 (10 LN, 4 NLN) | AVHRR SST (1981-2018) |

**Table 4: Positive (El Niño) and negative (La Niña and non-La Niña) SST anomaly events picked for generating coral and satellite composite records shown in Figure 7 and Figure 8.**





| 19-20th century Coral Composite | Period | Years with events | Number of events | Records used |
|---|---|---|---|---|
| Postive SST anomalies | 1830-1929 | 1849, 1853, 1863, 1873, 1879, 1981, 1886 (2x), 1889, 1894, 1895, 1896, 1897, 1902, 1907, 1911, 1916, 1926 | 18 | B8 (1836-1867), E3 (1870-1909), GIM (1880-1995) |
| | 1930-1964 | 1932, 1940, 1951, 1958, 1963 | 5 | GIM (1880-1995) |
| | 1965-1995 | 1969, 1973, 1977, 1979, 1983, 1987 | 6 | GIM (1880-1995) |
| Negative SST anomalies | 1830-1929 | 1846, 1858, 1860, 1865, 1872, 1883, 1890, 1891, 1893, 1895, 1900, 1902, 1903, 1906, 1920, 1924 | 16 | B8 (1836-1867), E3 (1870-1909), GIM (1880-1995) |
| | 1930-1964 | 1932, 1947, 1952, 1956, 1964 | 5 | GIM (1880-1995) |
| | 1965-1995 | 1970, 1974, 1982, 1984, 1994 | 5 | GIM (1880-1995) |

**Table 5: 19-20th century (divided into three periods) positive (El Niño) and negative (La Niña and non-La Niña) SST anomaly events picked for generating coral composite records shown in Figure 9.**

670



| | Positive SST Anomalies | | | | | | Negative SST Anomalies | | | |
| | Events in Records | | Published ENSO events | | | | Events in Records | | Published ENSO events | |
| | | | Quinn (1993) | | Brönnimann et al. (2006) | | | | Brönnimann et al. (2006) | |
| | Years | Numbers of events | Years of very strong (VS), strong (S), medium (M) and weak (W) events | Numbers of events | Years of strong events | Numbers of events | Years | Numbers of events | Years of strong events | Numbers of events |
|---|---|---|---|---|---|---|---|---|---|---|
| E5 (1675-1716) | 1678/79, 1682/83, 1685/86, 1686/87, 1691/92, 1707/08 | 6 | 1681 (S), 1684 (M+), 1687-88 (S+), 1692-93 (S), 1696-97 (M+), 1701 (S+), 1707-09 (M/S), 1715-16 (S) | 8 | 1674, 1675, 1677, 1681, 1682, 1691, 1702 | 7 | 1680, 1683/84, 1697, 1697/98, 1702 | 5 | 1676, 1678, 1698, 1704 | 4 |
| B8 (1836-1867) | 1848/49, 1853/54, 1862/63 | 3 | 1837 (M+), 1844-46 (M/S+), 1850 (M), 1852 (M), 1854 (M), 1857-58 (M), 1860 (M), 1862 (M-), 1864 (S), 1866 (M+), 1867-68 (M+) | 11 | 1833, 1846, 1852, 1856, 1869 | 5 | 1845/46, 1858, 1860/61, 1864/65 | 4 | 1842, 1847, 1863 | 3 |
| E3 (1870-1909) | 1872/73, 1879/80, 1885/86, 1893/94, 1894/95, 1906/07 | 6 | 1871 (S+), 1874 (M), 1877-78 (VS), 1880 (M), 1884 (S+), 1887-89 (M+), 1891 (VS), 1897 (M+), 1899-1900 (S), 1902 (M+), 1904-05 (M-), 1907 (M) | 12 | 1869, 1877, 1878, 1889, 1897, 1900, 1903, 1906, 1912 | 9 | 1872, 1882/83, 1891, 1895/96, 1903/04 | 5 | 1872, 1887, 1890, 1893, 1904, 1910 | 6 |
| GIM (1880-1995)* | 1880/81, 1885/86, 1888/89, 1896/97, 1897, 1902, 1911/12, 1916/17, 1925/26, 1931/32, 1939/40, 1950/51, 1957/58, 1962/63, 1969/70, 1972/73, 1977/78, 1978/79, 1982/83, 1986/87 | 20 | 1880 (M), 1884 (S+), 1887-89 (M-/M+), 1891 (VS), 1897 (M+), 1899-90 (S), 1902 (M+), 1904-05 (M-), 1907 (M), 1910 (M+), 1911-12 (S), 1914-15 (M+), 1917 (S), 1923 (M), 1925-26 (VS), 1930-31 (M), 1932 (S), 1939 (M+), 1940-41 (S), 1943 (M+), 1951 (M-), 1953 (M+), 1957-58 (S), 1965 (M+), 1969 (M-), 1972-73 (S), 1976 (M), 1978-79 (W), 1982-83 (VS), 1987 (M), 1991-92 (M), 1994-95 (M-) | 32 | 1878, 1889, 1897, 1900, 1903, 1906, 1912, 1915, 1919, 1926, 1931, 1940, 1941, 1952, 1958, 1966, 1973, 1977, 1983, 1987, 1992 | 21 | 1889/90, 1893, 1899/00, 1901/02, 1906, 1919/20, 1924, 1931/32, 1947, 1956, 1951/52, 1964, 1970, 1973/74, 1982, 1983/84, 1993/94 | 17 | 1887, 1890, 1893, 1904, 1910, 1917, 1925, 1934, 1943, 1950, 1956, 1968, 1971, 1974, 1976, 1985, 1989 | 17 |
| AVHRR* | 1982/83, 1986/87, 1987/88, 1997/98, 2002/03, 2004/05, 2006/07, 2014/15, 2015/16 | 9 | 1982/83 (VS), 1986/87 (M), 1987/88 (S), 1991/92 (S), 1994/95 (M), 1997/98 (VS), 2002/03 (M), 2004/05 (W), 2006/07 (W), 2009/10 (M), 2014/15 (W), 2015/16 (VS) | 12 | / | / | 1984/85, 1989, 1989, 1992, 1995, 1996/97, 1998/99, 1999/00, 2004, 2007/08, 2010/11, 2011/12, 2014, 2016/17 | 14 | 1983/84 (W), 1984/85 (W), 1988/89 (S), 1992, 1995/96 (M), 1996/97, 1998/99 (S), 1999/00 (S), 2000/01 (W), 2004, 2005/06 (W), 2007/08 (S), 2008/09 (W), 2010/11 (S), 2011/12 (M), 2014, 2016/17 (W), 2017/18 (W) | 18 |

\* Note: Recent events (from 1980 on) were additionally picked using events listed on this website: https://www.ggweather.com/enso/oni.htm (Date accessed: 18 October 2018)

**Table 6: Overview of all events found in the coral Sr/Ca records and of ENSO events of the corresponding time periods listed in publications. Events in coral records were matched with published events in consideration of age model uncertainties of each coral record.**

675