# Peer review of "ENSO and internal sea surface temperature variability in the tropical Indian Ocean since 1675"

_Climate of the Past, 2020_

## Short Comment (SC1) · 6 Mar 2020

'Maunder Minimum' is a name given for a period of reduced sunspots observations [Eddy 1976], and not the name describing a period of past climate or climate change. The term is being incorrectly used in this manuscript.

References: Eddy, J.A., The Maunder Minimum, Science, Vol. 192, Issue 4245, pp. 1189-1202, 1976
* * *

---

## Author Comment (AC1) · 27 Mar 2020

Thank you for your comment. We agree, that the term is used in a misleading/incorrect way. We will adjust it in the text and also change title of the manuscript.

---

## Referee Comment (RC1) · Anonymous Referee #1 · 16 Apr 2020

**General comment**

The manuscript represents a substantial contribution to scientific progress in regard to the link between Pacific ENSO and Indian Ocean ENSO. The conclusion reached in this paper brought substantial new proof concerning the lack of impact of ENSO on the central Indian ocean warming.

The manuscript is on average well written, title clearly reflect the contents of the paper, the abstract does provide a concise and accurate summary of the study. The method used in this paper are thorough, with multiple validation techniques to study both the instrumental data and the coral-derived data.

One modification could be to re organize a bit your result section - you might want to do less sections but more structured sections.

**Specific comments**

- l104: "ENSO Indices" There is in my opinion more recent ENSO time series that you coul have. Why did you chose to use only "old" reconstructions?

- l137: "concentration of approximately 8ppm Calcium". I'm a bit surprised by this value. Usually with 0.6 mg of carbonate powder in 6mL you are getting 50ppm of Calcium. Additionally, Villiers et al. (2002) that you are citing above recommend analyzing between 40 to 60 ppm of Ca.

- l140: Can you, please, indicate why you did not use any CRM such as JCp-1? Can you, please, be more specific and give the mean % of recovery ± SD and the N values

- l149: "We assigned the highest Sr/Ca value to the SST minimum of each year and interpolated linearly between these anchor points to obtain a time series with equidistant time steps" Which software did you use? Why only interpolate between two consecutive high values? Why not between two lows or between one high and one low as it is usually the case?

- l193: I would have liked to see how you determine the annual mean (average between two max Sr/Ca) ? and how did you determine the standard error?

- l218: Can you please explain how and especially why you decided to detrend the record?

- l223: You should not cite Figure 7-9 while talking about ENSO event frequency, those figures are not at all giving information on frequencies.

- l228: I'm confused l226 you indicate that El Nino events occurs every 5 years between 1965 and 1995 and l228 you mention a recurrence time of 3.6 years ...

- l229: Can you please indicate here which threshold you used when considering an anomaly and therefore considering that it is an El Nino or La Nina year?

- l240: Using data from Quinn (1993). I do believe that there is more up to date studies on ENSO events... I would feel more confident in your results if you had compared to multiple studies.

- l241: "error of each coral age model". What values did you use as a bracket for the age model uncertainty?

- l242: "we added negative SST anomalies occurring in years after the El Niño years to the composite." I do not understand - are those considered La Niña-like events?

- l266: "the greater sensitivity of the corals to reef-scale". Can it reflect also issues with the calibration you used to convert Sr/Ca to SST?

- l301-032: "However, … "Something is wrong with this sentence. How could differences between means are not significantly different with one p value of 0.9 and one of 0.07 …. Additionally, does that mean that the decrease in amplitude of the negative anomalies are not statistically significative?

- l322-323: Can you please develop a bit more on this idea.

- l325: "comparable". Comparable to what? You also need to be consistent, you have been using the term "coral composite" and know you are using "Chacos coral"… it is a bit confusing, we are wondering if you are not referring to something else …

- l344: You mean the frequency because the strength of the events are different, there is no change in strength of ENSO events in your study during the 20th century (Figure 9).

**Technical corrections**

You need to change all the reference of the Supplementary figures as they are wrongly numbered in the text
You need to change the numbering of the Supplementary Figures legends.
You need to cite Figure S1 in the text

- l37 : "There are only some studies including Sr/Ca measurements for SST reconstructions (e.g. Pfeiffer et al., 2006), while few studies included Sr/Ca measurements for SST reconstructions (e.g. Pfeiffer et al., 2006)."There is a problem with this sentence. You wrote twice the same idea.

- l39: "Most studies are focusing on either the western or the eastern Indian Ocean (Abram et al., 2003; Watanabe et al., 2019) and/or are sampled at only bimonthly … " Be careful it is not the study that are sampled at bi monthly resolution but the coral.

- l51: "The modern core was included in a composite reconstruction of large-scale SST (Pfeiffer et al., 2017) and the core top (1950-1995) was shown to record SST variability at Chagos on grid-SST scale (Pfeiffer et al., 2009)." I'm a bit confused by sentence. Maybe should explain a bit more what you had in mind while writing it. One core records more global scale SST variability while the other more the local variability? Is that it?

- l53: "41 years of the Maunder" Can you write instead "41 years during the Maunder

- l54: "39 years of the mid-19th to early 20th century (1870-1909) covering 39 years". You wrote twice the same info about 39 years.

- l55: "We identify past warm and cold events in each record and use these events to compile composites to evaluate the symmetry of positive and negative ENSO-driven SST anomaly events in the tropical Indian Ocean." This paragraph seems out of context here.

- l62: "… water exchange with the open ocean is substantial." Do you specify that because your coral core is from inside the lagoon?

- l98: "Averaged over the entire area of the Chagos (70-74° E; 4-8° S), SST is similar…" It would be interesting to add the mean values for both sites.

- l104: "ENSO Indices" You might want to introduce this paragraph as the time series you will compare your records to? Or something in this line.

- l125: "The core top (1950-1995) was shown to record SST variability at Chagos on a grid-SST scale (Pfeiffer et al., 2009). The entire record was included in a composite reconstruction of large-scale western

Indian Ocean SST (Pfeiffer et al., 2017)." Those sentences are similar to the ones l51 that I did not understand. The top core was comapred to grid-SST data and it maches perfectly and then the entire record was used in a coral composite but with which other corals? Can you, please, add more information here.

- l128: "From the slabs of the sub-fossil corals, powder samples were drilled at 1 mm increments using a micro-milling machine (type PROXXON FF 500 CNC). This depth resolution can be translated to monthly temporal resolution with average growth rates being 12 mm/yr. The subsampling paths were always set along the optimal growth axis that was determined based on x-ray images (Fig. S2)." Can you please add some information on the sampling over laps that you had to do when switching sampling paths? How did you determine the temporal resolution, by looking at the density band or by looking at the seasonal cycles in Sr/Ca data? You might want to move this paragraph up right below where you talk about your new coral core samples.

- l138: "The intensities of Strontium and Calcium were converted into Sr/Ca ratios in mmol/mol." Which method did you use to convert the instrument output in intensity to concentration values 1) the calibration given to you by the instrument? or 2) the deVillier et al., 2002 ratio method?

- l165: Statistic Section : I would like to see this section a bit above as you use statistics in above paragraphs

- l166: "Composite were generated calculating…" replace by "Composite were generated by calculating…"

- l175: Can you please indicate in which occasion you use the t-test

- l188: It would be interesting to have at the end of the Diagenesis section  a summary sentence stating that your sampled should all be good for geochemical analysis and that the results should not be impacted by secondary calcification …

- l189: In my opinion you do not need subsections, but instead a big paragraph labelled Sr/Ca data description, where you describe the results core by core

- l192: Porites needs to be in italic

- l196: "The range …" Is that the mean range or the maximum range?

- l206: Can you please describe how you determine the mean annual cycle?

- l207: "The seasonal amplitudes in coral SST [°C] are slightly higher" You should be using parenthesis instead of brackets

- l209: I  do believe you should spend a little more time describing Figure 4.

- l224: "Our results show that,…" I'm guessing that these conclusions derived from Table 4-6: you might want to refer to it as well as indicate some stats about this change of frequency. Maybe the percentage of increased frequency?

- l225: Replace the ":" by "."

- l226: Remove here also the reference to Figure 7-9.

- l241: "referring to Figure S6". Figure S6 correspond to the detcoral Sr/Ca records after detrending. Which Figure are you referring to here?

- l245 – 251: This paragraph should be in the method section.

- 256-258: This sentence has no link with the previous sentence and should be separated from it.

- l259: "we compared". I do not think "compare" is the right word. You do not compare, you use the same technique to discriminates El Nino from La Nina from negative events other than La Nina years, right?

- l263: "All SST anomalies were … of -0.06 mmol/mol per 1°C (see Leupold et al., 2019)." This section looks more like a material and method section. You do not talk at all about what you found.

- l265: "Coral SST proxy". What is that? Is it your so-called ENSO composite?

- l265: "Ocean record similar, but higher anomalies". If it is higher it is not similar. What do you mean by "similar"?

- l269-271: Those two paragraphs talk about the same subject; you should not separate them.

- l274: "from those". You mean from the coral composite, right? It is not clear; you might want to rephrase.

- l288: "On average … (p=0,75)". This sentence is a bit similar to the first sentence of the paragraph, no? You might want to regroup them.

- l293: You forgot the "." at the end of the sentence.

- l298: You need to regroup this sentence with the next paragraph as they discuss the same idea.

- Figure S1 : Can you please add an arrow to actually point at the boulder you sampled? Can you please add a symbol of the lagoon of Peros Banhos site location on your Map

---

## Referee Comment (RC2) · Anonymous Referee #2 · 24 Apr 2020

General comments This paper seeks to address the question whether or not more El Nino than La Nina events occurred in the Central Indian Ocean, that was found by the study of Roxy et al. 2014 for the western Indian Ocean. The Indian Ocean has been warming over the past century and the cause of this warming may be linked to warm El Niño events and/or a lack of La Nina cooling events. To address this question, the authors sampled and produced coral-based Sr/Ca-SST reconstructions with three sub-fossil corals and one modern coral cored alive in 1995. The three sub-fossil cores were dating with two U-Th each with errors of ∼1-2 years that are confirmed by the coral geochemistry. They then assess the coral records using a wide variety of data and spectral analyses methods to assess the records for the presence of ENSO events.

They report similar magnitudes of El Nino and La Nina events in their coral records for the three intervals of the Little ice age they investigated. This work has the potential to provide some insight into central Indian Ocean interannual variability for snapshot internals since 1675 CE during the Little Ice Age period observed in Europe; however, the manuscript presented here should be refined to make their research question and results clearer and more convincing.

Specific comments While this study is addressing an important question and producing valuable coral SST reconstructions for a location with few such records, this reviewer finds they do not address the research question posed for their study, were there more El Nino events than la Nina in a robust manner. They do look at magnitudes of these events but not the "asymmetry" they discuss in the introduction or that as suggested by Roxy et al. 2014. This should be a straight forward analysis to test this question but the authors use a wide variety of software programs and several data analysis methods to try and address this question that leads to confusion and as a whole, misses the point of their analysis. For example, they spend considerable time and present several figures with spectral analysis that look for periodicities/frequencies in their data. Since El Nino and La Nina are opposites phases of the ENSO variability or "periodicity" they are looking for, the spectral analysis tells you nothing about whether or not more El Ninos occurred than La Ninas. Spectral analysis is suggestive of periodicities similar to ENSO but is NOT conclusive evidence, see Hochman et al. 2019 (doi: 10.1175/jamc-d-18-0331.1) and Liu et al 2007 (doi: 10.1175/2007jtecho511.1). A large anomaly with the width of 2-7 years can be manifested as a significant 2-7 year periodicity in a spectrum leading to the misinterpretation of ENSO periodicity (try for yourself, do a FFT spectrum and wavelet spectrum of the volcanic explosivity index and compare). Furthermore, why do breakpoint detrending, removing monthly anomalies, etc. it is not necessary to answer your question. Additionally, using one-tie point per year to build the coral chronology introduces a large amount of uncertainty to your time series, especially in the monthly anomalies that could mask any real signal in time and frequency, see figure 12 of Williams et al. 2014 (http://dx.doi.org/10.1016/j.gca.2014.04.006), and

Table 5 in DeLong et al., 2014 (doi:10.1002/2013PA002524). If you are removing the annual cycle from your data, at least two tie points should be used, four is better otherwise your residuals will have a annual cycle still there that introduces spectral noise. There are a considerable number of other studies that look at ENSO variability to address similar questions. Why "reinvent" the data analysis approach? Just use the methods everyone else uses, band pass filter to remove low frequency variability (> 10 year) and trends and higher frequency annual cycle, see collective work of Kim Cobb's lab, (Cobb 2003, 2013, Sayani 2019, Grothe 2019 doi: 10.1029/2019GL083906; Chen et al., 2018 https://agupubs.onlinelibrary.wiley.com/doi/abs/10.1002/2018GL077619, Nurhati et al., 2011 DOI: 10.1175/2011JCLI3852.1) and McGregor 2010 (www.clim-past.net/6/1/2010/ ) not to mention the excellent work by Hereid et al., 2013 (doi:10.1130/g33510.1 and doi: 10.1029/2012PA002352) and the new study published by Lawman et al. 2020 doi: 10.1029/2019PA003742 where they use ENSO variability via histograms and probability density functions to assess ENSO variability in the past that built upon the work of Emile-Geay et al 2016 where they used probability density to assess ENSO variability in a network of coral and mollusks reconstructions and climate models (DOI: 10.1038/NGEO2608). Furthermore, McGregor et al., used a Cluster Analysis to assess El Nino and La Nina amplitudes in fossil corals (DOI: 10.1038/NGEO1936) and they use wavelets to band pass filter their coral reconstructions in their 2011 paper (doi:10.1016/j.gca.2011.04.017). The PAST software you are using is capable of doing band pass filters. My second concern is the coral Sr/Ca records in the fossil corals that show large cold anomalies (up to 6°C?) in Figure 4. The labels in this figure are hard to read but a 4-6°C anomaly is not expected, even for a La Nina event. The anomaly in BoddamB (1856-1862) spans ∼6 years and would be manifest in a spectral analysis as a 6-7 periodicity. Look at the Wavelet spectrum for this coral, it will show you if this periodicity is center on this anomaly and this would be why you see 6-7 year peak in Figure 6b. Same could be said for Eagle 3 (1890-1894) and the three year peak. Please include the wavelet spectrum from each series in your paper (better than the spectrums you have and more convincing if not driven by

these anomalies). Back to the cold anomalies. Looking at the x-radiographs: B8 from (1856-1862) appears fuzzy, could this be dissolution or suboptimal alignment of the corallite to the slab surface (see DeLong 2013 doi:10.1016/j.palaeo.2012.08.019)? If you were to resample this time interval to the far right of that slab, is that cold anomaly still there? I would guess not. Coral E3 has the anomaly from 1890-1894 and this is over the core break in the x-ray image. Do these two paths overlap and how well do they agree with each other? The second path is very close the edge of the coral, could there be local diagenesis there? If you were to sample the second core piece just below the first path, is that cold anomaly still present? For Core Eagle5, the mean shift occurs as the sampling paths shifts from the top to bottom piece of the coral. If you sample a different path with optimal corallites, is this shift still there? All this shifts may be real but any large anomalies should be replicated to see if local diagenesis or suboptimal sampling produce the anomaly. I will note: if you use band-pass filters for your ENSO data analysis, these shifts are less meaningful, but you should make sure your coral Sr/Ca is reflecting the SST signal and not something introduced by sampling. Please include a figure of your raw coral Sr/Ca data with paths in depth in your supplemental materials. Additionally, mark where the XRD and SEM samples were removed from eh slab. It is possible to get pockets of diagenesis in small areas of the coral away from where you did the XRD, thin section, and SEM samples. See Quinn 2006 doi:10.1029/2005GL024972; Sayani 2011 doi:10.1016/j.gca.2011.08.026, Hendy 2007 doi:10.1029/2007PA001462).

The public comments have already questioned the use of the Maunder Minimum in the title and as a climate interval or temporal marker. The paper makes not connections to solar cycles and ENSO variance in the central Indian Ocean and the coral do not span the entire Maunder Minimum so why mention it in the title? I suggest the use of the Little Ice Age in its place, as the records presented are part of this interval and that term is accepted within the climate and paleoclimate literature.

The authors need to improve their review of coral Sr/Ca reconstructions in the Indian

Ocean. While it is true that there are not many records currently published from the region, there are more than the authors suggest, seven by my count. Line 38-39 if there are few coral Sr/Ca studies, why not list them all to be comprehensive and not just cite the authors own papers. I count 7 studies so is that really a few? Hennekam 2018 doi: 10.1002/2017PA003181 Zinke 2014 doi: 10.1038/ncomms4607 Zinke 2004 doi: 10.1016/j.epsl.2004.09.028 Zinke 2008 doi:10.1029/2008GL035634. The introduction section would also benefit from a more in-depth review of the literature on coral-based SST reconstructions of ENSO, both from the Indian Ocean perspective and also the Pacific Ocean. Lawman et al. (2020) in Paleoceanography and Paleoclimatology, McGregor et al. (2019) in Nature Geosciences, Grothe et al. (2019) in Geophysical Research Letters, and Tangri et al. (2018) in Paleoceanography and Paleoclimatology would all be useful for comparison, and have data available online. These and other ENSO reconstructions can be used for comparisons between basins back to the 1600s.

I question the authors' decision to count all positive SST anomalies in their coral records as El Niño events, despite the fact that they acknowledge the existence of warm IOD events occurring independently of ENSO (Section 2.2 Climate, lines 92-93). If the authors are comparing other ENSO records to this one, why not remove any positive anomaly events that are unconfirmed by other ENSO records as potential IOD events? Or, why not also compare their record with IOD records? Barring the complete removal of IOD-associated events from the record, I think it would be worthwhile for the authors to compare reconstructions with and without the positive SST anomalies that are not confirmed ENSO events to provide a more complete perspective on potential overestimation of El Niño frequency and strength. I also recommend that the authors review recent literature regarding the IOD, including the recently published Abram et al. (2020) Nature article reconstructing the IOD back to the 13th century AD.

In section 2.4, "ENSO Indices", the authors list the indices that they use for comparison with their coral records. However, they do not discuss whether these records are coher-

ent, or how they vary, over time. It also appears that they generated their own Niño3.4 anomaly record, which they call an index. From what I understand, the Niño3.4 index only extends back to 1870, using HadISST, not ERSST. While I applaud the authors for applying their own analysis to the data, it is unclear exactly how they calculated their anomaly record from the ERSST data, and as such they need to describe that process in more detail. Do not call the Wilson ENSO reconstruction Nino3.4, that name has already been taken, just call it Wilson ENSO.

Especially questionable is the application of the Quinn 1993 record (Ortlieb 2000 provides an updated version), which is subjective and based on written records, though I understand the authors are limited in the number of records that they can use due to the limited temporal scope of most ENSO records. I'm particularly confused as to why they did not compare some of their 19th century records to the extended multivariate ENSO index (MEI.ext), which spans 1871-2005 (Wolter and Timlin, 2011), or the more recent series of indices published by Sullivan et al. (2016) that include central, eastern, and mixed-type ENSO events back to 1854? Or any of the other ENSO reconstructions on the NOAA paleoclimate website, there are several to choose from (Cobb 2013, McGregor, 2010, Li 2011, Braganza 2009, Cook 2008, Gergis 2009).

At the very least, a comparison between the two main indices used (earlier than table 6/section 4.5) would greatly strengthen the authors' conclusions and help the reader understand their criteria surrounding the selection of El Niño events from these records for comparison. The authors cite Wilson et al. (2010), which analyzes the coherence between several ENSO reconstructions extending back to the 17th century, but do not address the paper's conclusion that inter-reconstruction coherence breaks down in the 19th century. Thus, using the Wilson et al. (2010) record to identify individual events in the late 17th – early 19th century seems questionable. Labeling this record Niño3.4 was also confusing, making it hard to differentiate between the Wilson record and the ERSST-based anomaly record from the Niño3.4 region.

This paper has a lot of potential, but needs extensive work. I commend the authors for

attempting an in-depth analysis of their data, but encourage them to consider alternative methods for analysis that would be both simpler to accomplish and ultimately more powerful in their application.

Technical corrections

Figure 6 The authors do not standardize their spectra in time, so that it becomes difficult to interpret the individual plots of Figure 6. Most of the plots are based on monthly resolved data with frequency as cycles/month, except for 6e which is based on annually resolved data and cycles/year and is thus shifted in frequency space.

In section 3.1 "Coral collection and preparation" more information about the x-ray system used and the settings applied in the generation of the x-radiographs would be helpful for replication or reproduction by later studies. Are these x-ray positive or negative images? It would also be useful to know how the coral collected from the derelict building arrived there – was it via human activity or storm or tsunami deposited? This is not necessary for publication, but could help guide the location and collection of other specimens.

In section 3.2 "Coral Sr/Ca analysis" was just one standard or known value used in the ICP analysis? Most labs use 2 or 3 (a gravimetric, a coral, and JCP international standard). The Schrag (1999) and de Villiers et al. (2002) methods bracket each sample for drift correction. which is typical for ICP-OES whereas every 5th sample is used for ICP-MS since that instrument does not drift as much. The exact analytical precision(s) $\pm$ 1sigma should be given with # of measurements and error bars of analytical precision on all graphs with coral Sr/Ca. It would also be good to see the raw Sr/Ca values plotted, not just anomalies. It is difficult to gauge the individual records from the anomaly plots alone.

In section 3.3 "Chronology" the authors suggest that they only use the minima of seasonal SST cycles as their chronological tie points, but their chronology would likely be more robust if they used at least 2 ties points (maxima and minima) for time assignment.

In section 3.5 "Statistics", it would be helpful to know which version of PAST (with citation) and MATLAB the authors used. I am confused as to why the authors chose to use the web application T-Test Calculator (web link needs to be given) rather than a t-test function in the other software listed or just use a t-table in a statistics textbook. Also, in general, the authors tend not to list the $\alpha$, n, or other key statistical values for their data throughout the paper (except in some figures). All averages should be report with their standard deviations, and number of values, correlations should have p-value and n, and all errors as either 1 or 2 sigma, which are standard statistical practices.

In section 4.4, "ENSO Interannual SST variability", the authors suggest that all of their coral records show statistically robust typical ENSO periodicities (3-8 years), but fail to address varying levels of statistical robustness. Their earliest composite record (E5, Figure 6a) for example has an ENSO periodicity that is only statistically significant at the $\alpha$=0.1 level, but the authors do not discuss this in the text. Despite detrending before analysis, there is also evidence of roughly annual periodicities in both B8 (Figure 6b) and E3 (Figure 6c). Figures S8-10 and S11, supplementary analyses, are cited as confirming the power spectrum analysis results, but also bring out issues in the temporal continuity of these spectra and their directionality.

The Brönnimann et al. paper was published in 2007, not 2006 (this issue could be present in other references, and should be checked).

The GIM coral data seem to have been first published in Pfeiffer et al. 2009, not 2017.

All of the supplemental figures are mislabeled, and should be corrected. I recommend, in fact, that the entire Supplemental file be carefully reviewed and edited, as I noticed consistent issues in the labeling of materials and numerous typographical errors.

Ln 30: The opening sentence of the introduction reads a bit awkwardly, I would suggest rewording to something like "As the impacts of global climate change increase,

paleoclimate research is more important than ever". On the same line, I would remove the first word of the second sentence ("Especially") and simply begin the sentence with "The Indian Ocean...". Ln 32: should be "basin", not "basing". Ln 34: Remove "As" and begin the sentence with "Tropical corals". Ln 35: the sentence here continued from Ln 34 is somewhat awkwardly worded, and should be ended with "variability" not "variabilities". Ln 37-39: the sentence in this section repeats its point in the second half, I would delete the section after the first citation of Pfeiffer et al. 2006. Ln 40: change "are focusing on" to "focus on". Ln 41-42: change "lack of data in" to "lack of data from", remove "still" and "the" from the phrase "still limits the" and replace "the" with "our", and change "variabilities" to the singular. Ln 46: change "In fact, it was suggested" to "It is suggested". Ln 68: change "form" to "from". Also recommend moving the phrase "from October to April" from beginning to end of sentence. Ln 166: "Composite" should be "Composites". Ln 200 and 203: ranges in both of these lines contain values to three significant digits, while all others reported in paper are only to two. Ln 293: there is a period missing between "Indian Ocean" and "For". Ln 308: the end of the sentence here should read "Brönnimann et al. (2006) (Table 6)". Ln 313: should read "Indian" not "India" monsoon. Ln 337: remove the "events" before "non-La Niña", and make sure to correct the spelling of La Niña.

Please also note the supplement to this comment:
https://www.clim-past-discuss.net/cp-2020-22/cp-2020-22-RC2-supplement.pdf

**Supplement:**

[revised manuscript text omitted]

---

## Author Comment (AC2) · 28 May 2020

We are grateful for the feedback provided by an anonymous reviewer. The reviewer raises seventeen (RC1 –1 to RC1 – 17) specific comments, which are addressed in detail below. Additionally, technical corrections are provided by the reviewer, which are addressed below, as well. In the following, we will repeat the reviewer's statements (in bold font) and our reply to it. Below the responses to these specific comments, we respond to the technical corrections except for cases where e.g. typos are highlighted.

**Specific comments**

**RC1 - 1**

**- l104: "ENSO Indices" There is in my opinion more recent ENSO time series that you coul have. Why did you chose to use only "old" reconstructions?**

We wanted to use as few indices as possible, and the same indices for all coral time windows shown in our study, for consistency. However, we did not only rely on the Quinn record from 1993. We compared Quinn 1993 with the list of ENSO events compiled in Brönnimann et al., 2007 (https://doi.org/10.1007/s00382-006-0175-z). We believe that this gives some indication of the sensitivity of our results with respect to different ENSO reconstructions. Both records cover all our coral time windows, including our 17th century coral record. Brönnimann et al. (2007) combined several reconstructed ENSO indices (ERSST NINO3 by Smith and Reynolds, 2004 (https://doi.org/10.1175/1520-0442(2004)017<2466:IEROS>2.0.CO;2); Mann NINO3 by Mann et al., 2000 (https://doi.org/10.1175/1087-3562(2000)004<0001:GTPIPC>2.3.CO;2); Cook/D'Arrigo NINO3 by Cook, 2000 (https://www.ncdc.noaa.gov/paleo-search/study/6250) and D'Arrigo et al., 2005 (https://doi.org/10.1029/2004GL022055); Stahle SOI by Stahle et al., 1998 (https://doi.org/10.1175/1520-0477(1998)079<2137:EDROTS>2.0.CO;2), climate field reconstructions and early instrumental data and also assessed the data for consistency.

**RC1 - 2**

**- l137: "concentration of approximately 8ppm Calcium". I'm a bit surprised by this value. Usually with 0.6 mg of carbonate powder in 6mL you are getting 50ppm of Calcium. Additionally, Villiers et al. (2002) that you are citing above recommend analyzing between 40 to 60 ppm of Ca.**

The concentration of Ca in the measuring solution is determined by the concentration of the trace element to be analyzed:

For Mg (Mg/Ca) in foraminifers which is the target analyte of the de Villiers et al. (2002) and Schrag (1999) papers we use Ca at 40-50 mg/L giving best response of the Mg 279 line in our instrument at given conditions.

For Sr (Sr/Ca) in corals we use the Sr-407 (or Sr-421) emission line with very high sensitivity in our instrument so that we had to reduce the Ca concentration to 8 mg/L. This also allows for reduced sample weights and, therefore, for higher time resolution in the coral.

It is the concentration and response of the trace element that defines the required concentrations in solution for getting best counting statistics. The Ca concentrations are, as a consequence, determined by the trace element response and the Mg/Ca and Sr/Ca ratio of the sample, and not vice versa.

**RC1 - 3**

**- l140: Can you, please, indicate why you did not use any CRM such as JCp-1? Can you, please, be more specific and give the mean % of recovery ± SD and the N values**

In total, 5 different standards were used, including the international standards JC-p-1 and JC-t-1. They were measured before and after the entire measurement sequence. We will explain this in more

detail in the methods section in a revised version. Furthermore, we will add error bars on all coral Sr/Ca graphs and will give precision values.

**RC1 - 4**

- **l149: "We assigned the highest Sr/Ca value to the SST minimum of each year and interpolated linearly between these anchor points to obtain a time series with equidistant time steps" Which software did you use? Why only interpolate between two consecutive high values? Why not between two lows or between one high and one low as it is usually the case?**

We developed the age model following the pioneering work of Charles et al., 1997 (https://science.sciencemag.org/content/277/5328/925), who has proposed to use the month of August as one single anchor point in any given year at the Seychelles, a site located slightly further west than Chagos with a similar monsoon-dominated SST seasonality. Charles et al. have demonstrated that with their approach, monthly anomalies can be computed from coral proxy data, and that these monthly anomalies can be correlated (calibrated, in fact) with instrumental SST anomalies. See Charles et al., 1997, Figure 2 B.

Due to the strong cooling of the western and central Indian Ocean following the onset of the Indian summer monsoon in boreal summer, which is seasonally phase-locked, this age model is very precise (the non-cumulative age model error is +/-1 month in any given year). Each additional anchor point would introduce an additional error which, in the Indian Ocean, tends to be larger during the other seasons of the year.

The approach proposed by the Reviewer (interpolating between two lows or between one high and one low) would only be applicable at sites that have large-amplitude, sinusoidal seasonal cycles, where age model errors become a problem in the transitional seasons in fall and spring due to the rapid change in SST during a short time period.

**RC1 - 5**

- **l193: I would have liked to see how you determine the annual mean (average between two max Sr/Ca) ? and how did you determine the standard error?**

The annual means were generated by averaging every values of one year (Jan-Dec). When averaging different values, you can determine a standard deviation for this calculation what we did with Excel. With the standard deviation, we calculated the standard error with the formula below.

The standard error (SE) were used and calculated as follows:

SE = (standard deviation ($\sigma$))/$\sqrt{}$ (Number of values (n))

**RC1 - 6**

- **l218: Can you please explain how and especially why you decided to detrend the record?**

For detrending we used published methods by Mudelsee, 2000 (https://doi.org/10.1016/s0098-3004(99)00141-7) and Mudelsee, 2009 (https://doi.org/10.1140/epjst/e2009-01089-3). As it can be seen in figure S6 in the Supplementary Material, the long-term trend was subtracted and not, e.g., the annual cycle. Detrending was necessary for time series analysis (singular spectrum and power spectrum analysis) and to compile the composite records. With the long-term trend being subtracted out, the anomaly events could be detected (see also figure S6).

**RC1 - 7**

- **l223: You should not cite Figure 7-9 while talking about ENSO event frequency, those figures are not at all giving information on frequencies.**

We agree, that figures 7-9 do not say anything about ENSO frequency. We will change the reference in the revised version.

**RC1 - 8**

- l228: I'm confused l226 you indicate that El Nino events occurs every 5 years between 1965 and 1995 and l228 you mention a recurrence time of 3.6 years ...

We agree that this might lead to confusion. The recurrence time of 3.6 years between 1965-1995 mentioned in l228 is taken from the list of events in Quinn 1993 (as mention one sentence before in l227). We wanted to compare the recurrence times calculated from our coral records with that calculated from published data. We will adjust this paragraph in a revised version so that it will not lead to any confusion anymore.

**RC1 - 9**

- l229: Can you please indicate here which threshold you used when considering an anomaly and therefore considering that it is an El Nino or La Nina year?

The events described in this paragraph are the same listed in Tables 4, 5 and 6. They were picked as described in section 4.5 in the manuscript: "…Positive SST anomalies in the coral records were interpreted as positive ENSO events when the year of occurrence was listed as one with large-scale ENSO event in Quinn (1993) and Brönnimann et al. (2007) within the error of each coral age model and **when the anomaly exceeds 1.5 standard deviations of the mean of each coral record** (Fig. S6). In addition to the strong La Niña events listed in Brönnimann et al. (2007), we added negative SST anomalies occurring in years after the El Niño years to the composite."

**RC1 - 10**

- l240: Using data from Quinn (1993). I do believe that there is more up to date studies on ENSO events... I would feel more confident in your results if you had compared to multiple studies.

As also mentioned in l240, we not only used data from Quinn (1993), but also from Brönnimann et al., 2007 (https://doi.org/10.1007/s00382-006-0175-z), who provides a synthesis of multiple ENSO reconstructions. For an extended explanation why we used these lists/indices, see our comment below reviewer comment RC1 - 1 in this document.

**RC1 - 11**

- l241: "error of each coral age model". What values did you use as a bracket for the age model uncertainty?

As mentioned in section 3.3 Chronology (in l152), the uncertainties of the age models are approximately ±1.9 years (E5), ±2.2 years (B8) and ±2.4 years (E3).

**RC1 - 12**

- l242: "we added negative SST anomalies occurring in years after the El Niño years to the composite." I do not understand - are those considered La Niña-like events?

As La Niña events tend to occur after El Niño events, e.g. Cai et al., 2015 (https://doi.org/10.1038/nclimate2492), we interpreted negative anomaly events exceeding 1.5 standard deviations of the mean of each coral record (Fig. S6) and occurring after El Niño events as La Niña events.

**RC1 - 13**

- l266: "the greater sensitivity of the corals to reef-scale". Can it reflect also issues with the calibration you used to convert Sr/Ca to SST?

A detailed calibration for modern corals from the same site (Chagos Archipelago) was presented in Leupold et al., 2019 (https://doi.org/10.1029/2018GC007796). In this study, the regression of coral Sr/Ca with satellite data indicates a significant correlation (r-squared: 0.62, p<<0.01, n=265). As we used the slope of this regression in our study, it probably does not reflect issues with the calibration we used to convert Sr/Ca to SST.

**RC1 - 14**

**- l301-032: "However, … "Something is wrong with this sentence. How could differences between means are not significantly different with one p value of 0.9 and one of 0.07 …. Additionally, does that mean that the decrease in amplitude of the negative anomalies are not statistically significative?**

In most field of scientific work, it is common to apply a confidence level of 99%. In this case, the corresponding significance level is 0.01. Both p values (0.9 and 0.07) are larger than our significance level of 0.01, i.e. that the difference between the means is not statistically significant. So, yes, this means that the decrease in amplitude of the negative anomalies between the period 1830-1929 and the period 1965-1995 is not statistically significant.

**RC1 - 15**

**- l322-323: Can you please develop a bit more on this idea.**

We are not sure what idea is exactly meant. We described larger anomaly amplitudes during a period of general cooler mean temperatures, which is consistent with results shown in Pfeiffer et al., 2017 (https://doi.org/10.1038/s41598-017-14352-6) and Zinke et al. 2004 (https://doi.org/10.1016/j.epsl.2004.09.028).

**RC1 - 16**

**- l325: "comparable". Comparable to what? You also need to be consistent, you have been using the term "coral composite" and know you are using "Chacos coral"... it is a bit confusing, we are wondering if you are not referring to something else ...**

With this sentence we want to say that El Niño magnitudes are comparable to La Niña magnitudes. We used Chagos coral records in this sentence, because the El Niño and La Niña events are recorded in the corals from Chagos, they are not recorded in the "Composites". The composites were then generated by selecting the El Niño and La Niña event anomalies recorded in the Chagos corals.

**RC1 - 17**

**- l344: You mean the frequency because the strength of the events are different, there is no change in strength of ENSO events in your study during the 20th century (Figure 9).**

We are not sure what the reviewer wants to point at with this comment, because in l344 we did not write anything about the strength of events.

**Technical corrections**

**- l51: "The modern core was included in a composite reconstruction of large-scale SST (Pfeiffer et al., 2017) and the core top (1950-1995) was shown to record SST variability at Chagos on grid-SST scale (Pfeiffer et al., 2009)." I'm a bit confused by sentence. Maybe should explain a bit more what you had in mind while writing it. One core records more global scale SST variability while the other more the local variability? Is that it?**

We simply wanted to say that core GIM has been calibrated with SST in a previous study. In addition, it has been part of a composite coral record for the western equatorial Indian Ocean (together with corals from the Seychelles). We will revise this sentence.

**- l55: "We identify past warm and cold events in each record and use these events to compile composites to evaluate the symmetry of positive and negative ENSO-driven SST anomaly events in the tropical Indian Ocean." This paragraph seems out of context here.**

With this sentence we explain what we will focus on in this study. We think, it is essential to have it in the introduction. However, we will add more text on ENSO asymmetry in the introduction of a revised version of our manuscript and revise this sentence to better convey the main aims of our study.

- **l62: "… water exchange with the open ocean is substantial." Do you specify that because your coral core is from inside the lagoon?**

We specified it to give a better idea of the setting. We do not know where the coral lived exactly as they were found as boulders at the beach or in derelict buildings.

- **l98: "Averaged over the entire area of the Chagos (70-74° E; 4-8° S), SST is similar…" It would be interesting to add the mean values for both sites.**

We will mention the mean SST for both sites in the revised version. We do not interpret the mean values of our Sr/Ca data. In corals, these are influenced by vital effects, see e.g. Sayani et al., 2019 (https://doi.org/10.1029/2019GC008420).

- **l104: "ENSO Indices" You might want to introduce this paragraph as the time series you will compare your records to? Or something in this line.**

We agree, we will add to this paragraph that the indices presented here are the ones we use for comparison with our coral data.

- **l125: "The core top (1950-1995) was shown to record SST variability at Chagos on a grid-SST scale (Pfeiffer et al., 2009). The entire record was included in a composite reconstruction of large-scale western Indian Ocean SST (Pfeiffer et al., 2017)." Those sentences are similar to the ones l51 that I did not understand. The top core was comapred to grid-SST data and it maches perfectly and then the entire record was used in a coral composite but with which other corals? Can you, please, add more information here.**

The GIM core was included in the coral composite of the Seychelles-Chagos thermocline ridge. This composite comprises cores from the Seychelles and Chagos. We will add this in a revised version of this manuscript.

- **l128: "From the slabs of the sub-fossil corals, powder samples were drilled at 1 mm increments using a micro-milling machine (type PROXXON FF 500 CNC). This depth resolution can be translated to monthly temporal resolution with average growth rates being 12 mm/yr. The subsampling paths were always set along the optimal growth axis that was determined based on x-ray images (Fig. S2)." Can you please add some information on the sampling over laps that you had to do when switching sampling paths? How did you determine the temporal resolution, by looking at the density band or by looking at the seasonal cycles in Sr/Ca data? You might want to move this paragraph up right below where you talk about your new coral core samples.**

All sampling paths were selected so that we get a continuous record for each coral sample. This includes, e.g., both sampling paths on coral slab E3. For this sample, there is an overlap of 10 mm, which means 10 subsamples, for each sampling path, i.e. there is around one year of overlap. We determined the temporal resolution by combining the interpretation of the annual bands visible in the x-rays and the seasonal cycles recorded in the Sr/Ca data.

We will move this paragraph below the paragraph in which we talk about our new coral core samples. In a revised version of this manuscript, we will show the raw data, including the overlaps, as requested by reviewer 2.

- **l138: "The intensities of Strontium and Calcium were converted into Sr/Ca ratios in mmol/mol." Which method did you use to convert the instrument output in intensity to concentration values 1) the calibration given to you by the instrument? or 2) the deVillier et al., 2002 ratio method?**

We convert the intensities of Sr and Ca into Sr/Ca ratios following the methodology proposed by deVillier et al., 2002.

- **l165: Statistic Section : I would like to see this section a bit above as you use statistics in above paragraphs**

We do not see why we should put the statistic section above the other sections in the methods section, because it only introduces statistics used and described in the results and discussion part of the manuscript.

**- l189: In my opinion you do not need subsections, but instead a big paragraph labelled Sr/Ca data description, where you describe the results core by core**
We think that subsections give a better overview as we have coral samples from different time windows.

**- l196: "The range …" Is that the mean range or the maximum range?**
It is the range between the maximum and minimum Sr/Ca value (see also table 2).

**- l206: Can you please describe how you determine the mean annual cycle?**
The mean annual cycles were calculated by averaging interpolated Sr/Ca values for every month over the given time period covered by each coral record. For example, B8 covers 31 years from 1836-1867. For this period, all Sr/Ca values for January were averaged, all Sr/Ca values for February were averaged, all Sr/Ca values for March were averaged, and so on.

**- l209: I do believe you should spend a little more time describing Figure 4.**
We will add a more detailed description of Figure 4 in the revised version.

**- l224: "Our results show that,…" I'm guessing that these conclusions derived from Table 4-6: you might want to refer to it as well as indicate some stats about this change of frequency. Maybe the percentage of increased frequency?**
Yes, these conclusions derived from Tables 4-6. We indicated this with an introducing sentence that included the reference to Tables 4-6 at the beginning of the paragraph. However, we can include it once again at the end of the follow-up sentence. We do not see how percentage of increased frequency could help to better illustrates the results described in this paragraph.

**- l241: "referring to Figure S6". Figure S6 correspond to the detcoral Sr/Ca records after detrending. Which Figure are you referring to here?**
We are referring to Figure S6. In the lower plot, 1.5x of the standard deviation is plotted as dashed lines. Peaks above this standard deviation were considered as anomaly events when listed in Quinn (1993) or Brönnimann et al. (2007). Detrending was necessary for compiling the composite records. With the long-term trend being subtracted out, the anomaly events could be detected. That is why we indicated the standard deviation in this figure and that is why we referred to this figure in this paragraph.

**- l245 – 251: This paragraph should be in the method section.**
We did not put this in the methods section, because it already consists of our interpretation. We interpret the anomaly events to be El Niño and La Niña events. We think, putting this paragraph in the method section would confuse the reader as it requires information which is provided only later in the manuscript.

**- l259: "we compared". I do not think "compare" is the right word. You do not compare, you use the same technique to discriminates El Nino from La Nina from negative events other than La Nina years, right?**
That is correct, we also selected the events recorded in the satellite data using the same techniques. But in the end, we compared the results of the satellite data (how many events and the amplitudes) with our coral data.

**-** **l263: "All SST anomalies were … of -0.06 mmol/mol per 1°C (see Leupold et al., 2019)." This section looks more like a material and method section. You do not talk at all about what you found.**

This paragraph primarily focusses on the anomaly events we interpreted. However, we agree that we should put sentences like this, regarding the calculation of the SST anomalies, in the methods section.

**-** **l265: "Coral SST proxy". What is that? Is it your so-called ENSO composite?**

It is the coral Sr/Ca data, which is used as an SST proxy. We used this expression here, because the ENSO composite is the result of calculations we did with our coral data. We will improve the wording in the revision.

**-** **l288: "On average … (p=0,75)". This sentence is a bit similar to the first sentence of the paragraph, no? You might want to regroup them.**

Thank you for pointing that out. We will adjust it in a revised version.

**-** **Figure S1 : Can you please add an arrow to actually point at the boulder you sampled? Can you please add a symbol of the lagoon of Peros Banhos site location on your Map**

In figure S1a, a saw can be seen. Above this saw there is pile of boulders. Two of these boulders are E3 and E5. We will add arrows marking E3 and E5. In figure 1 (location of our study area), the lagoon of Peros Banhos is already labeled.

---

## Author Comment (AC4) · 28 May 2020

In our Response to RC1, a figure reference was mentioned without including the corresponding figure. Please find here the figure to the corresponding paragraph in line 24: "ENSO (and IOD)-induced SST anomalies are small in the Indian Ocean ($\sim$0.5-0.7°C, see Figure 2 of our manuscript and Roxy et al., 2014) relative to the background variability ($\sim$0.3-0.4°C in the peak ENSO season from December-February; see Figure below)."
* * *
[Figure]

**sd Dec−Feb averaged NCEP OIv2 1/4 SST anom [Celsius]**
**1981:2019**

0.2  0.3  0.4  0.5  0.6  0.7

**Fig. 1.**

---

## Author Response (AR1)

We are grateful for the feedback provided by an anonymous reviewer. The reviewer raises seventeen (RC1 –1 to RC1 – 17) specific comments, which are addressed in detail below. Additionally, technical corrections are provided by the reviewer, which are addressed below, as well. In the following, we will repeat the reviewer's statements (in bold font) and our reply to it. Below the responses to these specific comments, we respond to the technical corrections. Numbers in brackets at the beginning of our comments indicate line numbers of the revised version of the manuscript with highlighted changes.

**Specific comments**

**RC1 - 1**
**- l104: "ENSO Indices" There is in my opinion more recent ENSO time series that you coul have. Why did you chose to use only "old" reconstructions?**
(l. 122) We wanted to use as few indices as possible, and the same indices for all coral time windows shown in our study, for consistency. However, we did not only rely on the Quinn record from 1993. We compared Quinn 1993 with the list of ENSO events compiled in Brönnimann et al., 2007 (https://doi.org/10.1007/s00382-006-0175-z). We believe that this gives some indication of the sensitivity of our results with respect to different ENSO reconstructions. Both records cover all our coral time windows, including our 17th century coral record. Brönnimann et al. (2007) combined several reconstructed ENSO indices (ERSST NINO3 by Smith and Reynolds, 2004 (https://doi.org/10.1175/1520-0442(2004)017<2466:IEROS>2.0.CO;2); Mann NINO3 by Mann et al., 2000 (https://doi.org/10.1175/1087-3562(2000)004<0001:GTPIPC>2.3.CO;2); Cook/D'Arrigo NINO3 by Cook, 2000 (https://www.ncdc.noaa.gov/paleo-search/study/6250) and D'Arrigo et al., 2005 (https://doi.org/10.1029/2004GL022055); Stahle SOI by Stahle et al., 1998 (https://doi.org/10.1175/1520-0477(1998)079<2137:EDROTS>2.0.CO;2)), climate field reconstructions and early instrumental data and also assessed the data for consistency.

**RC1 - 2**
**- l137: "concentration of approximately 8ppm Calcium". I'm a bit surprised by this value. Usually with 0.6 mg of carbonate powder in 6mL you are getting 50ppm of Calcium. Additionally, Villiers et al. (2002) that you are citing above recommend analyzing between 40 to 60 ppm of Ca.**
(l. 166) We do not dilute 0.6 mg of carbonate powder in 6mL. In a first step, we dilute the powder in 1.00 mL 0.2 M $HNO_3$. In a second step, we take a specified volume from this solution and add 0.2 M $HNO3$ to get a final concentration of approximately 8ppm Calcium in 6mL.
The concentration of Ca in the measuring solution is determined by the concentration of the trace element to be analyzed:
For Mg (Mg/Ca) in foraminifers which is the target analyte of the de Villiers et al. (2002) and Schrag (1999) papers we use Ca at 40-50 mg/L giving best response of the Mg 279 line in our instrument at given conditions.
For Sr (Sr/Ca) in corals we use the Sr-407 emission line with very high sensitivity in our instrument so that we had to reduce the Ca concentration to 8 mg/L. This also allows for reduced sample weights and, therefore, for higher time resolution in the coral.
It is the concentration and response of the trace element that defines the required concentrations in solution for getting best counting statistics. The Ca concentrations are, as a consequence, determined by the trace element response and the Mg/Ca and Sr/Ca ratio of the sample, and not vice versa.

**RC1 - 3**

**- l140: Can you, please, indicate why you did not use any CRM such as JCp-1? Can you, please, be more specific and give the mean % of recovery ± SD and the N values**

(l. 169) In total, 5 different CRM were used, including the international CRM JCp-1 and JCt-1. They were measured before and after each measurement sequence. We explained this in more detail in the methods section in the revised version. Furthermore, we added error bars on all coral Sr/Ca plots and added precision values in the methods section.

**RC1 - 4**

**- l149: "We assigned the highest Sr/Ca value to the SST minimum of each year and interpolated linearly between these anchor points to obtain a time series with equidistant time steps" Which software did you use? Why only interpolate between two consecutive high values? Why not between two lows or between one high and one low as it is usually the case?**

(l. 187) We developed the age model following the pioneering work of Charles et al., 1997 (https://science.sciencemag.org/content/277/5328/925), who has proposed to use the month of August as one single anchor point in any given year at the Seychelles, a site located slightly further west than Chagos with a similar monsoon-dominated SST seasonality. Charles et al. have demonstrated that with their approach, monthly anomalies can be computed from coral proxy data, and that these monthly anomalies can be correlated (calibrated, in fact) with instrumental SST anomalies. See Charles et al., 1997, Figure 2 B.

Due to the strong cooling of the western and central Indian Ocean following the onset of the Indian summer monsoon in boreal summer, which is seasonally phase-locked, this age model is very precise (the non-cumulative age model error is +/-1 month in any given year). Each additional anchor point would introduce an additional error which, in the Indian Ocean, tends to be larger during the other seasons of the year.

Using only the lows would also introduce a larger error, as the austral summer warm season is longer and more variable then the monsoon-induced cool season.

**RC1 - 5**

**- l193: I would have liked to see how you determine the annual mean (average between two max Sr/Ca) ? and how did you determine the standard error?**

(l. 228) The annual means were generated by averaging every value of one year (Jan-Dec). When averaging different values, you can determine a standard deviation for this calculation, what we did with Excel. With the standard deviation, we calculated the standard error with the formula below.

The standard error (SE) were used and calculated as follows:

SE = (standard deviation ($\sigma$ ))/$\sqrt{}$ (Number of values (n))

**RC1 - 6**

**- l218: Can you please explain how and especially why you decided to detrend the record?**

For detrending we used published methods by Mudelsee, 2000 (https://doi.org/10.1016/s0098-3004(99)00141-7) and Mudelsee, 2009 (https://doi.org/10.1140/epjst/e2009-01089-3). As it can be seen in figure S7 in the revised Supplementary Material, the long-term trend was subtracted and not, e.g., the annual cycle. Detrending was necessary for time series analysis (singular spectrum and power spectrum analysis) and to compile the composite records. With the long-term trend being subtracted out, the anomaly events could be detected (see also figure S7).

**RC1 - 7**

**- l223: You should not cite Figure 7-9 while talking about ENSO event frequency, those figures are not at all giving information on frequencies.**

95 (l. 269) We agree, that figures 7-9 do not say anything about ENSO frequency. We revised this paragraph to shorten the interpretation on ENSO frequency and to make our focus more central.

**RC1 - 8**
**- l228: I'm confused l226 you indicate that El Nino events occurs every 5 years between 1965 and 1995 and l228 you mention a recurrence time of 3.6 years ...**
100 We agree that this might lead to confusion. The recurrence time of 3.6 years between 1965-1995 mentioned in l228 is taken from the list of events in Quinn 1993 (as mention one sentence before in l227). We wanted to compare the recurrence times calculated from our coral records with that calculated from published data. However, we revised this paragraph anyway as we shortened the sections on the time series analysis, as these were only used to show that ENSO periodicity is 105 observed in the coral records and may distract the reader from our main results.

**RC1 - 9**
**- l229: Can you please indicate here which threshold you used when considering an anomaly and therefore considering that it is an El Nino or La Nina year?**
110 The events described in this paragraph are the same listed in Tables 4, 5 and 6. They were picked as described in section 4.5 in the manuscript: "…Positive SST anomalies in the coral records were interpreted as positive ENSO events when the year of occurrence was listed as one with large-scale ENSO event in Quinn (1993) and Brönnimann et al. (2007) within the error of each coral age model and **when the anomaly exceeds 1.5 standard deviations of the mean of each coral record** (Fig. S7 115 of the revised supplementary material). In addition to the strong La Niña events listed in Brönnimann et al. (2007), we added negative SST anomalies occurring in years after the El Niño years to the composite."

**RC1 - 10**
**- l240: Using data from Quinn (1993). I do believe that there is more up to date studies on ENSO**
120 **events... I would feel more confident in your results if you had compared to multiple studies.**
(l. 283) As also mentioned in l240, we not only used data from Quinn (1993), but also from Brönnimann et al., 2007 (https://doi.org/10.1007/s00382-006-0175-z), who provides a synthesis of multiple ENSO reconstructions. For an extended explanation why we used these lists/indices, see our comment below reviewer comment RC1 - 1 in this document.

125 **RC1 - 11**
**- l241: "error of each coral age model". What values did you use as a bracket for the age model uncertainty?**
(l. 289) As mentioned in section 3.3 Chronology (in l152 of the initially submitted manuscript; in l. 177 of the revised version), the uncertainties of the age models are approximately ±1.9 years (E5), 130 ±2.2 years (B8) and ±2.4 years (E3).

**RC1 - 12**
**- l242: "we added negative SST anomalies occurring in years after the El Niño years to the composite." I do not understand - are those considered La Niña-like events?**
(l. 290) As La Niña events tend to occur after El Niño events, e.g. Cai et al., 2015 135 (https://doi.org/10.1038/nclimate2492), we interpreted negative anomaly events exceeding 1.5 standard deviations of the mean of each coral record (Fig. S7 of the revised supplementary material) and occurring after El Niño events as La Niña events.

**RC1 - 13**
**- l266: "the greater sensitivity of the corals to reef-scale". Can it reflect also issues with the**
140 **calibration you used to convert Sr/Ca to SST?**

(l. 314) A detailed calibration for modern corals from the same site (Chagos Archipelago) was presented in Leupold et al., 2019 (https://doi.org/10.1029/2018GC007796). In this study, the regression of coral Sr/Ca with satellite data indicates a significant correlation (r-squared: 0.62, p<<0.01, n=265). As we used the slope of this regression in our study, it probably does not reflect issues with the calibration we used to convert Sr/Ca to SST.

**RC1 - 14**

- **l301-032: "However, … "Something is wrong with this sentence. How could differences between means are not significantly different with one p value of 0.9 and one of 0.07 .... Additionally, does that mean that the decrease in amplitude of the negative anomalies are not statistically significative?**

(l. 349) In most fields of scientific work, it is common to apply a confidence level of 99%. In this case, the corresponding significance level is 0.01. Both p values (0.9 and 0.07) are larger than our significance level of 0.01, i.e. they show that the difference between the means is not statistically significant. So, yes, this means that the decrease in amplitude of the negative anomalies between the period 1830-1929 and the period 1965-1995 is not statistically significant.

**RC1 - 15**

- **l322-323: Can you please develop a bit more on this idea.**

(l. 383) In this paragraph, we described larger anomaly amplitudes during a period of general cooler mean temperatures, which is consistent with results shown in Pfeiffer et al., 2017 (https://doi.org/10.1038/s41598-017-14352-6) and Zinke et al. 2004 (https://doi.org/10.1016/j.epsl.2004.09.028). We are not sure why this is the case. However, the tropical Indian Ocean is the warmest tropical Ocean, and recent instrumental data suggests that, as the Indian Ocean continues to warm, the temperature variability reduces particularly in the warm season, while SSTs in the cold season show strongest warming (e.g. Leupold et al., 2019 (https://doi.org/10.1029/2018GC007796); Roxy et al., 2014 (https://doi.org/10.1175/JCLI-D-14-00471.1)) and largest spatial variability (Leupold et al, 2019 (https://doi.org/10.1029/2018GC007796)).

**RC1 - 16**

- **l325: "comparable". Comparable to what? You also need to be consistent, you have been using the term "coral composite" and know you are using "Chacos coral"... it is a bit confusing, we are wondering if you are not referring to something else ...**

(l. 360) With this sentence we want to say that El Niño magnitudes are comparable to La Niña magnitudes. We used Chagos coral records in this sentence, because the El Niño and La Niña events are recorded in the corals from Chagos, they are not recorded in the "Composites". The composites were then generated by selecting the El Niño and La Niña event anomalies recorded in the Chagos corals.

**RC1 - 17**

- **l344: You mean the frequency because the strength of the events are different, there is no change in strength of ENSO events in your study during the 20th century (Figure 9).**

We are not sure what the reviewer wants to point at with this comment, because in l344 we did not write anything about the strength of events.

**Technical corrections**

**You need to change all the reference of the Supplementary figures as they are wrongly numbered in the text**

190   We did not find that they were wrongly numbered. However, as the other reviewer also mentioned wrong numbering of the supplementary figures, we changed the titles of each chapter in the supplementary material which may have caused the misunderstanding. In the originally submitted version, e.g. chapter 3 of the supplementary material was named 'S3 X-ray images' that included 'Figure S2'. After revision, this chapter's title is now '3 X-ray images'.

195   **You need to change the numbering of the Supplementary Figures legends.**
      Revised.

      **You need to cite Figure S1 in the text**
      Figure S1 was already cited in the originally submitted manuscript (as 'Fig. S1' in line 118).

200

      **- l37 : "There are only some studies including Sr/Ca measurements for SST reconstructions (e.g. Pfeiffer et al., 2006), while few studies included Sr/Ca measurements for SST reconstructions (e.g. Pfeiffer et al., 2006)."There is a problem with this sentence. You wrote twice the same idea.**
      (l. 37) Thank you for pointing this out. We revised this sentence.

205

      **- l39: "Most studies are focusing on either the western or the eastern Indian Ocean (Abram et al., 2003; Watanabe et al., 2019) and/or are sampled at only bimonthly … " Be careful it is not the study that are sampled at bi monthly resolution but the coral.**
      (l. 37) We revised this sentence.

210

      **- l51: "The modern core was included in a composite reconstruction of large-scale SST (Pfeiffer et al., 2017) and the core top (1950-1995) was shown to record SST variability at Chagos on grid-SST scale (Pfeiffer et al., 2009)." I'm a bit confused by sentence. Maybe should explain a bit more what you had in mind while writing it. One core records more global scale SST variability while the other more**
215   **the local variability? Is that it?**
      (l. 66) We simply wanted to say that core GIM has been calibrated with SST in a previous study. In addition, it has been part of a composite coral record for the western equatorial Indian Ocean (together with corals from the Seychelles). We revised this paragraph.

220   **- l53: "41 years of the Maunder" Can you write instead "41 years during the Maunder**
      (l. 65) We adjusted this sentence but exchanged "Maunder Minimum" with "period of the Little Ice Age", as we used the term Maunder Minimum in a misleading/incorrect way.

      **- l54: "39 years of the mid-19th to early 20th century (1870-1909) covering 39 years". You wrote**
225   **twice the same info about 39 years.**
      We shortened this paragraph.

      **- l55: "We identify past warm and cold events in each record and use these events to compile composites to evaluate the symmetry of positive and negative ENSO-driven SST anomaly events in**
230   **the tropical Indian Ocean." This paragraph seems out of context here.**
      (l. 70) With this sentence we explain what we will focus on in this study. We think, it is essential to have it in the introduction. However, we added more text on ENSO asymmetry in the introduction of a revised version of our manuscript and revised this sentence to better convey the main aims of our study.

235   **- l62: "… water exchange with the open ocean is substantial." Do you specify that because your coral core is from inside the lagoon?**

(l. 80) We specified it to give a better idea of the setting. We do not know where the coral lived exactly as they were found as boulders at the beach or in derelict buildings.

240 **- l98: "Averaged over the entire area of the Chagos (70-74° E; 4-8° S), SST is similar…" It would be interesting to add the mean values for both sites.**

(l. 114) We added the mean SST for both sites in the revised version. We do not interpret the mean values of our Sr/Ca data. In corals, these are influenced by vital effects, see e.g. Sayani et al., 2019 (https://doi.org/10.1029/2019GC008420).

245

**- l104: "ENSO Indices" You might want to introduce this paragraph as the time series you will compare your records to? Or something in this line.**

(l. 122) We agree, we added a sentence stating that the indices presented here are the ones we use for comparison with our coral data.

250

**- l125: "The core top (1950-1995) was shown to record SST variability at Chagos on a grid-SST scale (Pfeiffer et al., 2009). The entire record was included in a composite reconstruction of large-scale western Indian Ocean SST (Pfeiffer et al., 2017)." Those sentences are similar to the ones l51 that I did not understand. The top core was comapred to grid-SST data and it maches perfectly and then**
255 **the entire record was used in a coral composite but with which other corals? Can you, please, add more information here.**

(l. 155) The GIM core was included in the coral composite of the Seychelles-Chagos thermocline ridge. This composite comprises cores from the Seychelles and Chagos. We added this information in the revised version.

260

**- l128: "From the slabs of the sub-fossil corals, powder samples were drilled at 1 mm increments using a micro-milling machine (type PROXXON FF 500 CNC). This depth resolution can be translated to monthly temporal resolution with average growth rates being 12 mm/yr. The subsampling paths were always set along the optimal growth axis that was determined based on x-ray images (Fig. S2)."**
265 **Can you please add some information on the sampling over laps that you had to do when switching sampling paths? How did you determine the temporal resolution, by looking at the density band or by looking at the seasonal cycles in Sr/Ca data? You might want to move this paragraph up right below where you talk about your new coral core samples.**

(l. 151) All sampling paths were selected so that we get a continuous record for each coral sample.
270 This includes, e.g., both sampling paths on coral slab E3. For this sample, there is an overlap of 10 mm, which means 10 subsamples, for each sampling path, i.e. there is around one year of overlap. We determined the temporal resolution by combining the interpretation of the annual bands visible in the x-rays and the seasonal cycles recorded in the Sr/Ca data.

We moved this paragraph below the paragraph in which we talk about our new coral core samples.
275 In the revised version of this manuscript (in the supplementary material), we show the raw Sr/Ca data, including the overlaps, as requested by reviewer 2.

**- l138: "The intensities of Strontium and Calcium were converted into Sr/Ca ratios in mmol/mol." Which method did you use to convert the instrument output in intensity to concentration values 1) the calibration given to you by the instrument? or 2) the deVillier et al., 2002 ratio method?**
280 (l. 168) We converted the intensities of Sr and Ca into Sr/Ca ratios following the methodology proposed by deVillier et al., 2002.

**- l165: Statistic Section : I would like to see this section a bit above as you use statistics in above paragraphs**

285     (l. 198) We do not see why we should put the statistic section above the other sections in the methods section, because it only introduces statistics used and described in the results and discussion part of the manuscript.

**- l166: "Composite were generated calculating…" replace by "Composite were generated by**
290 **calculating…"**
    (l. 205) Done.

**- l175: Can you please indicate in which occasion you use the t-test**
    (l. 209) T-tests were used to determine if the mean values of two data sets, e.g. mean annual cycles in chapter 4.3 or mean values of coral composites in chapter 4.5, are significantly different from each
295     other. We added this sentence to the corresponding paragraph in the manuscript.

**- l188: It would be interesting to have at the end of the Diagenesis section a summary sentence stating that your sampled should all be good for geochemical analysis and that the results should not be impacted by secondary calcification ...**
300     (l. 224) Done.

**- l189: In my opinion you do not need subsections, but instead a big paragraph labelled Sr/Ca data description, where you describe the results core by core**
    (l. 226) We think that subsections give a better overview as we have coral samples from different
305     time windows.

**- l192: Porites needs to be in italic**
    (l. 200) Done.

310 **- l196: "The range …" Is that the mean range or the maximum range?**
    (l. 231) It is the range between the maximum and minimum Sr/Ca value (see also table 2). We added the word "maximum".

**- l206: Can you please describe how you determine the mean annual cycle?**
315     (l. 240) The mean annual cycles were calculated by averaging interpolated Sr/Ca values for every month over the given time period covered by each coral record. For example, B8 covers 31 years from 1836-1867. For this period, all Sr/Ca values for January were averaged, all Sr/Ca values for February were averaged, all Sr/Ca values for March were averaged, and so on.

320 **- l207: "The seasonal amplitudes in coral SST [°C] are slightly higher" You should be using parenthesis instead of brackets**
    (l. 248) Done.

**- l209: I do believe you should spend a little more time describing Figure 4.**
325     (l. 241-246) We added a more detailed description of Figure 4 in section 4.3. in the revised version.

**- l224: "Our results show that,…" I'm guessing that these conclusions derived from Table 4-6: you might want to refer to it as well as indicate some stats about this change of frequency. Maybe the percentage of increased frequency?**

330    (l. 269) Yes, these conclusions derived from Tables 4-6. We indicated this with an introducing sentence that included the reference to Tables 4-6 at the beginning of the paragraph. However, we included it once again at the end of the follow-up sentence. Furthermore, we revised this paragraph anyway as we shortened the sections on the time series analysis.

335    **- l225: Replace the ":" by "."**
       (l. 271) Done.

       **- l226: Remove here also the reference to Figure 7-9.**
       (l. 273) Done.

       **- l241: "referring to Figure S6". Figure S6 correspond to the detcoral Sr/Ca records after detrending.**
340    **Which Figure are you referring to here?**
       (l. 290) We are referring to Figure S6 (now Figure S7 in the revised version of the supplementary material). In the lower plot, 1.5x of the standard deviation is plotted as dashed lines. Peaks above this standard deviation were considered as anomaly events when listed in Quinn (1993) or Brönnimann et al. (2007). Detrending was necessary for compiling the composite records. With the
345    long-term trend being subtracted out, the anomaly events could be detected. That is why we indicated the standard deviation in this figure and that is why we referred to this figure in this paragraph.

       **- l245 – 251: This paragraph should be in the method section.**
       (l. 293) We did not put this in the methods section, because it already consists of our interpretation.
350    We interpret the anomaly events to be El Niño and La Niña events. We think, putting this paragraph in the method section would confuse the reader as it requires information which is provided only later in the manuscript.

       **- 256-258: This sentence has no link with the previous sentence and should be separated from it.**
       (l. 305) Done.

355

       **- l259: "we compared". I do not think "compare" is the right word. You do not compare, you use the same technique to discriminates El Nino from La Nina from negative events other than La Nina years, right?**
       (l. 308) That is correct, we also selected the events recorded in the satellite data using the same
360    techniques. But in the end, we compared the results of the satellite data (how many events and the amplitudes) with our coral data.

       **- l263: "All SST anomalies were … of -0.06 mmol/mol per 1°C (see Leupold et al., 2019)." This section looks more like a material and method section. You do not talk at all about what you
365    found.**
       (l. 199) This paragraph primarily focusses on the anomaly events we interpreted. However, we agree that we should put sentences like this, regarding the calculation of the SST anomalies, in the methods section, which we now did in the revised version of the manuscript.

370    **- l265: "Coral SST proxy". What is that? Is it your so-called ENSO composite?**

(l. 313) It is the coral Sr/Ca data, which is used as an SST proxy. We used this expression here, because the ENSO composite is the result of calculations we did with our coral data. We improved the wording in the revision.

375  - **l265: "Ocean record similar, but higher anomalies". If it is higher it is not similar. What do you mean by "similar"?**
(l. 313) We deleted "similar".

- **l269-271: Those two paragraphs talk about the same subject; you should not separate them.**
380  (l. 317) Revised.

- **l274: "from those". You mean from the coral composite, right? It is not clear; you might want to rephrase.**
(l. 322) Revised.

385  - **l288: "On average … (p=0,75)". This sentence is a bit similar to the first sentence of the paragraph, no? You might want to regroup them.**
(l. 329) Thank you for pointing that out. We adjusted it in a revised version.

- **l293: You forgot the "." at the end of the sentence.**
(l. 341) Added.

390  - **l298: You need to regroup this sentence with the next paragraph as they discuss the same idea.**
(l. 329) Done.

- **Figure S1 : Can you please add an arrow to actually point at the boulder you sampled? Can you please add a symbol of the lagoon of Peros Banhos site location on your Map**
We added arrows marking E3 and E5 in the revised version of the supplementary material. In figure
395  1 (location of our study area) of the main manuscript, the lagoon of Peros Banhos is already labeled.

Point-to-point reply to Reviewer #2 comments by Leupold et al.

We are grateful for the feedback provided by an anonymous reviewer. The reviewer raises fourteen (RC2 –1 to RC2 – 14) specific comments, which are addressed in detail below. Additionally, technical corrections are provided by the reviewer, which are addressed below, as well. Furthermore, additional comments to individual points of the manuscript are provided in an annotated pdf. In the following, we will repeat the reviewer's statements (in bold font) and our reply to it. Below the responses to these specific comments, we respond to the technical corrections and to the additional comments on the manuscript. Numbers in brackets at the beginning of our comments indicate line numbers of the revised version of the manuscript with highlighted changes.

General Remarks to the comments of Reviewer #2:

Reviewer 2 has problems with understanding our concept of ENSO asymmetry, which **does not refer to the question whether there are more El Niños than La Niñas**, but their **magnitude in terms of SST anomalies** in the Indian Ocean. A quote from the abstract of our original manuscript (Line 17-19): 'El Niño events have occurred more frequently during recent decades **and** it has been suggested that **an asymmetric ENSO teleconnection (warming during El Niño events is larger than cooling during La Niña events)** caused the pronounced warming of the western Indian Ocean.' We agree that our manuscript requires an unambiguous definition of ENSO asymmetry in the central Indian Ocean and we will provide this in the introduction of a revised version of our manuscript.

We do not aim to reconstruct ENSO frequency with the central Indian Ocean corals, as done in numerous studies with corals from the tropical Pacific (cited as examples by Reviewer 2), where ENSO dominates and causes large SST anomalies that can be unequivocally identified in coral proxy data. ENSO (and IOD)-induced SST anomalies are small in the Indian Ocean (~0.5-0.7°C, see Figure 2 of our manuscript and Roxy et al., 2014; (https://doi.org/10.1175/JCLI-D-14-00471.1)) relative to the background variability (~0.3-0.4°C in the peak ENSO season from December-February; see Figure below) and their identification requires a reference record of past ENSO events. We are aware of the excellent coral reconstructions from the tropical Pacific that record past ENSO events that would be ideal for this purpose.

[Figure]

Map of the Indian Ocean with standard deviation (in °C) of SST anomalies from Dezember to February averaged over the time period 1981-2019.

35   However, to date all these reconstructions are restricted to certain time windows and do not cover the entire time intervals of our central Indian Ocean corals. We therefore used the classical list of ENSO events from Quinn alongside with the updated list of Brönnimann et al., 2007 (https://doi.org/10.1007/s00382-006-0175-z), who evaluated and synthesized a number of ENSO reconstructions (ERSST NINO3 by Smith and Reynolds, 2004 (https://doi.org/10.1175/1520-

40   0442(2004)017<2466:IEROS>2.0.CO;2); Mann NINO3 by Mann et al., 2000 (https://doi.org/10.1175/1087-3562(2000)004<0001:GTPIPC>2.3.CO;2); Cook/D'Arrigo NINO3 by Cook, 2000 (https://www.ncdc.noaa.gov/paleo-search/study/6250) and D'Arrigo et al., 2005 (https://doi.org/10.1029/2004GL022055); Stahle SOI by Stahle et al., 1998 (https://doi.org/10.1175/1520-0477(1998)079<2137:EDROTS>2.0.CO;2); Quinn and Neal, 1992

45   (Quinn, W., & Neal, V., 1992: The historical record of El Niño events. Climate Since AD 1500: 623–48. R. Bradley and P. Jones.)). So, our interpretation does not rely on an outdated version of ENSO events. Rather, by including the original list of Quinn, we aim to evaluate the sensitivity of our analysis to different ENSO reconstructions.

50   The excellent coral IOD reconstruction of Abram et al. (2020) was published after the submission of our manuscript. However, Abram et al. (2020) demonstrate the tight coupling between the IOD and ENSO during the past millennium, lending confidence to our approach.

**Anonymous Reviewer #2**

**Specific comments:**

**RC2 - 1**
**While this study is addressing an important question and producing valuable coral SST**
60   **reconstructions for a location with few such records, this reviewer finds they do not address the research question posed for their study, were there more El Nino events than la Nina in a robust manner. They do look at magnitudes of these events but not the "asymmetry" they discuss in the introduction or that as suggested by Roxy et al. 2014. This should be a straight forward analysis to test this question but the authors use a wide variety of software programs and several data analysis**
65   **methods to try and address this question that leads to confusion and as a whole, misses the point of their analysis. For example, they spend considerable time and present several figures with spectral analysis that look for periodicities/frequencies in their data. Since El Nino and La Nina are opposites phases of the ENSO variability or "periodicity" they are looking for, the spectral analysis tells you nothing about whether or not more El Ninos occurred than La Ninas.**
70   The term 'ENSO asymmetry' is based on the conceptual work of Burgers and Stevenson ('The Normality of ENSO', 1999, GRL, vol 8) and An and Fin ('Nonlinearity and Asymmetry of ENSO', 2004, Journal of Climate, https://doi.org/10.1175/1520-0442(2004)017<2399:NAAOE>2.0.CO;2). ENSO asymmetry refers to the fact that El Niño events are often stronger than La Niña events, as seen in the tropical Pacific. This does not always apply to teleconnected sites. For example, Brönninam et al.
75   (2007) find that 'the responses to El Niño and La Niña are close to symmetric' in Europe during winter and spring.
In the abstract of our original manuscript, we state that: 'El Niño events have occurred more frequently during recent decades **and** it has been suggested that **an asymmetric ENSO teleconnection (warming during El Niño events is larger than cooling during La Niña events)** caused
80   the pronounced warming of the western Indian Ocean.' (Line 17-19)
Based on the comments of reviewer 2, we assume that he believes we aim to address the question **whether or not more El Niños occurred than La Niñas. This is not the point addressed in our manuscript.** We do not want to focus on the frequency of past EN/LN events or to address the question whether or not more El Niño than La Niña events occurred in the central Indian Ocean. The
85   question we address is: do El Niño events warm the Indian Ocean more than La Niña events cool it. However, we agree that the main aim of our study should be expressed more clearly. In the revised version of the manuscript, we try to clarify these points and define what is meant by asymmetric

ENSO teleconnection and how we use this term in our study. We also shortened the sections on the time series analysis, as these were only used to show that ENSO periodicity is observed in the coral records and may distract the reader from our main results. In the main manuscript, we now only show the wavelet coherency plots (Figure 6), as these show the correlation between our coral Sr/Ca records with ENSO as a function of frequency over time.

**RC2 - 2**
**Spectral analysis is suggestive of periodicities similar to ENSO but is NOT conclusive evidence, see Hochman et al. 2019 (doi: 10.1175/jamc-d-18-0331.1) and Liu et al 2007 (doi: 10.1175/2007jtecho511.1). A large anomaly with the width of 2-7 years can be manifested as a significant 2-7 year periodicity in a spectrum leading to the misinterpretation of ENSO periodicity (try for yourself, do a FFT spectrum and wavelet spectrum of the volcanic explosivity index and compare).**

We used different approaches to test if ENSO frequencies are present in the coral time series: Power Spectrum, Singular Spectrum and Wavelet Coherence Analysis. While the power spectra do not provide conclusive evidence of ENSO, the Wavelet Coherence Analysis does: it shows that there is a positive correlation between an ENSO index (we used the Nino3.4 from Wilson et al. 2010; see discussion further below) and the coral time series at interannual periodicities. However, we realize that this analysis is actually more important than the Power Spectra, and exchanged the figure 6 with figure S11 (wavelet coherence analysis).

**RC2 - 3**
**Furthermore, why do breakpoint detrending, removing monthly anomalies, etc. it is not necessary to answer your question.**

For detrending we used published methods by Mudelsee (2000; https://doi.org/10.1016/s0098-3004(99)00141-7) and Mudelsee 2009; https://doi.org/10.1140/epjst/e2009-01089-3). Detrending was necessary to compile the composite records. We then also used this detrended data for the power spectrum analysis. Removing monthly anomalies is a standard procedure to investigate interannual variability.

**RC2 - 4**
**Additionally, using one-tie point per year to build the coral chronology introduces a large amount of uncertainty to your time series, especially in the monthly anomalies that could mask any real signal in time and frequency, see figure 12 of Williams et al. 2014 (http://dx.doi.org/10.1016/j.gca.2014.04.006), and**
**Table 5 in DeLong et al., 2014 (doi:10.1002/2013PA002524). If you are removing the annual cycle from your data, at least two tie points should be used, four is better otherwise your residuals will have a annual cycle still there that introduces spectral noise. There are a considerable number of other studies that look at ENSO variability to ad-dress similar questions.**

We developed the age model following the pioneering work of Charles et al., 1997 (https://science.sciencemag.org/content/277/5328/925), who has proposed to use the month of August as one single anchor point in any given year at the Seychelles, a site located slightly further west than Chagos with a similar monsoon-dominated SST seasonality. Charles et al. have demonstrated that with their approach, monthly anomalies can be computed from coral proxy data, and that these monthly anomalies can be correlated (calibrated, in fact) with instrumental SST anomalies. See Charles et al., 1997, Figure 2 B.

Due to the strong cooling of the western and central Indian Ocean following the onset of the Indian summer monsoon in boreal summer, which is seasonally phase-locked, this age model is very precise (the non-cumulative age model error is +/-1 month in any given year). Each additional anchor point would introduce an additional error which, in the Indian Ocean, tends to be larger during the other seasons of the year.

140    We note that other studies recommended by reviewer 2 as examples also rely on one anchor point per year, e.g. McGregor et al., 2013 (https://doi.org/10.1038/ngeo1936); Hennekam et al., 2018 (https://doi.org/10.1002/2017PA003181).

The approach proposed by the Reviewer (using more anchor points in any given year) would only be applicable at sites that have large-amplitude, sinusoidal seasonal cycles, where age model errors become a problem in the transitional seasons in fall and spring due to the rapid change in SST during

145    a short time period. In fact, the examples cited by the reviewer are from sites with large seasonality, in particular the paper of Williams et al. 2014 (http://dx.doi.org/10.1016/j.gca.2014.04.006), that focuses on red algae from high northern latitudes.

To validate our approach, error estimates based on the standard error where shown in the composites for each mean monthly value.

150

**RC2 - 5**

**Why "reinvent" the data analysis approach? Just use the methods everyone else uses, band pass filter to remove low frequency variability (> 10year) and trends and higher frequency annual cycle, see collective work of Kim Cobb's lab, (Cobb 2003, 2013, Sayani 2019, Grothe 2019 doi:**

155    **10.1029/2019GL083906; Chen et al., 2018 https://agupubs.onlinelibrary.wiley.com/doi/abs/10.1002/2018GL077619,Nurhati et al., 2011 DOI: 10.1175/2011JCLI3852.1) and McGregor 2010 (www.clim-past.net/6/1/2010/ ) not to mention the excellent work by Hereid et al., 2013(doi:10.1130/g33510.1 and doi: 10.1029/2012PA002352) and the new study published by Lawman et al. 2020 doi: 10.1029/2019PA003742 where they use ENSO**

160    **variability via histograms and probability density functions to assess ENSO variability in the past that built upon the work of Emile-Geay et al 2016 where they used probability density to assess ENSO variability in a network of coral and mollusks reconstructions and climate models (DOI: 10.1038/NGEO2608). Furthermore, McGregor et al., used a Cluster Analysis to assess El Nino and La Nina amplitudes in fossil corals (DOI:10.1038/NGEO1936) and they use wavelets to band pass filter**

165    **their coral reconstructions in their 2011 paper (doi:10.1016/j.gca.2011.04.017). The PAST software you are using is capable of doing band pass filters.**

We are aware of the excellent work of Kim Cobb's lab and the many other excellent coral-based ENSO reconstructions from the tropical Pacific. For reconstructing ENSO frequency, we agree that the approaches mentioned above are appropriate. However, as mentioned above, reconstructing ENSO

170    frequencies is not our aim and we therefore did not apply any of these methods. Instead we show with our wavelet coherence analysis results that there is a positive correlation between an ENSO index (we used the Nino3.4 from Wilson et al. 2010 (https://doi.org/10.1002/jqs.1297); regarding this index see further below).

175    **RC2 - 6**

**My second concern is the coral Sr/Ca records in the fossil corals that show large cold anomalies (up to 6∘C?) in Figure 4.The labels in this figure are hard to read but a 4-6∘C anomaly is not expected, even fora La Nina event. The anomaly in Boddam B (1856-1862) spans~6 years and would be manifest in a spectral analysis as a 6-7 periodicity. Look at the Wavelet spectrum for this coral, it will**

180    **show you if this periodicity is center on this anomaly and this would be why you see 6-7 year peak in Figure 6b. Same could be said for Eagle 3 (1890-1894) and the three year peak. Please include the wavelet spectrum from each series in your paper (better than the spectrums you have and more convincing if not driven by these anomalies).**

This anomalously cold peak in Boddam 8 is a relative extreme peak in a phase of generally colder sea

185    surface temperatures. Such decadal variability in the Indian Ocean was already describe in Cole et al., 2000 (https://doi.org/10.1126/science.287.5453), Pfeiffer et al., 2006 and 2009 (https://doi.org/10.1130/g23162a.1; https://doi.org/10.1007/s00531-008-0326-z) and Charles et al., 1997 (https://science.sciencemag.org/content/277/5328/925), and the typical periodicity is 9-13 years. We therefore interpreted this anomaly as one cold event in a decadal cooler phase (note: a 6

190    year cold interval would not result in a 6-7 year, but in a ~12 year periodicity, as the cold interval would only represent one half of a warm-cold cycle). The extreme peak lasts < 1 year and the absolute

value of this cold event is smaller than 4-6°C: the difference between the cold anomaly event and the minimum peak the year before or the year after this cold peak, respectively, is only 2.6-3.1°C. Large short-term cool events are possible as Chagos lies in a region where open ocean upwelling occurs (see Leupold et al., 2019; https://doi.org/10.1029/2018GC007796).

We agree, however, that this figure is too small to be read properly. We therefore added a larger version of this figure to the revised version of the manuscript.

As mentioned above, we exchanged our power spectrum analysis plots with the wavelet coherence analysis plots (Figure S11). The wavelet coherence analysis was performed with the Nino3.4 from Wilson et al., 2010 (https://doi.org/10.1002/jqs.1297) and shows that there are EN/LN signals recorded in our corals from the central Indian Ocean.

**RC2 - 7**

**Back to the cold anomalies. Looking at the x-radiographs: B8 from(1856-1862) appears fuzzy, could this be dissolution or suboptimal alignment of the corallite to the slab surface (see DeLong 2013 doi:10.1016/j.palaeo.2012.08.019)? If you were to resample this time interval to the far right of that slab, is that cold anomaly still there? I would guess not. Coral E3 has the anomaly from 1890-1894 and this is over the core break in the x-ray image. Do these two paths overlap and how well do they agree with each other? The second path is very close the edge of the coral, could there be local diagenesis there? If you were to sample the second core piece just below the first path, is that cold anomaly still present? For Core Eagle5, the mean shift occurs as the sampling paths shifts from the top to bottom piece of the coral. If you sample a different path with optimal corallites, is this shift still there? All this shifts may be real but any large anomalies should be replicated to see if local diagenesis or suboptimal sampling produce the anomaly. I will note: if you use band-pass filters for your ENSO data analysis, these shifts are less meaningful, but you should make sure your coral Sr/Ca is reflecting the SST signal and not something introduced by sampling. Please include a figure of your raw coral Sr/Ca data with paths in depth in your supplemental materials. Additionally, mark where the XRD and SEM samples were removed from eh slab. It is possible to get pockets of diagenesis in small areas of the coral away from where you did the XRD, thin section, and SEM samples. See Quinn 2006 doi:10.1029/2005GL024972; Sayani 2011 doi:10.1016/j.gca.2011.08.026,Hendy 2007 doi:10.1029/2007PA001462).**

The X-ray of Boddam 8 shows traces of saw-cuttings, as the original slab cut in the field was a bit thin for further cutting. We will mention this in the Figure caption of a revised manuscript. This has nothing to do with the preservation of the sample. However, the orientation of the corallites can be seen clearly on the X-ray image. They are always parallel to the slab surface. In fact, the coral shows very even annual growth bands, so irregular growth patterns are not a problem.

All sampling paths were selected so that we get a continuous record for each coral sample. This includes also both sampling paths on coral slab E3. For this sample, there is an overlap of 10 mm, which means 10 subsamples, for each sampling path, i.e. there is around one year of overlap.

The effect of sampling path selection on coral Sr/Ca ratios has been checked systematically in a previous study using modern cores from Chagos (Leupold et al., 2019; https://doi.org/10.1029/2018GC007796; Figure S5 and S6). The reproducibility of Sr/Ca along various sampling paths (in the center and slightly off a major growth axis) is excellent (Figure S5 of Leupold et al., 2019, see below), and while there is some scatter, there is no systematic change of mean Sr/Ca ratios from a major growth axis to the adjacent valley. This may reflect the even growth of the Chagos corals (see Figure S2: E3 and B8 do not show any major growth axis, E5 shows two in the time period from 1890 to 1909, but growth rates are still very similar across the entire sample).

[Figure]

[Figure]

240   *Figure S5. (a) Scan of a modern Chagos coral core top from which subsamples were taken along two parallel sampling paths for Sr/Ca analysis; with results of Sr/Ca analysis (1st sampling path: blue curve; 2nd sampling path: orange curve). (b) Cross plot of both records with regression line (ordinary least squares) and r-squared value indicating a positive correlation. From Leupold et al., 2019.*

245   Regarding possible effects of diageneses: our diagenesis screening revealed that diagenetic modifications to the Sr/Ca record are neglectable. The combination of optical and scanning microscopy and XRD measurements is a well-established method for detecting diagenetic alterations in carbonates by Smodej et al., 2015 (https://doi.org/10.1002/2015GC006009), which has already been applied in several studies, e.g. Deik et al., 2019 (https://doi.org/10.1002/dep2.64),
250   Hallenberger et al., 2019 (https://doi.org/10.1038/s41598-019-54981-7), Pfeiffer et al., 2019 (https://doi.org/10.1029/2019PA003770), Utami & Cahyarini, 2017 (http://journals.itb.ac.id/index.php/jets/article/view/2270), Zinke et al., 2016 (https://doi.org/10.5194/bg-13-5827-2016). The method of Smodej et al. (2015) can identify localized areas of diagenetic calcite and we can therefore also assess potential heterogeneities in
255   preservation. Note that the samples for SEM and XRD measurements were taken from the edges of the coral slab where we expect the 'worst' preservation. From there, also the sub-samples used for U/Th measurements were taken. U/Th is even more sensible to diagenesis than coral Sr/Ca, but our U/Th data shows consistent results with small age errors. We can therefore conclude that even the edges of the coral slabs do not show significant amounts of diagenetic alterations.
260   We included a figure of our raw Sr/Ca data in the revised vision of the supplementary material (Fig. S2). We also provided better indications where XRD and SEM samples were taken from the coral slabs (Fig. S2).
      Please note that decadal-multidecadal temperature variability is common in the tropical Indian Ocean and has been described in numerous studies, e.g. Charles et al., 1997
265   (https://science.sciencemag.org/content/277/5328/925); Cole et al., 2000 (https://doi.org/10.1126/science.287.5453); Pfeiffer et al., 2009 (https://doi.org/10.1007/s00531-008-0326-z); Hennekam et al., 2018 (https://doi.org/10.1002/2017PA003181). The anomalies seen in our Sr/Ca data are in the range of observed temperature variability at Chagos (Pfeiffer et al., 2009) and there is no reason to suspect that they are artefacts from sampling or diagenesis.

270

**RC2 - 8**
**The public comments have already questioned the use of the Maunder Minimum in the title and as a climate interval or temporal marker. The paper makes not connections to solar cycles and ENSO variance in the central Indian Ocean and the coral do not span the entire Maunder Minimum so why**
275   **mention it in the title? I suggest the use of the Little Ice Age in its place, as the records presented are part of this interval and that term is accepted within the climate and paleoclimate literature.**
      We agree and as already mentioned in our reply to this public comment, we used the term Maunder Minimum in a misleading/incorrect way. We adjusted it in the text and also changed the title of the manuscript. However, we used "…since 1675" instead of "Little Ice Age" as suggested by the reviewer,
280   so that everyone is aware of the exact time interval we are focusing on.

**RC2 - 9**

**The authors need to improve their review of coral Sr/Ca reconstructions in the Indian Ocean. While it is true that there are not many records currently published from the region, there**
285 **are more than the authors suggest, seven by my count. Line 38-39 if there are few coral Sr/Ca studies, why not list them all to be comprehensive and not just cite the authors own papers. I count 7 studies so is that really a few? Hennekam 2018 doi: 10.1002/2017PA003181 Zinke 2014 doi: 10.1038/ncomms4607 Zinke 2004 doi:10.1016/j.epsl.2004.09.028 Zinke 2008 doi:10.1029/2008GL035634.**

290 In this paragraph, we wanted to point out, that there are only a few coral climate reconstruction studies exist from the central, tropical Indian Ocean (Maldives and Chagos Archipelago) using high-resolution (biweekly to monthly) Sr/Ca records to reconstruct SST. We agree that it is more than the mentioned study by Pfeiffer et al. (2006). We added references for the other studies.

However, we do not count Hennekam et al. (2018; doi: 10.1002/2017PA003181), Zinke et al. (2014
295 doi: 10.1038/ncomms4607), Zinke et al. (2004, doi:10.1016/j.epsl.2004.09.028) or Zinke et al. (2008 doi:10.1029/2008GL035634) into this category as these studies focused on corals from either the western or eastern Indian Ocean and/or included Sr/Ca of lower resolution. Additionally, from above mentioned Literature, already three studies have been cited in the initially submitted manuscript. However, we missed one study mentioned above. We added this one in the revised version. Besides,
300 we revised this paragraph in general, as parts of it were mentioned twice in the initially submitted manuscript.

Besides, we would like to reject the reproach that we just cite our own papers. Authors of our study, contributed to three studies that we have not cited in the initially submitted manuscript but that were mentioned by the reviewer.

305

**RC2 - 10**

**The introduction section would also benefit from a more in-depth review of the literature on coral-based SST reconstructions of ENSO, both from the Indian Ocean perspective and also the Pacific Ocean. Lawman et al. (2020) in Paleoceanography and Paleoclimatology, McGregor et al. (2019) in**
310 **Nature Geosciences, Grothe et al. (2019) in Geophysical Research Letters, and Tangri et al. (2018) in Paleoceanography and Paleoclimatology would all be useful for comparison, and have data available online. These and other ENSO reconstructions can be used for comparisons between basins back to the1600s.**

We included a more in-depth review of the literature on coral-based SST reconstructions of ENSO in
315 the introduction of the revised version. We welcome coral data from the Pacific as a baseline for ENSO variability for comparison with the central Indian Ocean data (see Pfeiffer et al., 2009 for a comparison of recent data from Chagos and Palmyra). For the extended time period investigated in this study, the coral data available is still spatially and temporally inhomogeneous, so we preferred to use the classical list of ENSO events from Quinn and the updated list of Brönnimann et al., 2007.

320

**RC2 - 11**

**I question the authors' decision to count all positive SST anomalies in their coral records as El Niño events, despite the fact that they acknowledge the existence of warm IOD events occurring independently of ENSO (Section 2.2 Climate, lines 92-93).If the authors are comparing other ENSO**
325 **records to this one, why not remove any positive anomaly events that are unconfirmed by other ENSO records as potential IOD events? Or, why not also compare their record with IOD records? Barring the complete removal of IOD-associated events from the record, I think it would be worthwhile for the authors to compare reconstructions with and without the positive SST anomalies that are not confirmed ENSO events to provide a more complete perspective on potential**
330 **overestimation of El Niño frequency and strength. I also recommend that the authors review recent literature regarding the IOD, including the recently published Abram et al. (2020) Nature article reconstructing the IOD back to the 13th century AD.**

The study by Abram et al. (2020) is indeed a very important one. It states that there are extreme IOD events that occurred independently of ENSO, but also that "a persistent, tight coupling existed between the variability of the IOD and the El Niño/Southern Oscillation during the last millennium." This supports our approach.

In fact, as it can be seen in Table 6 in the manuscript, all positive anomaly events found in the coral records can be explained with El Niño events listed in either Quinn 1993 or Brönnimann et al., 2007 (https://doi.org/10.1007/s00382-006-0175-z). We just wanted to point out, that there is the possibility, that such events can overlap with IOD events or can even occur independently. Abram et al. (2020) named three extreme IOD events (2019, 1961, 1675) that occurred independently of ENSO. However, both in 1675 and 1961 no positive anomaly events can be found in our records.

Furthermore, the main focus of our study was to study the ENSO teleconnection between the Indian Ocean and Pacific Ocean. Unfortunately, the coral time windows of Abram et al. only partly overlap with our data. In a revised version we mention the IOD events that do not coincide with ENSO events as documented in Abram et al., 2020.

**RC2 - 12**

**In section 2.4, "ENSO Indices", the authors list the indices that they use for comparison with their coral records. However, they do not discuss whether these records are coherent, or how they vary, over time. It also appears that they generated their own Niño3.4 anomaly record, which they call an index. From what I understand, the Niño3.4 index only extends back to 1870, using HadISST, not ERSST. While I applaud the authors for applying their own analysis to the data, it is unclear exactly how they calculated their anomaly record from the ERSST data, and as such they need to describe that process in more detail. Do not call the Wilson ENSO reconstruction Nino3.4, that name has already been taken, just call it Wilson ENSO.**

Regarding the introduction of indices, we agree that it might cause confusion as we used two indices named Niño3.4. However, we did not generate our own index. Following the reviewers suggestion, we renamed the Wilson Nino3.4 index. It is now referred to as 'Wilson Niño Index'. Furthermore, we explained in more detail which index we used to show what in section 2.4.

**RC2 - 13**

**Especially questionable is the application of the Quinn 1993 record (Ortlieb 2000 pro-vides an updated version), which is subjective and based on written records, though I understand the authors are limited in the number of records that they can use due to the limited temporal scope of most ENSO records. I'm particularly confused as to why they did not compare some of their 19th century records to the extended multivariate ENSO index (MEI.ext), which spans 1871-2005 (Wolter and Timlin, 2011), or the more recent series of indices published by Sullivan et al. (2016) that include central, eastern, and mixed-type ENSO events back to 1854? Or any of the other ENSO reconstructions on the NOAA paleoclimate website, there are several to choose from (Cobb 2013, McGregor, 2010, Li 2011, Braganza 2009, Cook 2008, Gergis 2009).**

The goal for future coral paleoclimate studies should be to compile a consistent coral data product which overlaps with the entire time period that is studied. However, up to this point there does not exist such a continuous index which overlaps with our records.

We are aware that the Quinn record is based on written records. However, we did not only rely on the Quinn record from 1993. We **compared** Quinn 1993 with the list of ENSO events compiled in Brönnimann et al., 2007 (https://doi.org/10.1007/s00382-006-0175-z). We believe that this gives some indication of the sensitivity of our results with respect to different ENSO reconstructions.

Both records cover all our coral time windows, including our 17th century coral record. We wanted to use as few indices as possible, and the same indices for all coral time windows shown in our study, for consistency. Brönnimann et al. (2007) combined several reconstructed ENSO indices (ERSST NINO3 by Smith and Reynolds, 2004 (https://doi.org/10.1175/1520-0442(2004)017<2466:IEROS>2.0.CO;2); Mann NINO3 by Mann et al., 2000 (https://doi.org/10.1175/1087-3562(2000)004<0001:GTPIPC>2.3.CO;2); Cook/D'Arrigo NINO3 by Cook, 2000 (https://www.ncdc.noaa.gov/paleo-search/study/6250) and D'Arrigo et al., 2005

(https://doi.org/10.1029/2004GL022055); Stahle SOI by Stahle et al., 1998 (https://doi.org/10.1175/1520-0477(1998)079<2137:EDROTS>2.0.CO;2)), climate field reconstructions and early instrumental data and also assessed the data for consistency.

**RC2 - 14**

**At the very least, a comparison between the two main indices used (earlier than table6/section 4.5) would greatly strengthen the authors' conclusions and help the reader understand their criteria surrounding the selection of El Niño events from these records for comparison. The authors cite Wilson et al. (2010), which analyzes the coherence between several ENSO reconstructions extending back to the 17th century, but do not address the paper's conclusion that inter-reconstruction coherence breaks down in the19th century. Thus, using the Wilson et al. (2010) record to identify individual events in the late 17th – early 19th century seems questionable. Labeling this record Niño3.4was also confusing, making it hard to differentiate between the Wilson record and the ERSST-based anomaly record from the Niño3.4 region. This paper has a lot of potential, but needs extensive work. I commend the authors for attempting an in-depth analysis of their data, but encourage them to consider alternative methods for analysis that would be both simpler to accomplish and ultimately more powerful in their application.**

In section 2.4 of the initial submitted manuscript we already introduce all indices we used for this study. However, we agree that we need to provide more detail on which index we used for what so that it does not lead to any confusion. For example, we did not use the index by Wilson et al. (2010) for identifying single ENSO events, as we are aware of the papers conclusion that inter-reconstruction coherence breaks down in the19th century. This is in fact the reason why we decided to use the lists of events from Quinn (1993) and Brönnimann et al., (2007). However, for Wavelet Coherence analysis, we need time series data, and for this purpose, we used the Wilson et al. (2010) record. In a revised version of the manuscript, we placed more emphasis on explaining and comparing the ENSO indices in section 2.4. We shortened the manuscript by omitting unnecessary interpretations on spectral analysis, and table 6/ section 4.5 are more central in the revised version.

**Technical corrections:**

**Figure 6 The authors do not standardize their spectra in time, so that it becomes difficult to interpret the individual plots of Figure 6. Most of the plots are based on monthly resolved data with frequency as cycles/month, except for 6e which is based on annually resolved data and cycles/year and is thus shifted in frequency space.**

We did not standardize these plots because they do not have to be compared with each other. Every sub-figure is there to show ENSO periodicities and each resolution is mentioned in the figure caption. However, we decided to exchange this figure with the figure showing the Wavelet Coherence Analysis anyway (see above).

**In section 3.1 "Coral collection and preparation" more information about the x-ray system used and the settings applied in the generation of the x-radiographs would be helpful for replication or reproduction by later studies. Are these x-ray positive or negative images? It would also be useful to know how the coral collected from the derelict building arrived there – was it via human activity or storm or tsunami deposited? This is not necessary for publication, but could help guide the location and collection of other specimens.**

We provided the additional information about the x-ray system in the revised version of the manuscript.

The derelict buildings were indeed built by humans living on the islands. However, it is not known whether they found their material as boulders on the beach or if they quarried them on the island to get their building material. There are no written records from Chagos from this time. As there can be found hundreds of boulders at the beaches of Chagos nowadays (see Figure S1a) it is likely that the Chagossians first used material they found at the beaches close to their Colony to build their buildings.

This is also suggested by the shape of some of the corals found in the walls, and the ages of the corals obtained by U/Th dating.

440

**In section 3.2 "Coral Sr/Ca analysis" was just one standard or known value used in the ICP analysis? Most labs use 2 or 3 (a gravimetric, a coral, and JCP international standard). The Schrag (1999) and de Villiers et al. (2002) methods bracket each sample for drift correction. which is typical for ICP-OES whereas every 5th sample is used for ICP-MS since that instrument does not drift as much. The exact**

445 **analytical precision(s)±1sigma should be given with # of measurements and error bars of analytical precision on all graphs with coral Sr/Ca. It would also be good to see the raw Sr/Ca values plotted, not just anomalies. It is difficult to gauge the individual records from the anomaly plots alone.**

In total, 5 different CRM were used, including the international CRM JCp-1 and JCt-1. They were measured before and after the entire measurement sequence. We will explain this in more detail in

450 the methods section in a revised version. Furthermore, we added error bars on all coral Sr/Ca graphs. During method development for Sr/Ca analysis we started off with standard-sample-standard bracketing as in Schrag et al. (1999) but found that inserting 6 samples did not compromise our results at all. (We re-measure every 12$^{th}$ coral sample at the end of each measurement run, and we find no evidence of drift problems). A similar strategy is also used in isotope geochemistry. The

455 resulting uncertainty of 0.8 permil (1SD) in our data is speaking for itself and is much better than all ICP-MS data we know of. The very general statement of reviewer 2 that ICP-OES instruments drift more than ICP-MS instruments is not valid, at least for our instrument.

**In section 3.3 "Chronology" the authors suggest that they only use the minima of seasonal SST cycles**

460 **as their chronological tie points, but their chronology would likely be more robust if they used at least 2 ties points (maxima and minima) for time assignment.**

See our comment below reviewer comment RC2 - 4. Note: at Chagos there are two SST maxima per season, one in boreal fall and one in spring. In most years, the spring maximum is largest, but this is not always the case (see Figure 3, year 2007-2009). Adding a second tie point in boreal summer

465 therefore adds a lot of chronological uncertainty.

**In section 3.5 "Statistics", it would be helpful to know which version of PAST (with citation) and MATLAB the authors used. I am confused as to why the authors chose to use the web application T-Test Calculator (web link needs to be given) rather than at-test function in the other software listed**

470 **or just use a t-table in a statistics textbook. Also, in general, the authors tend not to list the α, n, or other key statistical values for their data throughout the paper (except in some figures). All averages should be report with their standard deviations, and number of values, correlations should have p-value and n, and all errors as either 1 or 2 sigma, which are standard statistical practices.**

We agree, that these information should be added and we did it in the revised version.

475

**In section 4.4, "ENSO Interannual SST variability", the authors suggest that all of their coral records show statistically robust typical ENSO periodicities (3-8 years), but fail to address varying levels of statistical robustness. Their earliest composite record (E5,Figure 6a) for example has an ENSO periodicity that is only statistically significant at the α=0.1 level, but the authors do not discuss this**

480 **in the text. Despite detrending before analysis, there is also evidence of roughly annual periodicities in both B8 (Figure6b) and E3 (Figure 6c). Figures S8-10 and S11, supplementary analyses, are cited as confirming the power spectrum analysis results, but also bring out issues in the temporal continuity of these spectra and their directionality.**

Annual periodicities are still visible, because only the long-term trend was subtracted and not the

485 annual cycle. However, as mentioned above, we exchanged this figure with figure S11 and show the wavelet coherence analysis plots instead.

**The Brönnimann et al. paper was published in 2007, not 2006 (this issue could be present in other references, and should be checked).**

490 Reference corrected.

**The GIM coral data seem to have been first published in Pfeiffer et al. 2009, not 2017.**

Yes, that is correct. That is why we mentioned Pfeiffer et al. (2009) in the initially submitted manuscript in the Methods and Material section (chapter 3.1; lines 122-127) as well as in figure caption of figure 6. Note that Pfeiffer et al. (2009) only present data back until 1950, not 1880.

**All of the supplemental figures are mislabeled, and should be corrected. I recommend, in fact, that the entire Supplemental file be carefully reviewed and edited, as I noticed consistent issues in the labeling of materials and numerous typographical errors.**

We do not see that all supplemental figures are mislabeled. However, as the other reviewer also mentioned wrong numbering of the supplementary figures, we changed the titles of each chapter in the supplementary material which may have caused the misunderstanding. In the originally submitted version, e.g. chapter 3 of the supplementary material was named 'S3 X-ray images' that included 'Figure S2'. After revision, this chapter's title is now '3 X-ray images'.

Additionally, we went through the entire supplementary file and checked for consistency.

**Ln 30: The opening sentence of the introduction reads a bit awkwardly, I would suggest rewording to something like "As the impacts of global climate change increase, paleoclimate research is more important than ever". On the same line, I would remove the first word of the second sentence ("Especially") and simply begin the sentence with "The Indian Ocean...".**

(l. 30) Revised.

**Ln 32: should be "basin", not "basing".**

(l. 31) Changed.

**Ln 34: Remove "As" and begin the sentence with "Tropical corals".**

(l. 34) Done.

**Ln 35: the sentence here continued from**

(l. 35) Done.

**Ln 34 is somewhat awkwardly worded, and should be ended with "variability" not "variabilities".**

(l. 35) Done.

**Ln 37-39: the sentence in this section repeats its point in the second half, I would delete the section after the first citation of Pfeiffer et al. 2006.**

(l. 37) Deleted.

**Ln 40: change "are focusing on" to "focus on".**

(l. 39) Done.

**Ln 41-42: change "lack of data in" to "lack of data from", remove "still" and "the" from the phrase "still limits the" and replace "the" with "our", and change "variabilities" to the singular.**

(l. 41) Done.

**Ln 46: change "In fact, it was suggested" to "It is suggested".**

(l. 57) Revised paragraph.

**Ln 68: change "form" to "from". Also recommend moving the phrase "from October to April" from beginning to end of sentence.**

(l. 85) Done.

**Ln 166:"Composite" should be "Composites".**

(l. 205) Revised.

545

**Ln 200 and 203: ranges in both of these lines contain values to three significant digits, while all others reported in paper are only to two.**
(l. 235) Adjusted.

550 **Ln 293: there is a period missing between "Indian Ocean" and "For".**
(l. 341) Inserted.

**Ln 308: the end of the sentence here should read "Brönnimann et al. (2006) (Table 6)".**
(l. 275) Revised.

555

**Ln 313: should read "Indian" not "India" monsoon.**
Deleted entire paragraph.

**Ln 337: remove the "events" before "non-La Niña", and make sure to correct the spelling of La**
560 **Niña.**
(l. 393) Done.

**Additional Comments in Manuscript PDF (initial text of the manuscript with line numbers and in italic, reviewer comments in bold with our answer below each comment):**

565 *L 24 "All four coral records show typical ENSO periodicities, suggesting that the ENSO-SST teleconnection in the central Indian Ocean was stationary since the 17th century"*
**Comment 1: Abrams had a recent paper, what do they say?**
For a detailed reply, see our comment below the reviewer comment RC2 – 11.

*L 37: "There are only some studies including Sr/Ca measurements for SST reconstructions (e.g. Pfeiffer*
570 *et al., 2006),…"*
**Comment 2: if there are few, why not list them to be comprehensive and not just cite the authors own papers. I count 7 studies so is that really a few? Hennekam 2018 doi: 10.1002/2017PA003181 Zinke 2014 doi: 10.1038/ncomms4607 Zinke 2004 doi: 10.1016/j.epsl.2004.09.028Zinke 2008 doi:10.1029/2008GL035634.Zinke 2016 doi: 10.5194/bg-13-5827-2016 Bryan 2016 doi: 10.5194/bg-**
575 **13-5827-2016Abram 2020 https://doi.org/10.1038/s41586-020-2084-4**
(l. 37-41) We added missing literature mentioned above. For a detailed reply see our comment below reviewer comment RC2-9.

*L 40: "…and/or are sampled at only bimonthly (Zinke et al., 2004; Zinke et al., 2008) or annual resolution*
580 *(Zinke et al., 2014; Zinke et al, 2015)"*
**Comment 3: Bimonthly meaning every two months or sampled twice per month?**
**Regardless, bimonthly is probably fine for resolving the seasonal cycle, just as well as, monthly. So what is the point you are trying to make here?**
(l. 40) Bimonthly means in this case every two months. For resolving the seasonal cycle, it is of course
585 better to have 12 values per year than 6 values.

*L 45: "Strong El Niño Southern Oscillation (ENSO) events occur more frequently since the early 1980s (Baker et al., 2008; Sagar et al., 2016)…"*
**Comment 4: El Nino are the events, SOI is the cycle between SLP between Darwin and Tahiti linked**
590 **to walker circulations and includes EL nino and la nina events.**
**REvise to El Nino events only.**
(l. 44) Revised.

*L 46: "…demonstrating an existing stable SST-ENSO teleconnection between the Pacific Ocean and*
595    *Indian Ocean…"*
**Comment 5: How do you know this is stable? The premise of your paper is to assess if it weakens or exists in the past 200 years. Delete "stable"**
      (l. 52) With "stable" we meant "stationary". We see that we did not explain it very well. We defined what we mean with "stationary" in the revised version.

600

*L 49: "This asymmetric ENSO teleconnection has been suggested to contribute to the overall*
*50 warming of the tropical Indian Ocean."*
**Comment 6: The use of "asymmetrical" was confusing from the abstract to here. At first I thought you were referring to a spatial asymmetry but you mean a temporal or different response to La Nina-**
605    **El Nino events. Scientist talk a lot about the ENSO spatial pattern so this is easy misinterpretation. Why not use a better term? Yes, Roxy 2014 use the "asymmetry' term  but their paper is confusing as well. Help the reader out and explain better what is meant by asymmetrical ENSO teleconnection between Indian and Pacific oceans.**
      (l. 67) We agree, that we had to explain better what we mean with "asymmetrical ENSO
610    teleconnection". We did this in the revised version.

*L 53: "…the core top (1950-1995) was shown to record SST variability at Chagos on grid-SST scale*
*(Pfeiffer et al., 2009)."*
**Comment 7: Explain what grid-SST scales are? Do you mean a particular gridded SST data product(s)?**
615    Pfeiffer et al. used ERSST version 2, but that data was consistent with other SST products such as HadISST.

*L 55: "We identify past warm and cold events in each record and use these events to compile*
*composites to evaluate the symmetry of positive and negative ENSO-driven SST anomaly events in the*
620    *tropical Indian Ocean."*
**Comment 8: By Symmetry you mean the magnitude of the La nina nad El nino events are the same or not. Why not just say you are looking at magnitude differeneces?**
      (l. 70) As mentioned above, we explained what we mean with "ENSO asymmetry" in the revised version. The concept is widely used in conceptual papers on ENSO and ENSO teleconnections

625

**Comment 9: Roxy 2014 Fig 5 shows Western Indian Ocean has most of this warmer El nino events, not the central Indian ocean.**
      We agree, that the western Indian Ocean is most affected by warming as shown in Roxy et al., 2014 using HadISST data. However, this warming trend is still visible in the Seychelles-Chagos-Thermocline-
630    Ridge region, which also suffers from a lack of observations on historical timescales (see Pfeiffer et al., 2017; https://doi.org/10.1038/s41598-017-14352-6). Furthermore, depending on the SST dataset, warming in the Indian Ocean is largest in the Arabian Sea (Roxy et al., 2014) or in the central Indian Ocean                    (Roxy                    et                    al.,                    2020;
      http://www.rocksea.org/bin/research/roxy_indian_ocean_warming_climate_change_assessment_
635    2020.pdf). Besides, in our conclusions we suggest compiling composite records of negative and positive SST anomaly events from sub-fossil corals from the western Indian Ocean to further test Roxy et al.'s hypothesis of an asymmetric ENSO teleconnection in the western Indian Ocean.

*L 59: "The Chagos Archipelago is located in the tropical Indian Ocean, about 500 km south of the*
640    *Maldives."*
**Comment 10: can you provide a latitude and longitude to be more specific?**
      (l. 76) Done.

*L 90: "A coral-based reconstruction of past IOD events extends until 1846 and suggests a recent*
645    *intensification of the IOD (Abram et al., 2008)."*

**Comment 11: Should add more recent paper by Abram 2002. https://doi.org/10.1038/s41586-020-2084-4**

(l. 107) Added. For a detailed reply, see our comment below the reviewer comment RC2 – 11.

*L 101: "Both anomaly records are not significantly different (t-value = 0.34; p-value = 0.37)."*
**Comment 12: 2011 la nina has different magnitudes.**

(l. 120) We are not sure what the reviewer wants to point at with this comment.

*L 102: "This suggests that the magnitudes of ENSO signals at Chagos should be recorded in all coral records analyzed in this study, as it is independent from the reef setting."*
**Comment 13: You cannot say this for all time I would drop the last part of this sentence.**

(l. 121) Revised.

*L 154: "…measured in 2017 in the HISPEC laboratory of the Department of Geosciences, NTU, following techniques described in Shen et al. (2012). These age determinations are consistent with our Sr/Ca chronologies."*
**Comment 14: what about dating uncertainties? were U-th a single annual band? how and where were these taken, please mark x-ray images to dating samples.**

We marked where the U/Th samples were taken on the X-ray images (Figure S2). The age model was developed in the following way: For each coral sample: 1st age dated by U/Th (in 2016) → from this age band the years were counted on the x-ray images (and combined with raw Sr/Ca) → upper or lower most counted year was compared with 2nd age dated by U/Th (in a second measurement run in 2017). As every second age that was dated with the second U/Th measurement fitted to the age model developed using the x-rays and raw Sr/Ca data, dating uncertainties due to sampling for U/Th measurements are neglectable.

*L 160: "…analysis were selected from all corals based on the X-ray images."*
**Comment 15: please mark x-ray where these samples were located? were they along your sampling path?**

Figure revised (Fig. S2).

*L 170: "Power spectra analysis was performed twice using the open source software PAST (Hammer et al., 2001)"*
**Comment 16: what windows, smoothing, etc. please provide more info.**
**Why use these software programs? did you do more than remove a linear trend? What do these programs do? Why not just filter like cobb 2013 and many others who look at coral and ENSO?**

We moved these parts of the methods to the Supplementary Material document as it describes methods used for analysis that was also moved to the Supplementary Material. There, we added additional information of the software PAST. PAST (Paleontological Statistics) is an open source software with a lot of different functions. We used it for Power Spectrum analysis (REDFIT function, Welch window, Oversample: 2-8, Segments: 4-6). As mentioned in the methods section, we did not use PAST, but *breakfit* and *rampfit* for detrending. Detrending was necessary to compile the composite records. We then also used this detrended data for the power spectrum analysis. These programs are able to remove not only linear trends (which was the case with E3 and GIM as it can be seen in Figure S7 of the revised version of the supplementary material), but they find breakpoints in time series where long-term trends change and calculate linear functions for these periods which are used for subtracting these long-term trends from the original time series. For example, coral record E5 in Figure S7 shows 4 linear graphs in red that were calculated with these programs overlying the original time series.

*L 175: "…and wavelet coherence plots were generated using the MATLAB software toolboxes."*
**Comment 17: citation**

(l. 202) Added.

700   *L 178: SE equation*
      **Comment 18**: **this is for a mean, not all errors.**
        (l. 212) Revised.

      *L 212: "The modern and the sub-fossil coral SST records were compared with the annually resolved El*
705   *Niño index Niño3.4 that extends back until 1607 (Wilson et al., 2010)…"*
      **Comment 19**: **Nino3.4 already has been used and defined in the literature, why not just call it**
      **Wilson Nino to make if clear which data set you are referring to.**
        (l. 128) We renamed the *Niño3.4* index in the revised version. It is now referred to as 'Wilson Niño
        Index'.
710
      *L 223: "All coral records show variations in the frequency of ENSO events (Figs. 7-9 and Tables 4-6)."*
      **Comment 20**: **reference figures before explaining them.**
        (l. 269) We revised this paragraph to shorten the interpretation on ENSO frequency and to make
        our focus more central.
715
      *L 610: Figure 3*
      **Comment 21**: **Y-axis should be labled SST anomalies**
        (l. 674) Done.

720   *L 615: Figure 4*
      **Comment 22**: **Cannot read this figures, lines and text to small. Fix this. Middle figure is missing a y-**
      **axis.**
      **what slope did you use?**
      **Why not just band pass filter?**
725     (l. 680) We included a larger version of this figure in the revised version. We used the slope -0.06
        mmol/mol per 1°C as described in l. 261 of the initially submitted manuscript. However, we moved
        this paragraph up to the methods section of the revised version.
        A detailed calibration for modern corals from the same site (Chagos Archipelago) was presented in
        Leupold et al., 2019 (https://doi.org/10.1029/2018GC007796). In this study, the regression of coral
730     Sr/Ca with satellite data indicates a significant correlation (r-squared: 0.62, p<<0.01, n=265).
        We are not sure why the reviewer is suggesting using band pass filter related to what is shown in
        Figure 4.

      *L 625: Figure 6*
735   **Comment 23**: **Do all in years, not months! make log log plots. This is confusing since you do not**
      **have units on the frequency. looks like a-d are in months and not years. Put units on all graphs.**
      **time interval for ERSST.**
        See our comment below the first technical comment by the reviewer regarding Figure 6.

740   *L S109: Figure S11*
      **Comment 24**: **How do you do this? Nino 3.4 is modern SST.**
        (l. 128) We agree, that we used the term Nino3.4 in a way that led to confusions. In this case, wavelet
        coherence analysis was performed for each coral Sr/Ca record with the Nino3.4 index by Wilson et
        al., 2010. This index extends beyond the instrumental period, until 1607. We changed the name of
745     this index. It is now referred to as 'Wilson Niño Index'.

[revised manuscript text omitted]

5   giving the years of El Niño/La Niña events used for the composite maps (Table S1) and the linear regression results between the coral SST records and the Wilson Niño Index (Table S2) are also part of this supplementary material.

**1 Supplementary Methods**

10  **Indices**

We use Niño 3.4 SST anomalies taken from NOAA ERSSTv5 (Huang et al. 2017) for power spectrum analysis (Fig. S11). These have been interpolated from sparse observational data and extend back until 1870.

**Statistics**

15  Power spectra analysis was performed twice using the spectral analysis function REDFIT (Welch window) of the open source software *PAST* (version 3.25; Hammer et al., 2001). One run was performed with the time series before detrending them, one run after detrending. Every time series was detrended using the softwares *breakfit* (Mudelsee, 2009) or *rampfit* (Mudelsee, 2000), respectively (Fig. S7).

Singular spectrum analysis (SSA) (Vautard and Ghil, 1989) were generated using the *MATLAB* (version R2019b) software
20  toolboxes by Groth and Ghil (2015).

**2 Table with years of events that were used in the composite maps (Fig. 2 in main document)**

| Event years | |
| --- | --- |
| El Niño | La Niña |
| 1982/83 | 1984/85 |
| 1986/87 | 1988/89 |
| 1987/88 | 1995/95 |
| 1991/92 | 1998/99 |
| 1994/95 | 1999/00 |
| 1997/98 | 2007/08 |
| 2002/03 | 2010/11 |
| 2009/10 | 2011/12 |
| 2015/16 | |

**Table S1: El Niño and La Niña event years used for the composite maps. Between 1982 and 2016, 9 El Niño events and 8 La Niña events occurred. Temperature anomalies from December to February were averaged for each event.**

**3 Pictures of coral sample sites**

[Figure]

**Figure S1:** Pictures of coral sample sites. (a) Boulder beach at Eagle Island where the samples E5 (1675-1716; blue arrow) and E3 (1870-1909; red arrow) were collected. (b) and (c) a derelict building at Boddam Island from which the sample B8 (1836-1867) was collected.

**4 X-ray images**

[Figure]

**Figure S2**: X-Ray images of coral samples analyzed in this study with raw Sr/Ca data (dark blue lines). Age models were interpreted using two U/Th measurements from each sample (sampling points for U/Th dating are indicated with circled numbers and light red-shaded areas; for determined ages see Table 1). Red lines indicate subsampling paths. Blue-shaded areas indicate sampling locations for subsamples used for Scanning Electron Microscopy (SEM). Please note that the slab of sample B8 is uneven as the slab was too brittle to polish out saw cuttings from field work and these are still seen on the x-ray image. Note the even growth patterns of all samples.

 **5 Thin section and SEM analysis images**

[Figure]

**Figure S3: Photomicrographs of coral sample E5 (1675-1716). Double arrows indicate corresponding photomicrographs. (a) PPL and (b) XPL overview photomicrograph of the coral skeleton. (c) PPL and (d) XPL photomicrograph of higher resolution where minor amounts of secondary calcite cement is visible (red circle). (e) SEM overview and (f) SEM detail image of (e) where only trace amounts of sugary cements can be found.**

[Figure]

**Figure S4: Photomicrographs of coral sample B8 (1836-1867). Double arrows indicate corresponding photomicrographs. (a) PPL and (b) XPL overview photomicrograph of the coral skeleton. (c) PPL and (d) XPL photomicrograph of higher resolution where small amounts of secondary aragonite cement is visible (red oval). (e) SEM overview image and (f) SEM detail image of B8 (1836-1867). Small amounts of sugary aragonitic cement can be seen.**

[Figure]

**Figure S5: Photomicrographs of coral sample E3 (1870-1909). Double arrows indicate corresponding photomicrographs. (a) PPL and (b) XPL overview photomicrograph of the coral skeleton. (c) PPL photomicrograph of higher resolution where small fragments of aragonite are found (red circle). (d) PPL microphotograph of higher resolution with no signs of diagenesis. SEM images showing (e) microborings (red arrows) and (f) areas which appear brighter due to dissolution.**

**6 Seasonal cycles inferred from Singular Spectrum Analysis**

60   Singular spectrum analysis (SSA) of the coral records with seasonal cycles reveal large interannual to decadal SST variabilities during both the 17-18[th] century and 19-20th century (not shown). The reconstructed components 2 and 3 (RC2, RC3) produced by SSA describe seasonal amplitudes for all samples and explain 28% (E5), 26% (B8), 32% (E3) of the coral Sr/Ca-SST variance. Decadal variabilities are larger during the 17-18[th] century compared to the 19-20th century. The first reconstructed component (RC1) of E5 with seasonal cycles explains 48% of the coral Sr/Ca-SST variance and describes a periodicity of

65   around 18 years. The coral records covering the 19-20th century do not show a strong decadal component in SSA. Instead, RC1 explains 49% (B8) and 39% (E3) of the coral Sr/Ca-SST variance and describes a periodicity of around 7 years, which can be interpreted as ENSO periodicity.

The SSA results were validated by power spectrum analysis of bimonthly coral SST anomalies, which were not detrended (Fig. S7). For this analysis, the coral record GIM (Pfeiffer et al., 2017) was included, which extends from 1880-1995. Power

[revised manuscript text omitted]

---

## Author Response (AR2)

Dear Luc Beaufort,

We are grateful for the feedback provided by the anonymous reviewer and for your decision letter. Below, we address all items raised by the reviewer in detail. We added the wavelet spectrums of all coral records to the supplementary material as suggested by the reviewer and explained why we still prefer showing the wavelet coherence plots in the main document of the manuscript. Furthermore, the manuscript was checked for grammar issues and unclear sentences and for consistent use of the tense.
In the following, we will repeat the reviewer's statement (in normal font) and our reply to it (in italic font). Numbers in brackets at the beginning of our comments indicate line numbers of the revised version of the manuscript with highlighted changes.

Yours sincerely, Maike Leupold

**Editor Decision: Publish subject to minor revisions (review by editor)** (12 Oct 2020) by Luc Beaufort
Comments to the Author:
Dear Maike Leupold and co-ahthors,
Your manuscript has been reviewed a second time by the most critical reviewer of the original submission. He/she is now satisfied with the revisions you made to the original manuscript. Some points need to be corrected and I suggest that you do so. I will check your corrections myself. Also, the reviewer is quite critical of the quality of the syntax, and asks you to have the article corrected by someone qualified. I would appreciate it if you would follow his/her advice.
Yours sincerely, Luc Beaufort

**Report Anonymous Referee #2:**
This revised paper is improved and many details that were unclear in the previous revision are now improved. The detailed analysis of ENSO events with two different reference data sources highlights the discrepancies that any reconstruction will have but still supports the results of the current study. The discussion section begins more like a conclusion section. The discussion should be a comparison to other work, discussion of the strength and weakness of your results, and other interpretations, and implications of the study.
  *We agree with the reviewer that the discussion section began like a conclusion. We revised the entire section.*

This paper still has many grammar issues and unclear sentences. I suggest the authors get help to address these issues. I do not take the time to note all these issues in my detailed edits since there are many and this is a second review. I tried to focus on the science. But this paper needs to have the grammar addressed before publication. I use the free website Grammarly.com often and recommend my students use it as well.
  *The manuscript was checked for grammar and syntax issues.*

**Science items to address:**
The wavelet coherence plots are interesting, but the authors do not take chronology error into account. Assuming the Wilson Nino reconstruction does not have chronology errors, if you shift the coral chronologies within the U-Th dating uncertainties, do the wavelet coherence plots

change or find better coherence? With U-Th errors or ±2-3 years, 2d, you could shift by 6 years to see if the results are better or worse. These plots as they as presented are not showing the level of coherence I would expect if there was a strong ENSO signal in these records. I would like to see the Wavelet spectrums of the coral records to see what the ENSO periodicities look like, I would buy that more than the coherence with two reconstructions that both have chronology errors that could greatly impact these results, see Comboul 2014, doi:10.5194/cp-10-825-2014.

*This comment is somewhat contradictory to an earlier comment from anonymous reviewer 2 (see our previous response letter):*

**„Spectral analysis is suggestive of periodicities similar to ENSO but is NOT conclusive evidence, see Hochman et al. 2019 (doi: 10.1175/jamc-d-18-0331.1) and Liu et al 2007 (doi: 10.1175/2007jtecho511.1). A large anomaly with the width of 2-7 years can be manifested as a significant 2-7 year periodicity in a spectrum leading to the misinterpretation of ENSO periodicity (try for yourself, do a FFT spectrum and wavelet spectrum of the volcanic explosivity index and compare).“**

*Because of this comment, we decided to include the wavelet coherency plots in the main text of our revised manuscript.*

*'Normal' Wavelet spectrums are – just like other methods of spectral analysis – suggestive of periodicities but NOT conclusive evidence, although they do show interannual variability in time/frequency space.*

*We therefore used wavelet coherence analysis to detect common (coherent) time-localized oscillations in the Wilson Nino Index and the Chagos coral Sr/Ca records. We view one time series (the Wilson Nino Index which is based on a multiproxy reconstruction) as influencing the other (Chagos SST, inferred from coral Sr/Ca). This means we can use the phase of the wavelet cross-spectrum to identify the relative lag between the two time series. This lag is mentioned in the manuscript text, and we suggest that it reflects the age uncertainty of the sub-fossil Chagos corals. [Note that Indian Ocean warming (cooling) appears during peak ENSO warming (cooling) in December-February and decays in the following boreal spring].*

*Comboul et al. (2014) assess time-variant age model errors that may result from the miscounting of annual density bands/seasonal coral Sr/Ca cycles and how these could influence the spectral characteristics of proxy time series. (Wavelet coherency analysis is not discussed in this paper). The authors show that these age model uncertainties may reduce the coherency between two time series (but they do not inflate it). While we cannot rule this out, we find significant coherencies between the Wilson Nino Index and our coral Sr/Ca records.*

*We show the raw Sr/Ca data together with the X-ray images in the supplements so that our age models can be assessed.*

*Furthermore, the GIM core (and two other modern replication cores from Chagos) which derive from absolutely dated modern corals capture ENSO variability in the central Indian Ocean (e.g. Pfeiffer et al., 2017, [https://doi.org/10.1038/s41598-017-14352-6](https://doi.org/10.1038/s41598-017-14352-6); Pfeiffer et al., 2009, [https://doi.org/10.1007/s00531-008-0326-z](https://doi.org/10.1007/s00531-008-0326-z); Pfeiffer et al., 2006, [https://doi.org/10.1130/g23162a.1](https://doi.org/10.1130/g23162a.1)). In fact, the corals show a significant linear correlation with Nino3.4 SST and the Palmyra coral $\delta^{18}O$ record published by Kim Cobb (Pfeiffer et al., 2009, [https://doi.org/10.1007/s00531-008-0326-z](https://doi.org/10.1007/s00531-008-0326-z)).*

*However, to address this scientific item, we added 'normal' wavelet power spectra of all coral records analyzed in our study to the supplementary material (Figure S7). All coral time series show significant interannual variability in the ENSO frequency band.*

**Specific items to address:**

What the tense in your paragraph and sections and make sure the tense is consistent. There are still many grammar errors and typos in this paper that need to be corrected before publication.

Line 22 By one sample and two samples, do you mean coral or one measurement for this entire interval. Just say one coral and two corals.

*(l. 22) We mean coral sample(s) and revised the sentence.*

Figure 1 What are numbers 640, 645, 650, 655 for on the right side of the map? Put the latitude and longitude degrees outside the box since the inset is covering part of the or move latitude to the right side of the map. It would be helpful if you map the mean SST of SST anomalies on this map to show the difference note in section 2.3.

*(l. 637) The numbers are the line numbers of the document. This probably happened during conversion from Word file to PDF file. We revised Figure 1. Latitudes now appear on the right side of the map. We do not display map mean SST on this map, because it is the seasonal amplitude that makes the difference between both settings and not the mean SST (see our reply to the reviewer's comment "line 113-114" in this document).*

Figure 2 The "Red" box for La Nina looks purple to me.

*(l. 643) It is the same color as for the box for El Nino. It probably appears different due to the surrounding color of the SST map. Zooming into the boxes confirms that they are actually of the same color.*

Figure 3 The blue line looks black to me.

*(l. 653) It is dark blue. We added this detail to the subtitle.*

Line 38 and 42 and elsewhere Comma is not needed after central "central tropical Indian Ocean".

*(l. 41 + 43) Revised.*

Line 44 The last phase of the sentence is confusing "…as these are phased-locked to the seasonal cycle and vary with the season" What does "these" refer to? The coral Sr/Ca, the ocean, climate phenomena, or something else. Perhaps clarify what "its" is in the same sentence. Revise to make the meaning clearer.

*(l. 44) In this sentence, "these" refers to climate phenomena. We revised this sentence.*

Line 46 Explain what you mean by "ENSO is centered". Is this spatially centered? I think you mean "where ENSO occurs". Same for Line 49. The central part of the tropical Pacific Ocean is not where ENSO occurs, ENSO occurs across the tropical Pacific Ocean — East, Central, and West. The central Pacific has the weakest climate response compared to the east and west Pacific. Additionally, there are different favors of ENSO, a central and eastern ENSO as well as a coastal. Therefore, using "centered" is confusing.

*(l. 48) We meant "centered" in terms of "mainly affecting". We revised it.*

Line 46 Revise "Strong events associated with ENSO have occurred more frequently since the early 1980s relative…"

*(l. 48) Revised.*

Line 48 and elsewhere - Do not use a "/" to replace the word "and". This is a non-standard replacement, reserve "/" to mean "divide by" or to indicate a ratio like "Sr/Ca". This is an informal usage.

*We exchanged every "/" used between "El Niño" and "La Niña" by an "and".*

Line 50 Use the adjective form "…oceanic-atmospheric parameters of the Indian Ocean…" Revise the "which" to a "that". The same sentence, use the same tense for the verbs.
  *(l. 52) Revised.*

Line 52-53 Another confusing sentence, what is demonstrating? Revise sentence, a conjunction is needed. "Strong El Niño/La Niña events influence the tropical Indian Ocean thus establishing a SST-ENSO teleconnection between the Pacific and the Indian Ocean." How you do know the teleconnection is "stationary" and for what time interval? Is this a question you can answer with your study or if others have shown this then say "previous studies have established a stationary SST-ENSO teleconnection…".
  *(l. 54-56) We revised this sentence. We quoted studies stating that the teleconnection was stationary. We consider it to be clear and in agreement with the overall citation style of this manuscript that the references provided in brackets at the end of the sentence refer to the sentence's statement. Therefore, we do not see any reason to revise the sentence with respect to this point.*

Line 49 "While" is the incorrect word, you do not mean "at the same time as. If you mean to highlight contrasting relationship use "whereas".
  *(l. 57) We exchanged "while" by "whereas".*

Line 64 Add for clarification the time interval you are referring to for "Indian Ocean warming". Do you mean for the Little Ice Age or just the 20th century?
  *(l. 67) We meant warming during the 20$^{th}$ century and revised the sentence.*

Line 64 Revise "We develop coral Sr/Ca records… to reconstruct past SST variability."
  *(l. 67) Revised.*

Line 99 Revise this confusing incomplete sentence.
  *(l. 100-101) Revised.*

Line 108 What corals are you referring to? The previous sentence refers to two studies. Perhaps you mean "Those corals reconstructions revealed a few strong IOD events…" After reading the following sentence, perhaps you are referring to your own reconstruction that has not been presented yet.
  *(l. 108) We refer to corals of the studies by Abram et al. And we are not referring to our own reconstructions here. We revised the sentence.*

Line 109-112 "However, neither in 1675 nor in 1961 a positive anomaly can be found in our coral SST records." This is a result and the following sentences are how you intend to interpret your results, it does not belong in the introduction, but in the methods or results. Additionally, this sentence is not properly written and thus confusing.
  *(l. 109-111) Revised.*

Line 113-114 How is "28.1±0.9°C for the open ocean and reef and 28.5±0.6°" different from each other? Did you do a statistical test of the mean difference? What is ±0.9ºC? the standard deviation of the mean or standard error of the mean? This is a poorly constructed sentence and all abbreviations should be defined at first use. Revise "Analysis of SST determined from the Advanced Very High-Resolution Radiometer (AVHRR) satellite SST product (Casey et al., 2010) for the varying

grid areas in Chagos (open ocean (give area in degrees) and lagoon (give area in degrees) reveals differences in SST means and seasonality at Chagos depending on the reef setting.

This analysis finds 28.1±0.9°C for the open ocean and reef and 28.5±0.6°C for the lagoon setting averaged over the interval from 1997–2012 (Fig. 3). Not sure what Leupold et al., 2019 is a reference for from the sentence construction. If this figure and analysis is from the study Leupold et al., 2019, this should be made clearer and referenced in Figure 3.

*(l. 116-118) We actually wanted to state that the main difference between both settings is rather SST variability than the mean SST and corrected this. We also revised the sentence as suggested by the reviewer.*

Line 119-120 Revise for improper use of /, use a hyphen. "such as El Niño in 1997-1998 or La Niña in 2010-2011".

*We revised the use of "/" as suggested by the reviewer with one exception: we did not change e.g. 1877/78 to 1877-1878 and similar cases (i.e. kept 1997/98), because in this case "/" is part of a name (e.g. the El Niño event 1997/98) and does not refer to a time period as in e.g. "Little Ice Age (1836-1867)".*

Line 119-120 What do you mean by "Both anomaly records are not significantly different (t-value = 0.34; p-value = 0.37)"? Is this a statistical test for the means, variance, or something else? You can also look at correlation to describe co-variance, which is more interesting for an SSTA time series looking at ENSO than if the means and or variance are the same or different.

*(l. 124) Yes, it is a common statistical test for the means; it is a t-test. Both values (t- and p-value) are the characteristic parameters for a t-test.*

Line 121 The reader does not know anything your coals or about the location your corals yet in the text, revise the text as needed, and refer to Fig. 1 where you have the coral locations.

*(l. 124) We do not agree with the reviewer. The entire Section 2 deals with the study site and the location (Chagos) and its climate are sufficiently introduced in Sections 2.1 and 2.2. The fact that this study investigates SST recorded by corals is explicitly mentioned at the end of Section 1. The sentence in Line 121 concludes on the information provided in the very same paragraph and does not need any further clarification.*

Line 125 Revise "We therefore use various ENSO indices for comparison with our coral data…"

*(l. 129) Revised.*

Line 128-129 "TexMex" is not the correct geographical term, use "Texas-Mexico". Define USA abbreviation at first use. Revise "and other locations in the Tropics". Revise "The annually-resolved El Niño Index Niño3.4…reconstructs past El Niño and La Niña events back to 1607." Why is Niño3.4 in italics in this usage? Do not use italics to make it appear different, give it a name different from Niño3.4, which is a defined index used by climatologists to determine ENSO, see https://climatedataguide.ucar.edu/climate-data/nino-sst-indices-nino-12-3-34-4-oni-and-tni. This is not what you mean. Wilson reconstructed the Nino3.4 index in their study but it is not the Nino3.4 index. Refer to it as Wilson Nino3.4 or reconstructed Nino3.4 to make it clear you are referring to the reconstruction and not the average of instrumental SST for 5N-5S, 170W-120W. Even better revision to this sentence "The study of Wilson et al. (2010) reconstructs an annually-resolved Niño3.4 index …of past El Niño and La Niña events back to 1607 beyond the instrumental era, which we will refer to as the "Wilson Niño Index."

*(l. 130-133) We revised this sentence.*

Line 129-130 Delete "We use the Wilson Niño Index for comparison with our coral SST records performing Wavelet Coherence Analysis in the time domain (see section 4.4)." This does not need to be stated here but in your results.

*(l. 133) We will not delete this sentence, because it explains for what we use this index in our study. Such an introduction was requested by Reviewer 1 in the previous review process.*

Line 135 Explain what you mean by "it should be relatively independent from statistical biases". Historical records interpreted with quantitative or qualitative methods do have a bias, just different biases from a coral or proxy biased reconstruction. See Paleoclimatology textbook by Bradley 2015 and the chapter on Historical reconstructions and Garcia-Herrera, R., Konnen, G., Wheeler, D., Prieto, M., Jones, P. & Koek, F. 2005: CLIWOC: A Climatological Database for the World's Oceans 1750-1854. Climatic Change 73, 1. and Ingram, M. J., Underhill, D. J. & Wigley, T. M. L. 1978: Historical climatology. Nature 276, 329-334, http://dx.doi.org/10.1038/276329a0.

*(l. 138) We agree with the reviewer and consequently revised the sentence.*

Line 138-140 This sentence is redundant and not needed here but in the methods or results section, Delete or move "We use both indices by Quinn (1993) and Brönnimann et al. (2007) for identifying past warm and cold events in each coral record and we use these events to compile composites (see section 4.5)."

*(l. 141) We will not delete this sentence, because it explains for what we use which index in our study. Such introduction was requested by Reviewer 1 in the previous review process.*

Line 150 Check this exposure time for the x-ray process, this is a really long time. It usually takes a fraction of a second unless this is an ancient machine. If you used digital plates for the X-ray, I doubt the exposure time is that long.

*(l. 152) We checked the exposure time. The exposure time for the X-ray machine we used is indeed 1-2 minutes. This produces very good X-ray images as it can be seen in the corresponding figures we provided in the supplementary material.*

*Note that the exposure time depends on the amperage, which can be determined by the user. Decreasing the amperage increases the exposure time, but from our experience this can help to produce better X-ray images (it is easier to optimize longer exposure times to best visualize the density bands of a coral slab).*

Line 200 The reference Groth and Ghil (2015) is for a Monte Carlo Singular Spectrum Analysis (SSA) not wavelet coherence. Please use the correct reference for Wavelet coherence, Grinsted et al. (2004) doi:10.5194/npg-11-561-2004 is the one I use.

*(l. 204) Thank you for pointing this out. We corrected the reference.*

Line 210 You mean "section" not "chapter". This is not a book or dissertation.

*(l. 207 + 212) Revised.*

Line 212-213 Eq. 1 The equation should be divided by the square root of the degrees of freedom. Data with sinusoidal cycles is not independent and violates the assumptions of many statistical tests. Your "n" values should be adjusted for degrees of freedom. It takes only two terms to define a sinusoid, the wave height, and length (or period), thus two degrees of freedom. If you have 15 years of data with a seasonal cycle, the degrees of freedom should be about 30. There are quantitative methods to determine your degrees of freedom, Runs test is the simplest dof test for a single data series and it is non-parametric. This adjustment is important because the larger your

n values, the smaller your error, therefore, if you use unadjusted n values, your analysis could result in false positives.

*(l. 216) We do not agree with the reviewer, because we do not apply Eq. 1 to data following a sinusoidal cycle. As stated in lines 212/213 Eq. 1 is applied to the data of the composite records. Composites are used to evaluate temperature anomalies identified in the coral records after removing the sinusoidal seasonal cycle. Consequently, the temperature anomalies are not data with sinusoidal cycles.*

*However, as all data are biased their statistical evaluation has to account for theses biases, which we do with Eq. 1. As the reviewer states "the larger your n values, the smaller you error", we calculate the standard error (Eq. 1), which also considers n, and not only the standard deviation, which does not consider n. Using the standard error allows us to better asses the bias associated with small values of n.*

Line 217 Revise for incorrect usage of which "…coral samples, that show a good…"
*(l. 219) Revised.*

Section 4.2 This is a short section with 9 lines of text, the two subsections are not needed.
*(l. 228) Adjusted.*

Table 2 Explain what median RSD% is, is this from your ICP-OES analysis? That is the only other place RSD has been mention up to line 226.
*(l. 705) We added this information as a note to the table.*

Line 230 The number of samples appears twice in the sentence, same for the other sentences for the other corals in this section. The two subsections basically put what is in table 2 into text. The reader can read the table. Tell the reader something else about your results, such as means and medians are the same, thus these corals are not biased towards one season. B8 has a different mean, this one reason we removed the mean from the coral Sr/Ca records. The ranges vary among the corals with E5 having the greatest range and E3 the smallest. Rewrite this whole section to be more informative. Table 2 is for your raw Sr/Ca data, do these statistics change after linear interpolation? That is more interesting to me.
*(l. 228-240) We revised this paragraph and added the description of the results as suggested by the reviewer.*
*Leupold et al. (2019; https://doi.org/10.1029/2018GC007796) show that the statistics of raw Sr/Ca do not change after interpolation.*

Line 241 Revise "Such decadal variability in the Indian Ocean is described in previous studies". Change tense and redundant in word usage.
*(l. 242) Revised.*

Line 245 How were the coral Sr/Ca records detrended? Linear trend, polynomial?
*In the "point-to-point reply" to Reviewer 2 previously in the review process we already showed how the coral Sr/Ca records were detrended:*
*l. 112-114: "For detrending we used published methods by Mudelsee (2000; https://doi.org/10.1016/s0098-3004(99)00141-7) and Mudelsee 2009; https://doi.org/10.1140/epjst/e2009-01089-3). Detrending was necessary to compile the composite records."*
*l. 689-694: "These programs are able to remove not only linear trends (which was the case with E3 and GIM as it can be seen in Figure S7 of the revised version of the supplementary material), but*

*they find breakpoints in time series where long-term trends change and calculate linear functions for these periods which are used for subtracting these long-term trends from the original time series. For example, coral record E5 in Figure S7 shows 4 linear graphs in red that were calculated with these programs overlying the original time series."*

*(Please note that Fig. S7 is now Fig. S8 in the revised version of the supplementary material.)*

*This means, four and three long-term linear trends were subtracted from E5 (1675-1716) and B8 (1836-1867), respectively, and one long-term linear trend was subtracted from E3 (1870-1909) and as well as from GIM (1980-1995).*

*The published methods we used are cited in the supplementary methods section of the supplementary materials.*

Line 247 Are the mean annual cycles determine from the detrended data? make this clear. Include a figure with the detrended coral Sr/Ca series so the reader can see how you detrended the data. "Mean annual cycles" is confusing, do you mean annual values or seasonal cycles. In line 248 the authors use "seasonal amplitudes in coral SST" that is clearer.

*"Mean annual cycles" is a commonly used term in this kind of paleoclimate analysis. For a detailed explanation please refer to the interactive discussion of this manuscript, author comment 2 (AC2), page 6, lines 10-14.*

Line 250 I could not find in the supplementary material and Fig. S6 where the "26-32% of the coral-SST variance" is explained. The section 8 in the supplementary material notes the SSA is done with monthly anomalies, so the seasonal cycle is already removed. To figure out the % variance in seasonal cycle. Take the variance of the monthly Sr/Ca (before detrending) – variance of the monthly Sr/Ca anomalies with the seasonal cycle removed. That is your % variance due to the seasonal cycle. Can do the same with before and after detrending.

*The "26-32% of the coral-SST variance" is explained in the supplementary material, lines 61-63. SSA was performed with both original and detrended data. This is described in Section 6 of the supplementary material.*

Line 271 State explicitly what is meant by "in recent periods".

*(l. 273) Revised.*

Line 269 How is an "anomaly events" events defined? >0.5ºC or something else? Give explicit details.

*Line 290 (now line 292) of the manuscript explains how we interpret an anomaly event: "when the anomaly exceeds 1.5 standard deviations of the mean of each coral record…."*

Line 290 Fig S7 is the detrending plot with the anomalies. Move this figure to the main paper.

*We will not move this Figure S7 (now Fig. S8) to the main paper, because Fig. S8 visualizes an aspect of the methodological approach used but is not required to understand the approach nor the results of their discussion and interpretation.*

Line 299 Revise "These sub-periods were selected because…"

*(l. 301) Revised.*

Line 400-411 "In summary…" this should be in the conclusion section.
The conclusion basically repeats lines 400-411. Conclusions should not have numerical results, that would be in the results or discussion section. I suggest just moving lines 400-411 to the conclusion and delete most of the present conclusion sentences.

*There are quite a few citations in lines 400-411 (now 403-414) used to compare our findings with other work and to elaborate on the implications of the study. These features clearly identify these lines as part of the discussion. Furthermore, citations should not be used in the conclusions. Therefore, we will not move these lines to the conclusion section.*

*We do not agree that the conclusion basically repeats lines 400-411. In addition, the content of the conclusion can be clearly identified as a conclusion whereas lines 400-411 can be clearly identified as part of the discussion as has been shown above. Furthermore, while not every paper might have numerical results in its conclusion section, providing these kind of information is not completely unusual, as can be seen in some articles published in Climate of the Past (e.g. https://doi.org/10.5194/cp-16-523-2020, https://doi.org/10.5194/cp-16-299-2020, https://doi.org/10.5194/cp-16-1187-2020).*

[revised manuscript text omitted]
. S8), singular spectrum analysis (SSA) plots (Figs. S9-S11) and power spectrum analysis plots of detrended coral SST (Fig. S12) including text descriptions. Two additional tables giving the years of El Niño and La Niña events used for the composite maps (Table S1) and the linear regression results between the coral SST records and the Wilson Niño Index (Table S2) are also part of this supplementary material.

**1 Supplementary Methods**

**Indices**

We use Niño 3.4 SST anomalies taken from NOAA ERSSTv5 (Huang et al. 2017) for power spectrum analysis (Fig. S12). These have been interpolated from sparse observational data and extend back until 1870.

**Statistics**

Power spectra analysis was performed twice using the spectral analysis function REDFIT (Welch window) of the open source software *PAST* (version 3.25; Hammer et al., 2001). One run was performed with the time series before detrending, one run after detrending. Every time series was detrended using the softwares *breakfit* (Mudelsee, 2009) or *rampfit* (Mudelsee, 2000), respectively (Fig. S8).

SSA (Vautard and Ghil, 1989) were generated using the *MATLAB* (version R2019b) software toolboxes by Groth and Ghil (2015).

**2 Table with years of events that were used in the composite maps (Fig. 2 of the main document)**

| Event years | |
|---|---|
| El Niño | La Niña |
| 1982/83 | 1984/85 |
| 1986/87 | 1988/89 |
| 1987/88 | 1995/95 |
| 1991/92 | 1998/99 |
| 1994/95 | 1999/00 |
| 1997/98 | 2007/08 |
| 2002/03 | 2010/11 |
| 2009/10 | 2011/12 |
| 2015/16 | |

**Table S1: El Niño and La Niña event years used for the composite maps. Between 1982 and 2016, 9 El Niño events and 8 La Niña events occurred. Temperature anomalies from December to February were averaged for each event.**

**3 Pictures of coral sample sites**

[Figure]

**Figure S1: Pictures of coral sample sites. (a) Boulder beach at Eagle Island where the samples E5 (1675-1716; blue arrow) and E3 (1870-1909; red arrow) were collected. (b) and (c) a derelict building at Boddam Island from which the sample B8 (1836-1867) was collected.**

**4 X-ray images**

[Figure]

35 **Figure S2: X-ray images of coral samples analyzed in this study with raw Sr/Ca data (dark blue lines with overlapping intervals (OI) in light blue). Age models were interpreted using two U/Th measurements from each sample (sampling points for U/Th dating are indicated with circled numbers and light red-shaded areas; for determined ages see Table 1). Red lines indicate subsampling paths. Blue-shaded areas indicate sampling locations for subsamples used for Scanning Electron Microscopy (SEM). Please note that the slab of sample B8 is uneven as the slab was too brittle to polish out saw cuttings from field work and these are still seen on**
40 **the X-ray image. Note the even growth patterns of all samples.**

**5 Thin section and SEM analysis images**

[Figure]

**Figure S3:** Photomicrographs of coral sample E5 (1675-1716). Double arrows indicate corresponding photomicrographs. (a) PPL and (b) XPL overview photomicrograph of the coral skeleton. (c) PPL and (d) XPL photomicrograph of the sample at higher resolution where minor amounts of secondary calcite cement are visible (red circle). (e) SEM overview and (f) detail image (red box in e) where only trace amounts of sugary cements can be found.

[Figure]

**Figure S4: Photomicrographs of coral sample B8 (1836-1867). Double arrows indicate corresponding photomicrographs. (a) PPL and (b) XPL overview photomicrograph of the coral skeleton. (c) PPL and (d) XPL photomicrograph of the sample at higher resolution where small amounts of secondary aragonite cement are visible (red oval). (e) SEM overview and (f) detail image of B8 (1836-1867). Small amounts of sugary aragonitic cement can be seen.**

[Figure]

55  **Figure S5: Photomicrographs of coral sample E3 (1870-1909). Double arrows indicate corresponding photomicrographs. (a) PPL and (b) XPL overview photomicrograph of the coral skeleton. (c) PPL photomicrograph of the sample at higher resolution where small fragments of aragonite are found (red circle). (d) PPL microphotograph of the sample at higher resolution without any signs of diagenesis. SEM images showing (e) microborings (red arrows) and (f) areas which appear brighter due to dissolution.**

**6 Seasonal cycles inferred from Singular Spectrum Analysis**

Singular spectrum analysis (SSA) of the coral records with seasonal cycles reveal large interannual to decadal SST variabilities during both the 17-18th century and 19-20th century (not shown). The reconstructed components 2 and 3 (RC2, RC3) produced by SSA describe seasonal amplitudes for all samples and explain 28% (E5), 26% (B8), 32% (E3) of the coral Sr/Ca-SST variance. Decadal variabilities are larger during the 17-18th century compared to the 19-20th century. The first reconstructed component (RC1) of E5 with seasonal cycles explains 48% of the coral Sr/Ca-SST variance and describes a periodicity of around 18 years. The coral records covering the 19-20th century do not show a strong decadal component in SSA. Instead, RC1 explains 49% (B8) and 39% (E3) of the coral Sr/Ca-SST variance and describes a periodicity of around 7 years, which can be interpreted as ENSO periodicity.

The SSA results were validated by power spectrum analysis of bimonthly resolved coral SST anomalies, which were not detrended (Fig. S6). For this analysis, the coral record GIM (Pfeiffer et al., 2017) was included, which extends from 1880 to 1995. Power spectrum analysis of E5 (Fig. S6a) shows a low-frequency band corresponding to a periodicity of 18-19 years, identical to RC1 of the SSA. In addition, RC2 describing an ENSO periodicity of 4-5 years is confirmed by the second highest low-frequency band in power spectrum analysis of E5. The power spectrum analysis of the corals covering the 19-20th century (B8 and E3) confirms their SSA results, as well (Figs. S6b & c). It shows high power on the low-frequency (5-6 years for B8; 6-7 years for E3) band, which was also described by the first reconstructed components in SSA. Power spectrum analysis for GIM (Fig. S6d) reveals high power on the ENSO band (4-5 years and 8 years) and the highest power at the decadal frequencies (26 years).

80

[Figure]

**Figure S6: Power spectrum analysis of each Chagos coral bimonthly resolved anomaly series.**

**(a) E5 (1675-1716)**

6.17 1.84 -2.5 -6.83

**(b) B8 (1836-1867)**

4.92 0.58 -3.75 -8.08

**(c) E3 (1870-1909)**

4.8 0.47 -3.87 -8.2

**(d) GIM (1880-1995)**

5.29 0.96 -3.37 -7.71

**Figure S7: Wavelet power spectra of (a) E5 (1675-1716), (b) B8 (1836-1867), (c) E3 (1870-1909) and (d) GIM (1880-1995) coral Sr/Ca records. Wavelet power spectra were computed using the Morlet wavelet. The cone of influence and the 95% confidence level are indicated by the black lines. All spectra were computed with the free software package PAST (version 3.25; Hammer et al., 2001).**

**8 Detrending of coral SST records**

[Figure]

(a) E5 (1675-1716)  (b) B8 (1836-1867)  (c) E3 (1870-1909)  (d) GIM (1880-1995)

**Figure S8:** Sr/Ca-SST anomalies with calculated trend lines (red lines; upper plot) and anomalies after detrending (lower plot; with plotted 1.5x of the standard deviation as dashed lines) for the coral records (a) E5 (1675-1716), (b) B8 (1836-1867), (c) E3 (1870-1909) and (d) GIM (1880-1995).

**9 Interannual SST variability inferred from Singular Spectrum Analysis**

The spectral results of the coral records with seasonal cycles were validated by singular spectrum analysis (SSA) of coral SST anomalies records and power spectrum analysis, to reveal stronger patterns of variance when seasonal cycles were subtracted (Figs. S9-S12). During the 17-18th century, the coral record shows a periodicity of 18 years in RC1, which explains 47% of the coral Sr/Ca-SST variance (Fig. S9). The second reconstructed component (RC2; Fig. S9) of E5 (1675-1716) explains 14% of the coral Sr/Ca-SST variance and describes an ENSO periodicity of 4-5 years. During the 19-20th century, the pattern of variance describing the ENSO periodicity in the coral records are found in two to three reconstructed components: For B8 (1836-1867), RC2 and RC3 describe an ENSO periodicity of 5-8 years with in total 62% of the corals Sr/Ca-SST variability (Fig. S10). For E3 (1870-1909) it is even higher with RC1-3 explaining 65% of the coral Sr/Ca-SST variance. Those three components describe a characteristic ENSO periodicity of 3-8 years (Fig. S11).

Power spectra of detrended coral SST time series all show the typical ENSO periodicity between 3 and 8 years (Fig. S12a-d). Those periodicities can also be found in the power spectra of the Niño3.4 indices (Fig. S12e & f). Even after detrending, the power spectrum of the GIM coral SST record still shows the highest power at low-frequencies, which translates to a period of 21-22 years.

[Figure]

**Figure S9: Reconstructed components from Singular Spectrum Analysis of E5 (1675-1716) Sr/Ca monthly anomalies. First reconstructed component (RC1) describes a periodicity of 18 years. RC2 and RC3 describe typical ENSO periodicities.**

[Figure]

**Figure S10: As Figure S9, but for coral Sr/Ca record of B8 (1836-1867).**

[Figure]

**Figure S11: As Figures S9 and S10, but for coral Sr/Ca record of E3 (1870-1909). E**NSO periodicities are described by all shown
reconstructed components RC1-3.

[Figure]

**Figure S12: Power spectrum analysis plots for detrended coral SST, the annually resolved Wilson Niño index (Wilson et al., 2010) and the monthly resolved Niño3.4 index based on NOAA ERSSTv5 (Huang et al., 2017) time series.**

**10 Linear regression**

Ordinary least square (OLS) regression and *PearsonT3* calculation results reveal no significant linear relation between annual coral SST records and the Wilson Niño index (Table S2).

| Method | Coefficient | E5 (1675-1716) | B8 (1836-1867) | E3 (1870-1909) | GIM (1880-1995) |
|--------|-------------|----------------|----------------|----------------|-----------------|
| Excel OLS | $R^2$ (p-value) | 4.09E-5 (0.9679) | 0.0006 (0.8979) | 0.0027 (0.7502) | 0.0444 (0.0232) |
| PearsonT3 | r [95% confidence interval] | -0.006 [-0.361; 0.350] | -0.024 [-0.739; 0.716] | 0.052 [-0.418; 0.500] | 0.211 [-0.005; 0.408] |

125 **Table S2: Correlation coefficients of given coral records with the Wilson Niño index.**

**Supplementary References**

130 Groth, A., and Ghil, M.: Monte Carlo Singular Spectrum Analysis (SSA) revisited: Detecting oscillator clusters in multivariate datasets, J. Climate, 28, 7873-7893, https://doi.org/10.1175/JCLI-D-15-0100.1, 2015.

Hammer, Ø., Harper, D. A. T., and Ryan, P. D.: Paleontological statistics software: package for education and data analysis, Palaeontol. Electron., (4), 2001.

Huang, B., Thorne, P. W., Banzon, V. F., Boyer, T., Chepurin, G., Lawrimore, J. H., ... and Zhang, H. M.: Extended 135 reconstructed sea surface temperature, version 5 (ERSSTv5): upgrades, validations, and intercomparisons, J. Climate, 30(20), 8179-8205, https://doi.org/10.1175/jcli-d-16-0836.1, 2017.

Mudelsee, M.: Ramp function regression: A tool for quantifying climate transitions, Comput. Geosci.-UK, 26(3), 293-307, https://doi.org/10.1016/s0098-3004(99)00141-7, 2000.

Mudelsee, M.: Break function regression: A tool for quantifying trend changes in climate time series, Eur. Phys. J.-Spec. Top., 140 174(1), 49-63, https://doi.org/10.1140/epjst/e2009-01089-3, 2009.

Pfeiffer, M., Zinke, J., Dullo, W. C., Garbe-Schönberg, D., Latif, M., and Weber, M. E.: Indian Ocean corals reveal crucial role of World War II bias for twentieth century warming estimates, Sci. Rep.-UK, 7(1), 14434, https://doi.org/10.1038/s41598-017-14352-6, 2017.

Vautard, R., and Ghil, M.: Singular spectrum analysis in nonlinear dynamics, with applications to paleoclimatic time series, 145 Physica D, 35, 395-424, https://doi.org/10.1016/0167-2789(89)90077-8, 1989.

Wilson, R., Cook, E., D'Arrigo, R., Riedwyl, N., Evans, M. N., Tudhope, A., and Allan, R.: Reconstructing ENSO: the influence of method, proxy data, climate forcing and teleconnections, J. Quaternary Sci., 25(1), 62-78, https://doi.org/10.1002/jqs.1297, 2010.